# Religion and educational mobility in Africa

Alberto Alesina[1], Sebastian Hohmann[2], Stelios Michalopoulos[3✉] & Elias Papaioannou[4✉]

The African people and leaders[1,2] have long seen education as a driving force of development and liberation, a view shared by international institutions[3,4], as schooling has large economic and non-economic returns, particularly in low-income settings[5]. In this study, we examine the educational progress across faiths throughout postcolonial Africa, home to some of the world's largest Christian and Muslim communities. We construct comprehensive religion-specific measures of intergenerational mobility in education using census data from 2,286 districts in 21 countries and document the following. First, Christians have better mobility outcomes than Traditionalists and Muslims. Second, differences in intergenerational mobility between Christians and Muslims persist among those residing in the same district, in households with comparable economic and family backgrounds. Third, although Muslims benefit as much as Christians when they move early in life to high-mobility regions, they are less likely to do so. Their low internal mobility accentuates the educational deficit, as Muslims reside on average in areas that are less urbanized and more remote with limited infrastructure. Fourth, the Christian–Muslim gap is most prominent in areas with large Muslim communities, where the latter also register the lowest emigration rates. As African governments and international organizations invest heavily in educational programmes, our findings highlight the need to understand better the private and social returns to schooling across faiths in religiously segregated communities and to carefully think about religious inequalities in the take-up of educational policies[6].

Africa hosts vibrant Christian and Muslim communities and, given the demographic trends, it will be home to the largest numbers of both creeds in the coming decades[7]. Religiosity is high, with 75% of Christians and Muslims attending a church and a mosque at least once a week, according to the 2016 Afrobarometer Surveys. Nevertheless, the most important divide between African Muslims and Christians is not their religion. As we show here, educational gaps circumscribe the religious landscape. This is not a recent phenomenon. At independence, Christians enjoyed more schooling, reflecting colonial investments and missionary activity[8–10]. In several countries, mostly in West Africa, primary school completion for Christians was more than double that of Muslims or Africans adhering to local religions (Supplementary Information, section A). Although the share of Africans with no schooling has declined from two-thirds for those born in the 1950s to less than half for the 1990s cohort, religious differences persist. For example, in Nigeria (Africa's most populous country, roughly equally split between Christians and Muslims), the primary school completion rate of Christians is 0.88, whereas it is 0.57 for Muslims. In Ethiopia, Africa's second-most populous country, 29% of Christians (about two-thirds of the population) born in the 1990s have completed primary education, with Muslims registering 16%. African Muslims are more educated only in a few countries where they are small minorities—South Africa, Zambia and Rwanda (about 1–2% of the populations of these countries) and Uganda (11%).

Policy briefs and reports[7,11] have discussed the considerable Muslim–Christian differences in education levels but a comprehensive account of the evolution of interdenominational educational gaps and their determinants is missing. Although some pioneering case studies analyse the role of religion in Nigeria[12,13], looking among others at the Yoruba[14,15], there has been limited comparative work on religion, as the vibrant research in economics on the political economy of African development mainly focuses on ethnicity[16–18]. Similarly, research on the interplay between religion and economic performance[19,20] often focuses on the role of Islam in the Middle East and Asia[21]. It is in Africa, though, that the interreligious gaps in education are striking and a plethora of narratives point to their rising salience[22]. An exception is the parallel work of refs. 23,24, who report considerable educational differences between Christians and Muslims in 11 African countries in 2017, showing further that gaps are considerable in predominantly Muslim areas. Focusing on Malawi, Nigeria and Uganda, these studies stress the role of religious leaders and social norms. Most importantly, the considerable body of research quantifying the role of educational reforms and government policies in the continent has not considered the role of religion[25]. Likewise, recent works mapping the intergenerational transmission of human capital across countries[26,27], US states[28], Chinese provinces[29] and African regions[30–32] do not study the role of religion. Others[33] explore regional, caste and religious differences in intergenerational mobility in education (IM; also referred to as educational mobility) across India; and like us, they uncover lagging educational mobility for the Muslim population, which contrasts with the rising mobility among the low socioeconomic status castes.

[1]Department of Economics, Harvard University, Cambridge, MA, USA. [2]Sihlquai 10, Adliswil, Switzerland. [3]Department of Economics, Brown University, Providence, RI, USA. [4]London Business School, London, UK. ✉e-mail: smichalo@brown.edu; eliasp@london.edu

We construct statistics on faith-specific IM across 21 African countries and 2,286 regions and uncover substantial differences between Muslims, Christians and Africans adhering to Traditional (Folk) religions (Animists). (We use the terms Traditionalists and Animists interchangeably throughout). We then trace the roots of these disparities. Interreligious differences in education levels at independence explain a substantial fraction of the observed variation. Nevertheless, even when we compare young Africans living in the same district with similarly (un)educated elders in their religious group, in households of comparable size and structure and in which the household head works in the same broad sector and occupation, IM gaps, albeit attenuated, endure. Muslim children underperform in regions where they are numerous compared to where they are a minority, a pattern that is not present for Christians and adherents of Traditional religions. Muslims benefit as much as Christians when they move early in life to high-IM regions. Given the explanatory power of religious segregation and the impact of regions on adherents of all creeds, we conclude with a primer on internal migration differences. Christians are much more likely to emigrate and exploit opportunities outside their birthplace in almost all countries. The low propensity of Muslims to move accentuates their initial educational disadvantage, as they typically reside in remote places, far from the capital and the coastline, with limited missionary activity and transportation investments. Muslims register the lowest emigration rates and largest IM deficits in these religiously segregated areas.

## Data and educational mobility statistics

We construct measures of IM across 2,286 districts in 21 African countries using information from the Integrated Public Use Microdata Series (IPUMS) International, which collects and harmonizes censuses, reporting representative (typically 10%) samples (see 'Data availability' in Methods). As the detail of religious denomination differs across censuses, we aggregate into Muslims, adherents of Traditional religions, Christians and two auxiliary categories (Other and No Religion). Extended Data Table 1 and Extended Data Fig. 1 give the shares in our 21-country sample (see also Data section). Our measures of educational IM reflect how children fare vis-a-vis their cohabiting elder members, mainly parents, in the household. We use primary school completion as the critical educational milestone for Africans born up until the 1990s, as secondary and college enrolment has increased mainly since the 2000s (Supplementary Fig. A2). Upward IM denotes the likelihood that 14–18-year-old children whose parents have not completed primary education manage to complete primary school; downward IM gives the likelihood that 14–18-year-old individuals whose parents have completed primary education will not manage to finish primary school. Our base sample consists of 7,188,717 children between 14 and 18 years of age. We also look at 14–25-year-old children as the sample increases to 13,018,904 with cohabitation rates of about 75%. We provide details in the Data and educational IM measures section and in Supplementary Information, section B.

## Religious IM across Africa
### Country patterns
Table 1 reports the newly constructed faith-based IM statistics across countries, the Christian–Muslim gaps in upward and downward IM and their level of statistical significance (Supplementary Information, section C provides additional measures.) With the exception of Mozambique, upward IM is lowest for Africans adhering to Traditional religions (column 4 of Table 1); it is less than 10% in Burkina Faso, Sierra Leone, Rwanda, Malawi, Uganda and Ethiopia. Upward IM is low for Traditionalists, even in countries with considerable representation. For example, in Togo where overall IM is 52.6%, Animists register 38.2%; and in Benin their 21.3% IM trails the country average by 8.5 percentage points. Christians enjoy the highest upward IM in 15 out of the

21 countries with a cross-country mean of 41.2%. The IM of Muslims exceeds that of Christians in South Africa (87.4% versus 74%), Zambia (47.6% versus 44.5%, although this difference is not significant at standard confidence levels) and Rwanda (27.4% versus 18.3%), where their shares are tiny (0.8–1.5%) and in Uganda (48.5% versus 40.4%), Mauritius (96% versus 87.8%) and Liberia (26.6% versus 21.8%), where Muslims constitute between 10% and 16% of the population. The Christian–Muslim gap in upward IM is considerable in many countries. In Nigeria, Muslim children of illiterate parents are 32 percentage points less likely to complete primary schooling than Christian children born to similarly uneducated parents (78.6% versus 46.6%). In Ethiopia, a country with very low primary school completion rates, the upward IM for Christians is 13.8%, whereas for Muslims it is 8.2%. The Christian–Muslim upward IM gap is largest in West Africa (that is, the mean is 22.1%), where Muslims form the majority (Guinea, Senegal and Burkina Faso) or substantial minorities (Nigeria, Benin, Cameroon and Ghana). Downward IM is high for Muslims (the mean is 27.5%) and Traditionalists (the mean is 42.6%). In West Africa, roughly one out of six Muslim children born to parents who completed primary schooling will fail to do the same. In Cameroon, the downward IM is 4.1% for Christians and a staggering 19.6% for Muslims. Downward IM for Nigerian Muslims is twice that of Christians (16.2% versus 7.8%). This pattern is reversed in countries with small Muslim communities.

### Regional patterns
Figure 1a portrays, for the 1980 and 1990 birth cohorts, the gap in upward IM between Christians and Muslims and Fig. 1b between Christians and adherents of Traditional religions across 1,731 and 1,071 African districts, respectively. (Supplementary Table C3 summarizes the districts' population and land area. Within countries, Muslim-majority districts are comparable in area to Christian). The depicted districts are fewer than the total number of districts as we need to observe in each district 14–18-year-old individuals born in the 1980 and 1990 cohorts from both religions. Figure 1c,d zoom into Ghana to illustrate the within-country variation. The Christian–Muslim gap is large in West Africa. (Supplementary Fig. C1 maps upward IM for Christians, Muslims and Traditionalists, respectively). Extended Data Table 2 reports regional statistics. Within-country variation is notable in Cameroon, Ghana, Guinea, Nigeria and Ethiopia. Christians fare better than Muslims in almost all the districts of Senegal and Burkina Faso, despite the numerical dominance of Muslims. In Ghana and Cameroon, the upward IM of Christians exceeds that of Muslims in three out of four districts. The Christian–Muslim upward IM gap is lower in the southern districts of West African countries, an issue we return to in the section on Religious IM across Africa. Educational outcomes for Muslims, Traditionalists and Christians move together as the cross-region correlation of religious upward IM is 0.60–0.72 and for downward IM 0.39–0.62 (Supplementary Table C6). However, Muslims and Traditionalists underperform compared to Christians, even when we zoom in on the same district. In the median region, upward IM for Christians is 0.44, 0.33 for Muslims and 0.21 for Traditionalists. Downward IM for Christians is 0.25, for Muslims 0.29 and 0.35 for Traditionalists (Supplementary Table C4).

## Explaining the religious IM gaps
### Approach
We estimate individual-level regressions that associate the IM of children with their own religious affiliation, using Christians as the omitted category. We drop 'other religion' and 'no religion' as these are not comparable across censuses and countries. A large body of research documents non-negligible differences in family arrangements, age of marriage, social practices and occupational specialization between Christians, Muslims and adherents of Traditional religions (Animists). To explore their role, we examine how the estimates on the Muslim and the Traditional religion indicators change as we account for:

## Table 1 | Country-group-level estimates of IM, ages 14–18 yr

| | Upward IM | | | | | Downward IM | | | | |
|---|---|---|---|---|---|---|---|---|---|---|
| | 1 | 2 | 3 | 4 | 5 | 6 | 7 | 8 | 9 | 10 |
| Country | Overall | Christian | Muslim | Traditional | Δ(c−m) | Overall | Christian | Muslim | Traditional | Δ(c−m) |
| Nigeria | 0.612 | 0.786 | 0.466 | 0.229 | 0.320*** | 0.096 | 0.078 | 0.162 | 0 | −0.084*** |
| Cameroon | 0.613 | 0.739 | 0.424 | 0.481 | 0.315*** | 0.056 | 0.042 | 0.196 | 0.185 | −0.154*** |
| Senegal | 0.244 | 0.527 | 0.235 | | 0.292*** | 0.264 | 0.163 | 0.274 | | −0.111*** |
| Botswana | 0.798 | 0.822 | 0.556 | 0.699 | 0.266 (0.0108) | 0.085 | 0.083 | 0.027 | 0.076 | 0.056 (0.038) |
| Benin | 0.298 | 0.415 | 0.214 | 0.213 | 0.201*** | 0.292 | 0.274 | 0.308 | 0.469 | −0.034 (0.006) |
| Ghana | 0.557 | 0.654 | 0.468 | 0.263 | 0.186*** | 0.173 | 0.157 | 0.263 | 0.471 | −0.106*** |
| Burkina Faso | 0.191 | 0.332 | 0.182 | 0.072 | 0.150*** | 0.235 | 0.199 | 0.269 | 0.569 | −0.070*** |
| Mali | 0.274 | 0.395 | 0.273 | 0.187 | 0.122*** | 0.237 | 0.219 | 0.237 | 0.491 | −0.018 (0.287) |
| Mozambique | 0.287 | 0.324 | 0.207 | 0.366 | 0.117*** | 0.249 | 0.225 | 0.314 | 0.220 | −0.089*** |
| Togo | 0.526 | 0.641 | 0.534 | 0.382 | 0.107*** | 0.190 | 0.165 | 0.214 | 0.361 | −0.049*** |
| Sierra Leone | 0.261 | 0.319 | 0.248 | 0.091 | 0.071*** | 0.332 | 0.257 | 0.385 | 0.600 | −0.128*** |
| Ethiopia | 0.116 | 0.138 | 0.082 | 0.017 | 0.056*** | 0.344 | 0.323 | 0.481 | 0.800 | −0.158*** |
| Guinea | 0.182 | 0.229 | 0.181 | 0.138 | 0.048*** | 0.439 | 0.500 | 0.418 | 0.724 | 0.082*** |
| Malawi | 0.133 | 0.143 | 0.096 | 0.095 | 0.047*** | 0.512 | 0.503 | 0.616 | 0.556 | −0.113*** |
| Egypt | 0.673 | 0.679 | 0.673 | | 0.006 (0.043) | 0.052 | 0.048 | 0.052 | | −0.004 (0.030) |
| Zambia | 0.438 | 0.445 | 0.476 | 0.449 | −0.031 (0.399) | 0.253 | 0.251 | 0.221 | 0.263 | 0.030 (0.261) |
| Liberia | 0.222 | 0.218 | 0.266 | 0.103 | −0.048*** | 0.538 | 0.537 | 0.544 | 0.632 | −0.007 (0.681) |
| Uganda | 0.400 | 0.404 | 0.485 | 0.019 | −0.081*** | 0.290 | 0.295 | 0.257 | 0.641 | 0.038*** |
| Mauritius | 0.917 | 0.878 | 0.960 | | −0.082*** | 0.018 | 0.028 | 0.014 | | 0.014 (0.006) |
| Rwanda | 0.181 | 0.183 | 0.274 | 0.077 | −0.091*** | 0.543 | 0.541 | 0.489 | | 0.052 (0.039) |
| South Africa | 0.731 | 0.740 | 0.874 | 0.764 | −0.134*** | 0.105 | 0.105 | 0.040 | 0.182 | 0.065*** |

The table reports upward and downward IM in educational attainment for the cohort born in the 1980s (the cohort with the broadest coverage) for individuals aged 14–18 yr by country and main religious group. Because of the timing of censuses, the values for Liberia, Mali, Nigeria and Togo correspond to those born in the 1990s. Columns 1–4 give upward IM estimates, whereas columns 6–9 give downward IM statistics. The numbers outside the parentheses in columns 5 and 10 show the IM gaps between Christians and Muslims. The values in the parentheses are $P$ values, where ***$P<0.001$. The $P$ values are computed using the formula for the distribution of the difference of two sample proportions, that is $2 \times (1 - \Phi(|\frac{IM_c - IM_m}{\sqrt{\frac{IM_c(1 - IM_c)}{N_c} + \frac{IM_m(1 - IM_m)}{N_m}}}|))$ where $IM_c$ and $IM_m$ are the (upward and downward) measures of IM for Christians and Muslims, respectively; $N_c$ and $N_m$ are the number of Christian and Muslim individuals entering into the computation of $IM_c$ and $IM_m$; and $\Phi$ is the standard normal cumulative distribution function. Countries are sorted by Christian–Muslim differences in upward IM (column 5). The censuses from Senegal, Egypt and Mauritius and Rwanda do not record Africans adhering to Traditional religions.

(1) household size and family structure variables, including the age of marriage; (2) proxies of household income (as, for example, the profession and the industry of employment of the household head); (3) the literacy of the older generation for the individual's coreligionists in the district; and (4) district constants interacted with urban–rural household status. These features may interact or be jointly determined by deeper factors. Our objective is not to identify causal effects but to examine the fraction of the variation in educational mobility gaps explained by these aspects. Figure 2 reports ordinary least-squares (OLS) estimates weighting by the population of a country to account for differences in IPUMS coverage, whereas Extended Data Fig. 2 reports country-specific estimates of Christian–Muslim differences. (Supplementary Information, section D gives results for various data subsets).

### Baseline IM gaps
The top bars simply condition on country cohort fixed effects and children's age constants to account for the increase in education over time. Among children whose parents have not completed primary education, Christians enjoy an advantage of 16 percentage points in completing primary school over Muslims and 20 percentage points over adherents of Traditional religions born in the same decade and country. Muslim children whose parents have completed primary education are 7 percentage points less likely than Christians to achieve the same qualification. If anything, Traditionalists have a lower rate of downward mobility than Muslims.

### Household structure and size
We condition on a rich set of household features: (1) size, distinguishing between individuals of the generation of the young, the head and the grandparents; (2) indicators for each 14–18-year-old individual regarding his/her relationship to the household head (for example, biological children, relative, nephews, spouse); (3) family organization indicators (for example, father and mother present, father or mother only, others only); and (4) mother's and father's age at their offspring's birth. We include all variables concurrently, as our objective is not to identify the most relevant one but to examine how much the religion coefficients move when we account for a saturated set of household traits. As we show in Supplementary Information, section D, there are important interreligious differences in household features. Muslim and Animist households are, on average, larger; 14–18-year-old Muslim girls are more likely to be spouses of the household head than Christian girls of the same age; and Muslim and Animist parents are, on average, younger at their children's birth than for Christians, reflecting earlier marriages. Despite these differences in the family organization, the educational gaps do not move much when we account for the former. The upward IM gap drops from 0.165 to 0.15 for Muslims and from 0.20 to 0.18 for adherents of Traditional religions. Household structure and size play a somewhat bigger role in narrowing the Christian–Muslim IM gap in West Africa but their influence is negligible in East and Southern Africa (Extended Data Fig. 2). Downward IM differences between

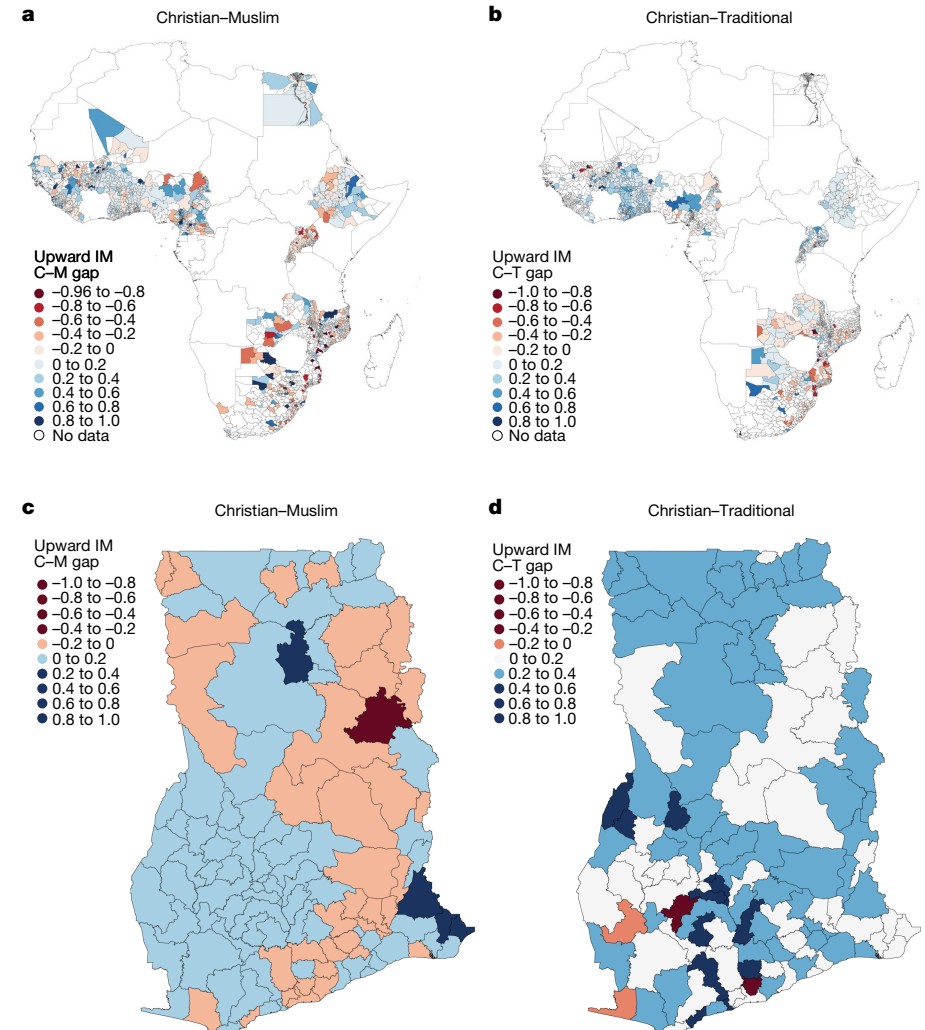

**Fig. 1 | Religious IM gaps across African districts. a,b,** Differences in educational upward IM between Christians and Muslims (**a**) and Christians and Africans adhering to Traditional religions (**b**) across all African districts (administrative level-2 and level-3 regions) in 21 countries. **c,d,** The Christian–Muslim (**c**) and the Christian–Traditionalist (**d**) difference in upward IM across districts in Ghana. We estimate absolute upward IM for young individuals, aged 14–18 yr, cohabiting with at least one older generation member in the household, usually a parent. Supplementary Figure C1 provides the mappings of upward IM for Christians, Muslims and Traditionalists used to compile the maps of Christian–Muslim and Christian–Traditionalist gaps in educational mobility.

Muslims and Christians are more affected by the inclusion of the rich set of household controls closing the overall gap by roughly 20% from 0.07 to 0.055.

### Household and parental occupational differences

We then turn to economic features, conditioning in model 3 of Fig. 2 on the following three (sets of) attributes (1) urban versus rural residence; (2) the industry of employment (six categories); and (3) the occupational specialization (ten categories) of the household head. As shown in Supplementary Information, section D, adherents of Traditional religions are more likely to live in rural areas (20%) and work in agriculture and are less likely to be professionals or skilled employees. Christian–Muslim differences in the employment industry, rural–urban status and profession are minor, perhaps masking larger ones owing to the coarse aggregation of IPUMS. In line with these patterns, differences between Christians and Muslims in living conditions and access to necessities in Afrobarometer Surveys and household income in the surveys[7] are also small (although Muslim households are larger). Variations in the industry of employment (broadly defined), occupation and urban–rural residence explain a non-negligible component of the IM differences between Christians and Traditionalists: the upward IM gap drops from

0.20 to 0.15 and the downward IM gap halves. These economic features do not explain the Christian–Muslim IM upward gap. If anything, the downward mobility gap becomes more pronounced when we compare children born in households of similar occupational structures.

### Initial literacy

In model 4 of Fig. 2, we control for the share of the older generation with completed primary education in the district for each religious group. This accounts for initial, group-specific regional development and schooling, stemming from the location of Christian missionaries, colonial educational investments and the spread of Islam[34]. The Christian–Muslim upward IM gap drops from 0.15 to 0.085. Likewise, the Christian–Muslim downward IM gap goes from 0.07 to 0.04, whereas the Christian–Traditionalist downward IM is eliminated. Differences in religious literacy rates of the older generation across districts seem first order, consistent with our earlier work[31], showing that initial literacy is the most important correlate of regional IM.

### Regional features

As recent studies[30–32] show that regions have a chief role in educational mobility, in model 5 of Fig. 2, we augment the specification

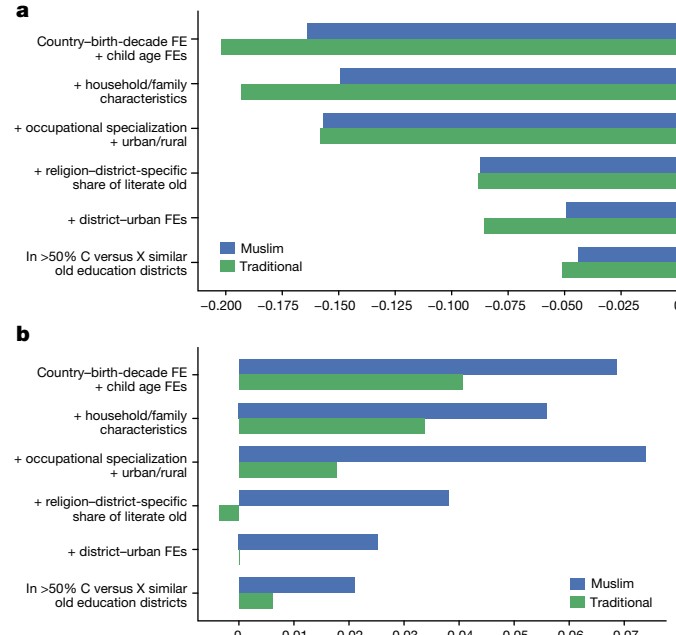

**Fig. 2 | Drivers of religious IM gaps. a,b**, Illustration of how the Christian–Muslim (blue bars) and the Christian–Animist (green bar) differences in upward IM (**a**) and downward IM (**b**) change as we add controls for household and family structure (model 2), parental occupational specialization, the industry of employment and rural–urban residence (model 3), the district's religion-specific illiteracy rate among the older generation (model 4), district times rural/urban fixed effects (model 5) and focusing on the top 50% most similar districts in each country in terms of the literacy gap of the elders in each religious group (model 6). The bars on the top (model 1) reflect the baseline interreligious differences in IM, simply conditioning on country–birth cohort fixed effects (FE) and age constants. The sample consists of Muslim, Animist and Christian young individuals (14–18 yr), matched to the previous generation in the household. The figure gives weighted linear probability model (OLS) estimates using the 1980 populations of the countries for the weighting to account for differential IPUMS sampling/coverage across countries. In models 1–5, the upward IM regressions are run in a sample of 4,989,952 young individuals who cohabit with older generation members who have not completed primary education and the downward IM regressions in a sample of 1,919,711 young individuals cohabiting with older generation members who have completed primary education. The figures omit standard-error bands to enable clear visualization of the patterns. The estimates on the Muslim indicator are statistically significant across all 12 specifications, whereas the Animist coefficient is significant across all upward IM specifications. C, Christian; X, Muslim or Animist.

with district-specific constants interacted with urban indicators (the interaction with the urban indicator does not affect the estimates). Regions explain a non-negligible fraction of the IM gaps, as there is considerable segregation across most African countries[35] and Muslims (and Traditionalists) reside in places with low opportunity (see below). The coefficient on the indicator for Traditional religions in the upward IM specification is not much affected (stays around 0.08), whereas the downward IM gap is eliminated. Accounting for regional features reduces the Christian–Muslim upward IM gap to 0.05 and the downward IM to 0.03. The explanatory power of regions for the Muslim–Christian mobility gaps seems first order in many countries, mainly in West Africa, where religious segregation is the highest (Extended Data Fig. 2). In Benin, Cameroon, Ghana and Senegal, the Christian–Muslim gap in upward IM halves when we account for residence (and initial differences in literacy). In Nigeria, the upward IM gap drops from 0.30 to 0.10, whereas in Mozambique, it goes from 0.08 to nil. This pattern echoes the patterns across India[36] of considerable heterogeneity in relative educational mobility across castes and religions in narrow geographic

areas. In the last row, we restrict estimation to half of the regions where interreligious differences in completed primary education of the older generation are the smallest to better account for 'initial' conditions in the district. The Christian–Muslim gap in upward IM continues to be close to 5 percentage points and 2 percentage points for downward IM.

### Taking stock

The analysis shows the following. First, differences in household structure between Muslims and Traditionalists, on the one hand, and Christians, on the other, explain a small fraction of educational mobility differences. Second, broad economic features have no role in the Christian–Muslim upward gap but reduce the Christian–Animist difference. Third, the literacy of the older generation in the district for each religious group and region-specific features explain roughly two-thirds of the interreligious IM gaps; this is especially the case in West Africa, where segregation is the highest. Fourth, Christian–Muslim educational gaps remain, even when we compare children with the same parental background and family composition residing in the same region (with comparable literacy rates in the older generation of the various religious groups).

## Sorting and childhood regional exposure

The prominence of regions gives rise to a plethora of questions ranging from their independent impact on the educational IM of the different denominations to unpacking the relevant regional characteristics. In this section, we address the former and, in the next section, the latter. Supplementary Information, section E gives descriptive statistics and further evidence.

### Empirical design

To isolate regional childhood exposure effects from spatial sorting, we use the approach of refs. 37,38 and leverage differences in the age of children's moves across districts[39]. The specification, detailed in the Methods, associates primary school completion for children of uneducated parents (upward educational IM) with the age of children's move and differences in educational mobility between birthplace ($o$) and destination district ($d$) among non-movers of the same cohort $b[\Delta_{odb} = \overline{\text{IM}}\_\text{up}^\text{nm}_{bd} - \overline{\%}\ \overline{\text{IM}}\_\text{up}^\text{nm}_{bo}]$. $\widehat{\text{IM}}\_\text{up}^\text{nm}_{b}$ summarizes all features of the economic, social, educational and institutional environment, shaping IM in a given district. These could reflect, among many others, local returns to schooling, school quality and quantity, accessibility, schooling fees and teacher–pupil ratios that differ considerably in low-income countries[40]. The age-specific parameters on destination minus origin IM, $\beta^\text{rel}_m$, capture how a child's probability of completing primary schooling varies with the age of their move to districts with higher or lower mobility. If regions matter for mobility, the earlier the move, the greater the impact. The variation comes from children born in the same place and decade moving to regions with different IM. Differences in the age-of-move slopes reflect the impact of an extra year in the high IM district—regional childhood exposure effects. The main identifying assumption, backed by our earlier work[31], is that the timing of moving for households is unrelated to children's ability.

### Sample

Not all censuses record the age of the move. These estimates come from a sample of 276,686 14–25-year-old individuals from 13 countries. These countries come from all major African regions; they are both former British and French colonies and non-colonized (Ethiopia); relatively poor and rich.

### Results

Figure 3 plots the age-of-move estimates, $\hat{\beta}^\text{rel}_m$, for Christians and Muslims against the child's age at the time of the household move. In Fig. 3a,b, origin–destination upward IM differences are calculated

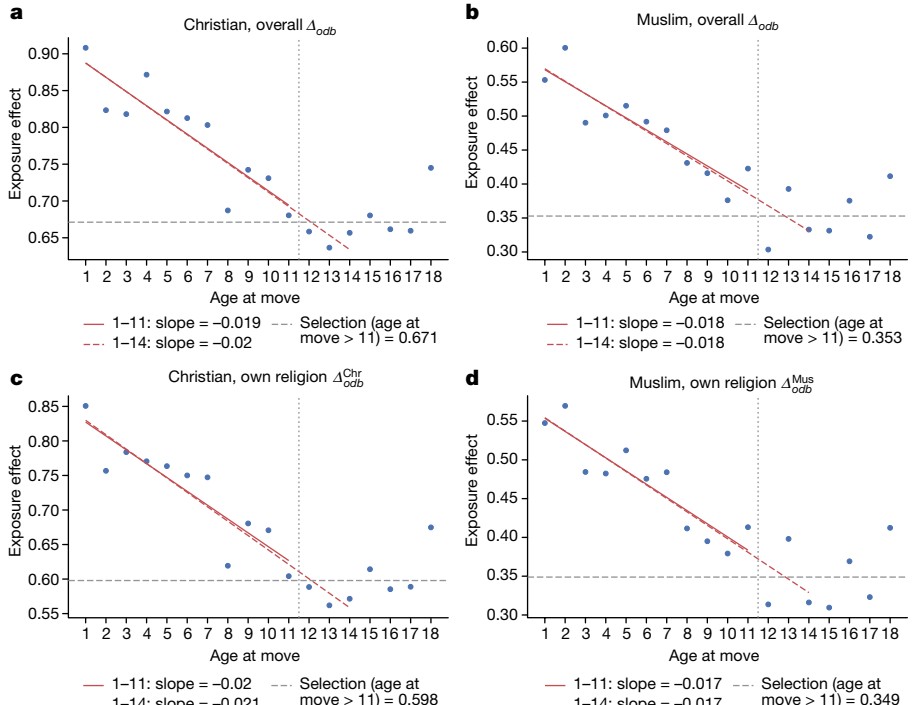

**Fig. 3 | Regional childhood exposure effects, semiparametric estimates.** Estimates of childhood regional exposure effects of upward IM. **a**–**d**, The figures plot age-of-move slopes on differences in upward IM (blue circles), estimated among non-movers of all faiths in **a** and **b** and of the same religion in **c** and **d** between origin and destination district against the child's age when the household moved. The dependent variable takes the value of one when the child of a household in which the older generation has not completed primary education completes primary schooling (upward IM). Panels **a** and **c** give estimates for Christians and **b** and **d** for Muslims. The dashed vertical line in **a**–**d** separates the data into moves before the age of 12 yr that are most relevant for primary education and after the age of 12 yr. Also shown is the OLS

regression fit of age-of-move slopes against the age of move before age 12 (red solid line) and age 14 (red dashed line). The slopes represent estimates of regional childhood exposure effects for primary school completion by ages 12 and 14 yr. The age-of-move estimates are statistically significant at standard confidence levels. The panels omit standard errors to avoid cluttering. The results for Christian children are based on 138,300 and 141,355 individual observations in the specifications with overall differences in non-movers and Christian-only non-movers, respectively. The Muslim results are based on 127,914 and 128,215 observations in the specifications with differences in IM among all non-movers and differences in IM among Muslims, respectively.

using all non-movers independently of religious affiliation, whereas in Fig. 3c,d, we use non-movers of the same religious group. The estimates for Traditionalists (not reported) are similar but imprecise, reflecting the small sample and the blending of heterogeneous religions. The figure uncovers two sets of patterns. First, the age-of-move slopes, $\hat{\beta}_m^{rel}$, are large for early-in-life moves and decline until around the age of 12–14 years. Children moving to 'better' districts earlier in life have a higher propensity to complete primary schooling. The relationship between age at move and exposure is negative and approximately linear, implying 'regional childhood exposure effects': moves in higher (lower) mobility districts are beneficial (detrimental) for younger kids. A further year of exposure before the age of 12 years for a child of illiterate parents to higher mobility district increases her chances for completing primary school by roughly 2 percentage points. Exposure effects are also 2% for Muslims and Christians when we compare them to their coreligionists. Both Muslim and Christian children benefit (lose) from early moves into high (low) upward mobility regions. Extrapolating over 14 years of childhood, Muslim and Christian children who move at birth to a district with one percentage point higher upward IM among Muslims will pick up roughly 30% of this difference through the impact of the region. Second, the slopes are significantly positive, even for children moving after 13–14 years. As moves after that age are unlikely to affect primary education, they reflect selection. Parametric specifications imposing a piecewise linear structure of exposure effects from 1 to 11 and from 12 to 18 yields similar regional childhood exposure estimates for Christians and Muslims. Besides, within-household specifications exploiting variation in the time of move across siblings

also yield regional childhood exposure effects of about 2% for both faiths (Supplementary Information, section E).

**Taking stock**

We obtained two results: (1) although spatial sorting is considerable, children whose families move earlier to areas where residents (of all faiths or of their religion only) have higher IM are more likely to complete primary school and (2) regional childhood exposure effects are comparable for Christians and Muslims.

## Religious IM gaps across African regions

Why do region-specific constants explain roughly half of the religious IM gaps? There are two possible explanations: (1) religious segregation, coupled with regional differences in educational opportunity, arising, for example, from more accessible and better-quality schools, make upward mobility more challenging for Muslims and adherents of Traditional religions and (2) the same regional feature may influence followers of different religions differently.

### Differences in residence across faiths

In Methods and in Extended Data Fig. 3 we explore how the residence characteristics compare across religious groups. On average, Muslims and Traditionalists reside in regions that, at the end of the colonial period, were less densely populated and more agriculture-oriented than Christians. Besides, Muslims and Traditionalists reside away from the capitals and the coastline; they live in districts with fewer colonial

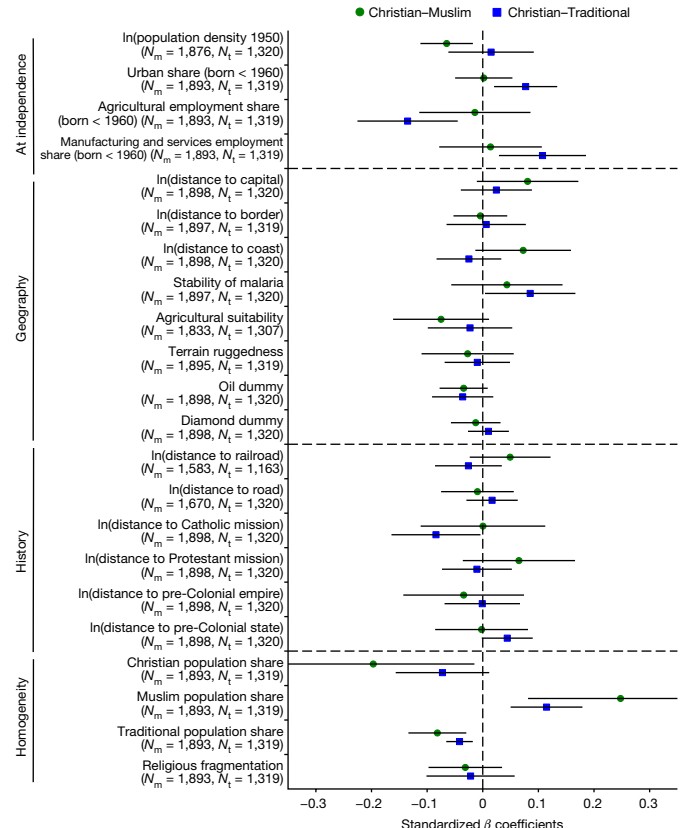

**Fig. 4 | District correlates of Christian–Muslim and Christian–Traditionalist upward IM gaps.** Correlations (standardized $\beta$ coefficients) between upward IM gaps of Christians and Muslims (green circles) and Christians and Traditionalists (blue squares), averaged across individuals in a district and four sets of regional characteristics: (1) proxies of development at independence (for example, population density and urban share); (2) location and geographic features (for example, distance to coast, border and capital); (3) historical aspects reflecting colonial-era investments and precolonial statehood; (4) religious homogeneity, proxied by the share of each religious group in the district and a frationalization index. Supplementary Information, section F gives variable definitions and sources. All specifications include country fixed effects (constants not reported). The 2 s.e. bands based on heteroskedasticity adjusted clustered at the country level are also reported. The estimates (green and blue symbols in the figure) were obtained by running separate regressions of the district-level Christian–Muslim and Christian–Traditionalist IM gap, respectively, on each district-level variable (indicated on the vertical axis of the figure). The IM gap is defined as the average IM of Christians minus the average IM of Muslims or Traditionalists in the district. Before running each regression, we standardize the dependent and independent variable by subtracting its sample mean and dividing it by its sample standard deviation.

investments, away from colonial railroads, roads and Christian missions. Consequently, the lower representation of Muslims in regions with better initial conditions is of first-order importance for explaining the observed Christian–Muslim IM gap.

## Approach

We estimate within-country district-level specifications associating regional Christian–Muslim and Christian–Animist differences in upward IM to various regional characteristics. Figure 4 reports standardized coefficients. As the literacy rate of the older generation is the strongest correlate of IM (Fig. 2), we also run specifications conditioning on it (Extended Data Fig. 6). Extended Data Figures 4 and 5 present regional specifications separately for adherents of Christianity, Islam and Traditional religions and compare the magnitudes of the

coefficients. Although the analysis does not have a causal interpretation, it allows for characterizing the geography of the considerable variation in religious educational IM within countries. Besides, it complements the evidence in Fig. 3, as the movers' design that distinguishes spatial sorting from childhood exposure effects does not pin down which regional features correlate with religious IM.

## Early development, historical investments and geography

Drawing on research on the roots of African development[41], we start examining the role of geographic, historical and at-independence features. As Extended Data Fig. 4 shows, upward IM is higher and downward IM is lower in more densely populated and urbanized regions, more specialized in services and manufacturing, close to the capitals, the coast and missionary activity and transportation infrastructure. However, these features do not explain the Christian–Muslim and the Christian–Traditionalist IM gaps, as the correlations are similar across groups.

## Segregation

We then turn to religious fragmentation and segregation. Our exploration is motivated by three observations. First, religious IM gaps are larger in segregated countries and Muslim educational mobility is the highest in countries with small Muslim communities. Second, US-centred research shows that racial segregation moves in tandem with underinvestments in education[42,43]. Third, recent work[44] shows that Muslims underperform (in education and health) in areas with precolonial Islamic states (for example, in Northern Nigeria and Cameroon and Senegal) owing to weak penetration of the colonial state and limited public-goods investments by missionaries. By contrast, in areas with modest Muslim communities, religious competition pushed Muslims (elites) to adopt Western education. Figure 4d looks at the association between the IM gaps and religious composition. Diversity, as captured by the Herfindahl index, is not a significant correlate. But the religious upward IM gaps are strongly correlated with segregation. The Christian–Muslim upward IM gap is significantly higher (lower) in predominantly Muslim (Christian) regions; Muslims underperform in districts where they are majorities. Evidently, the share of Muslims in the district is the strongest correlate of the Christian–Muslim upward IM differences. Figure 5 further unpacks this association. Figure 5a plots the Christian–Muslim gap in upward IM against the fraction of Muslims across districts, whereas Fig. 5b conditions on the religion-specific literacy rates of the older generation in the district that correlates strongly with IM. The strong positive association remains intact. Figure 5c,d plot the correlation between upward IM and own-religion share separately for Christians and Muslims, conditioning on the literacy of the elders. The likelihood that Christian children of parents without much education will complete primary school is similar in places with small, modest and prominent Christian communities. By contrast, completion of primary education for Muslims is high in regions with small Muslim communities but (very) low in (predominantly) Muslim districts. The negative association between educational opportunity and own-religion share for Muslims echoes US-based evidence that African American children underinvest in education in segregated communities and ghettos. Besides, it squares with India-based results[36], showing a negative association between caste segregation and relative educational mobility.

## Taking stock

We arrived at two takeaways. First, for all faiths, upward IM is higher and downward IM is lower in more developed regions, closer to the capital and the coast, with relatively more colonial investments. However, as Muslims and adherents of Traditional religions reside in less developed, more remote regions with less colonial infrastructure, they are at a disadvantage. Second, religious segregation seems instrumental (although the correlations do not identify causal effects). Christians do well independently of residence, whereas Muslim children underperform compared with other coreligionists in areas where Muslims

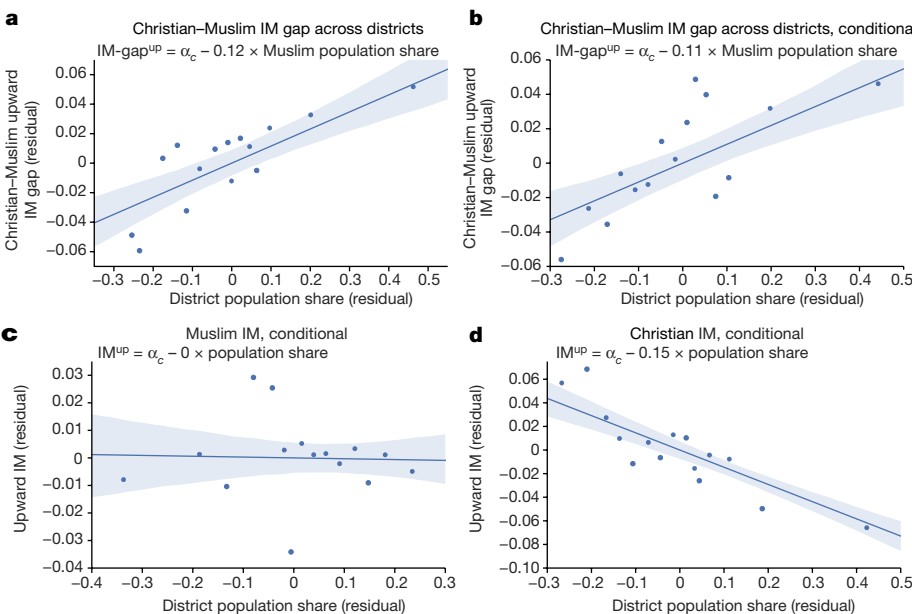

**Fig. 5 | District upward IM (gap) and Christian–Muslim population share.**
**a**,**b**, Binned scatterplots of the gap in upward IM between Christians and Muslims (**a**) against the share of Muslims in the district's population conditional on country fixed effects (**b**). Panel **b** also conditions on the religion-specific shares of the older generation with completed primary education in the district. **c**,**d**, Binned within-country scatterplots of upward IM for Christians (**c**) and Muslims (**d**) against their own religious share in the district's population, conditioning on the share of completed primary education of the older generation in the district for Christians and Muslims,

respectively. The figure also shows 95% confidence bands obtained by a simple bootstrap procedure. Specifically, for 10,000 bootstrap iterations, we resample the data with replacement, re-estimate the regression in the bootstrap sample and record the estimated coefficient values. We then use the 10,000 bootstrap estimates to predict the dependent variable along a regularly spaced grid from the minimum to the maximum of the independent variable. For each grid point, the lower end of the confidence band is the 2.5th percentile and the upper end is the 97.5th percentile of the 10,000 bootstrap predictions.

seem in greater numbers, even when we account for differences in the literacy of the older coreligionists in each district.

## Internal migration

Motivated by the evidence on regional childhood exposure effects for both Muslims and Christians (Fig. 3) and the significant correlation between segregation and religious IM gaps (Fig. 5), we zoom in on internal migration.

### Differential internal migration

We tabulate internal migration rates by religious affiliation. We classify as migrants those individuals who, at the time of the census, reside elsewhere than their birthplace district. Figure 6a plots the internal migrant shares for Christians, Muslims and Traditionalists, pooling all censuses across 20 countries (data unavailable for Nigeria). In Fig. 6a and 6b we indicate whether the probability of migration is significantly different between Christians and Muslims in each country. In 17 countries, Christians move at higher rates than Muslims. On average the propensity of Christians to migrate is 0.298 compared to 0.222 for Muslims and 0.194 for Traditionalists. In Cameroon, 40% of Christians reside somewhere other than their birthplace district, whereas the corresponding share for Muslims is 25%. In Ethiopia, the Christian–Muslim difference in emigration is 7 percentage points, whereas in Malawi it is 15 percentage points. The emigration rates of Muslims exceed those of Christians only in Rwanda (by 10.8 percentage points), Uganda (by 4.3 percentage points) and Mozambique (by 0.9 percentage points). Migration decisions reflect the associated costs and benefits of doing so that may differ across religious lines. Muslims probably face higher migration costs as they reside in relatively remote regions with limited investments. Thus, in Fig. 6b, we report internal migration shares, netting out the mean at the individual's birthplace (weighted by the

population of the region in the country) to account for interreligious differences in residence (Extended Data Fig. 3). Differences in internal migration are evident, even when we compare individuals born in the same district, with Christians being 3.6 percentage points more likely to move out than Muslims. Only in Uganda, South Africa, Mozambique and Rwanda, where Muslims are in the minority, are they on average 4 percentage points more likely to move from their birth region compared to Christians. Which other factors shape the uncovered differences, economic, cultural or institutional (interacted with religion), remains uncertain[45–47].

### Migration and religious segregation

Motivated by the low educational mobility of Muslims in predominantly Muslim districts, we explored the association between migration propensity and religious segregation. Figure 7 illustrates the patterns, plotting internal migration rates for Christians and for Muslims against their own-religion population shares. The (within-country) association is negative for both religious groups but starker for Muslims. The correlation between internal migration and own population share in the birthplace is three times larger for Muslims (coefficient (standard error) −0.33 (0.0248)) than for Christians (estimate (standard error) −0.12 (0.0234)). Muslims, compared to Christians, have a much lower propensity to move out of regions with sizable Muslim communities, exacerbating their initial educational disadvantage.

## Discussion

We construct religion-specific educational mobility measures since independence across African countries and regions and explore their origins. Three regularities emerge. First, there are significant differences in IM between Christians and Muslims, even comparing Africans living in the same district, with similarly (un)educated elders in their

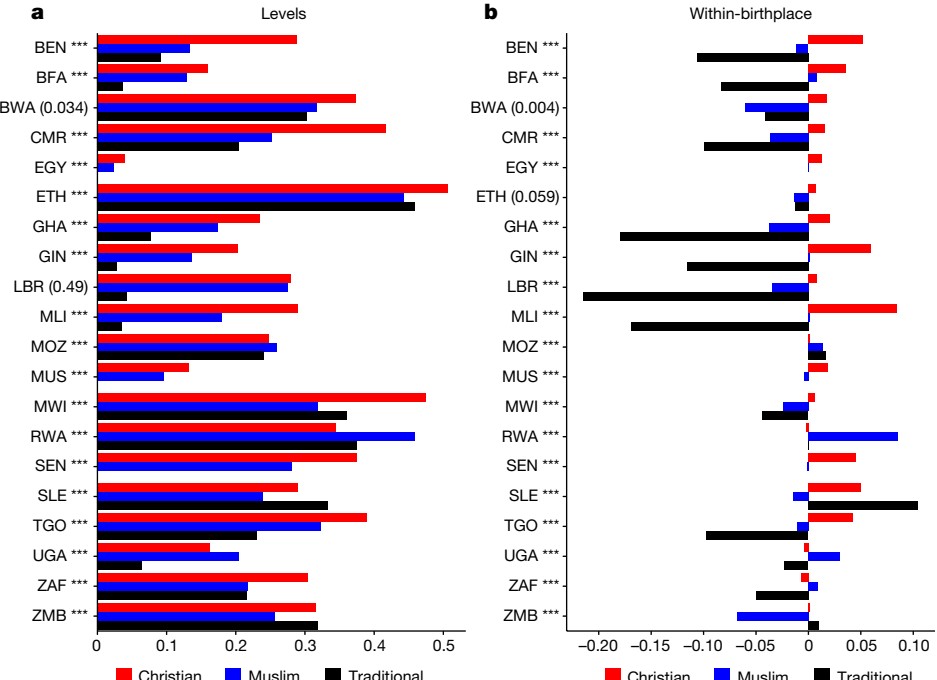

**Fig. 6 | Internal migration rates by religion. a**, Mean values for internal migration for Christians (red bars), Muslims (blue bars) and Africans adhering to Traditional religions (black bars) for each country, pooling all available censuses. A migrant is an individual residing at the time of the census in a district different from their birth region. **b**, Internal migration shares for Christians, Muslims and Traditionalists, netting out the mean value of the individual's birthplace. To get the demeaned-at-the-birth-district statistics, we proceed as follows. First, we get the total weights for individuals in a given country cohort census religion summing across all birthplace districts. Second, we calculate the number of district observations in a given country district cohort census major religion. Third, we divide the district (step 2) with the total (step 1) number of observations to get the district share. Fourth, we multiply the migrant share, individuals residing in other than the birthplace district, to get a weighted demeaned migrant share (step 3). Fifth, we sum the weighted migrant shares across all origin districts for each religious group in each country cohort. Sixth, we take the average across years, as there may be more than one census. The $P$ values of the difference in internal migration between Christians and Muslims appear in parentheses; ***$P < 0.001$. The $P$ values are computed as $2(1 - \Phi(|\frac{\mu^{mig}_{C-M}}{\sigma^{mig}_{C-M}}|))$, where $\Phi(\ )$ is the standard normal cumulative distribution and $\mu^{mig}_{C-M}$ and $\sigma^{mig}_{C-M}$ are the mean and standard deviation of the distribution of Christian − Muslim migrant share differences. The distribution of differences is computed from 1,000 bootstrap samples for each country, where each sample is a draw with replacement from the original data of the same size as the original data. BEN, Benin; BFA, Burkina Faso; BWA, Botswana; CMR, Cameroon; EGY, Egypt; ETH, Ethiopia; GHA, Ghana; GIN, Guinea; LBR, Liberia; MLI, Mali; MOZ, Mozambique; MUS, Mauritius; MWI, Malawi; RWA, Rwanda; SEN, Senegal; SLE, Sierra Leone; TGO, Togo; UGA, Uganda; ZAF, South Africa; ZMB, Zambia.

religious group, in households with comparable size and structure and with household heads in the same broad sector and occupation. Second, although Muslims benefit as much as Christians when they move early in life to high-mobility regions, they seem less likely to do so. The comparatively low internal mobility of Muslims accentuates their educational IM deficit, as they (and Traditionalists) reside in less urbanized areas, far from the capital and the coastline areas with limited infrastructure. Third, in areas with large Muslim communities, the Christian−Muslim IM gap is greatest; these highly segregated areas also have the lowest emigration rates among Muslims.

Our study begets more questions than it answers. First, as there is a great deal of variation within faiths, research should explore within-denomination variation distinguishing, for example, between Maliki and Shafi Suni, Ahmadis and Shia Muslims and between

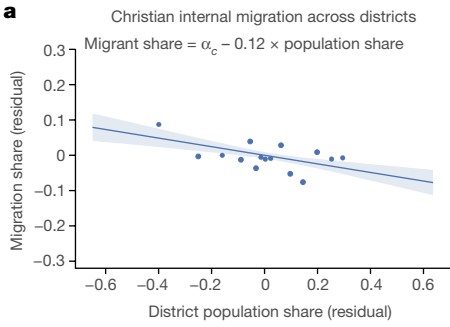

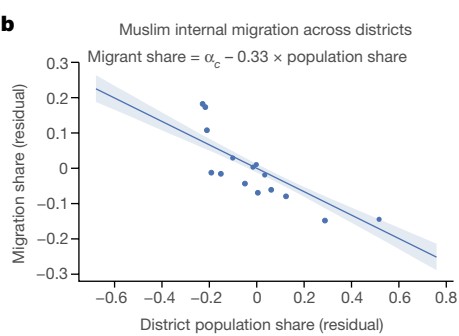

**Fig. 7 | Internal migration and own-religion population share. a,b**, Binned scatterplots of internal migration rates for Christians (**a**) and Muslims (**b**) against their own religious share in the district's population, conditional on country fixed effects. Also shown are the 95% confidence bands obtained by a simple bootstrap procedure. Specifically, for 10,000 bootstrap iterations, we resample the data with replacement, re-estimate the regression in the bootstrap sample and record the estimated coefficient values. We then use the 10,000 bootstrap estimates to predict the dependent variable along a regularly spaced grid from the minimum to the maximum of the independent variable. For each grid point, the lower end of the confidence band is the 2.5th percentile and the upper end is the 97.5th percentile of the 10,000 bootstrap predictions.

Protestants, Copts and Catholics. Doing so would allow delving into the probable causes, teasing apart the role of (1) social norms, (2) faith-specific schooling infrastructure, including *maktabs* and *madrasas* for Muslims and (3) religious leaders and their interaction with state institutions[24,44,45]. Second, our measures of intergenerational mobility in education do not capture how much learning takes place in schools and recent studies stress the low quality of schooling in the continent[48,49]. This may partially rationalize why the first-order differences in education between African Christians and Muslims do not translate into a stark interfaith gap in well-being and occupational specialization. It also highlights the need to estimate faith-specific private and social returns to schooling, both actual and perceived[50], in religiously segregated labour markets with denomination-specific risk-sharing institutions. The voluminous literature we review in Supplementary Information, section A1, documenting higher returns to primary education in low-income settings and Africa in particular[5,6], has paid little attention to the role of religion. Third, as millions are moving to Africa's new megacities, research should explore, and policy-makers should rethink, potential heterogeneity along religious lines of the economic return to migration, linking it with migration costs and labour markets both at the origin and the destination. Finally, as international institutions and African governments invest heavily in education with school construction programmes, abolishing fees and expanding access and georeferenced data on schools, educational quality and learning become available[40,51], future work on educational reforms needs to carefully explore the roots of inequalities in the take-up of educational policies by religious groups[25].

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

## Methods

### Data and educational IM measures

**Data reporting.** No experiments were performed. No statistical methods were used to predetermine the sample size of the harmonized census data provided by IPUMS. Supplementary Information, section B1 details our procedure for going from the IPUMS data to the sample used in the empirical analysis.

**Data.** IPUMS International records religious affiliation in 45 censuses from 20 African countries plus Nigeria, for which data come from household surveys between 2006 and 2010. The spatial disaggregation allows us to analyse a maximum of 2,304 districts, typically admin-2 or admin-3 regions. The number of districts used to construct faith-specific IM measures varies by cohort and whether one imposes restrictions in terms of sampling. (See also Supplementary Tables C4 and C5 and the discussion in Supplementary Information, section C3) Extended Data Table 1 reports religious shares by country. Egypt, Senegal, Mali and Guinea are predominantly Muslim (shares exceeding 85%), as well as Sierra Leone (77%) and Burkina Faso (59%). The share of those following Traditional religions is substantial in West Africa, Togo (29%), Benin (22%) and Burkina Faso (19%). When we weigh by the population of a country to account for the limited observations from Nigeria, Christians are about 50%, Muslims 42.7% and Traditionalists about 3% (Extended Data Fig. 1). We discuss conversion dynamics and their implications for IM in educational attainment estimates in Supplementary Information, section B3.

**Sample representativeness.** In 2020, the 21 countries hosted roughly 750 million of Africa's 1.35 billion people. North Africa is under-represented, as we have data only from Egypt. The sample includes both relatively rich, educated, with strongly institutionalized countries (for example, South Africa and Botswana with gross domestic product (GDP) per capita of about US $4,000–4,500 in 1995) and relatively poor, weakly institutionalized countries (for example, Ethiopia, Malawi and Mozambique with GDP per capita of about US $250 in 1995). The sample includes former British (Nigeria, Sierra Leone and Malawi), French (Burkina Faso, Senegal and Guinea), German-Belgian (Rwanda) and Portuguese (for example, Mozambique) colonies and protectorates, besides Liberia and Ethiopia. The sample also includes low-state capacity countries with lasting civil wars (Sierra Leone, Mozambique, Rwanda, Liberia and Ethiopia) and more stable ones (South Africa and Botswana). Supplementary Information, section B5 shows that the 21 countries are representative of the continent.

**Intergenerational mobility.** Our measures of educational IM reflect how 14–18-year-old children fare vis-a-vis cohabiting older generation members, typically biological parents, using primary school completion as the critical educational milestone (Supplementary Fig. A2).

Absolute upward IM in education: IM_up$_{igbcrt}$ = 1 in the case of child $i$, of religious affiliation $g$, born in decade $b$ in country $c$ (and residing in region $r$), observed in census-year $t$, born to parents who have not completed primary schooling completes primary education.

Absolute downward IM in education: IM_down$_{igbcrt}$ = 1 in the case of child $i$ whose parents have completed primary schooling does not complete primary education.

The IPUMS codebooks suggest that attendance at Christian and Islamic schools is accounted for as long as they do not solely cover religious topics.

**Educational dynamics and IM.** The literature on IM uses various statistics[52], like (one minus) the intergenerational coefficient obtained from a regression of children on parental schooling[53], rank–rank coefficients and rank movements[29]. Other studies[54–57] focus, as we do, on absolute transitions. Absolute mobility reflects both overall increases in education over time and movements in the distribution; hence relative and absolute IM measures are not necessarily correlated. See refs. 58,59 for a discussion on the link between absolute IM, relative IM, inequality and growth. We compile new statistics and study absolute upward and downward IM, using primary school completion as the educational cutoff. Focusing on the differences in upward and downward IM across denominations sheds light on the steady-state (ss) differences in educational achievement across religious groups. We can express the evolution of the share of completed primary education for birth cohort $b + 1$, of a religious group $g$, $\phi_{g,b+1}$, as a function of the share of those with completed primary education in the previous birth cohort, $\phi_{g,b}$, and the rates of upward IM, $u_{g,b}$, and downward IM, $d_{g,b}$.

$$\phi_{g,b+1} = \phi_{g,b}(1 - d_{g,b}) + (1 - \phi_{g,b})u_{g,b} \Leftrightarrow \Delta\phi_{g,b+1}$$
$$= u_{g,b} - \phi_{g,b}(u_{g,b} + d_{g,b}) \Leftrightarrow \phi_{g,ss} = \frac{u_{g,b}}{u_{g,b} + d_{g,b}}.$$

**Cohabitation.** We need to observe children's and parental education to build IM statistics. To maximize coverage, we use the average attainment of individuals one generation older than the child in the household. (The results are similar when we take the minimum or maximum or only the father's or mother's education). Matching young individuals to cohabiting older generation members raises concerns, as the transmission of education may differ for children living with and without older family member(s) and across religions. We focus on individuals aged 14–18 years, as primary education is mostly complete by then and cohabitation rates are high[57]. Supplementary Information, section B2 gives details. Cohabitation rates with older generation relatives, mainly biological parents and sometimes uncles and aunts, hover between 82% and 91%, without much difference between Christians, Muslims and Traditionalists.

**Religious affiliation across generations.** We explore the transmission of religious affiliation from parents to 14–18 year-old children. On the one hand, there is high intergenerational inertia for both Muslims and Christians. The likelihood that children of Muslim or Christian parents will report a different creed is, on average, less than 3%; in most countries, it is less than 1%. These estimates are close to the ones reported by ref. 60 across 19 African countries. On the other hand, it is common for African Muslims and Christians to follow traditional religious rituals and ceremonies[15]. However, the Census does not record 'mixed/dual' religious affiliation. To the extent that educated Africans adhering to Traditional religions alongside Christianity or Islam will respond that they are Christians or Muslims, the upward IM estimates for Traditionalists will be underestimated. Supplementary Information, section B3 gives details and graphical illustrations of the conversion dynamics across denominations and discusses their implications for our patterns.

**Ethnicity and religion.** Given the voluminous research on ethnicity in Africa[16–18], we examine the interplay between religion and ethnicity by tabulating censuses in which both are recorded. Religion transcends ethnicity (Supplementary Information, section B4). Although a few ethnicities are monoreligious (for example, Wolof and Fula in Senegal and the Somali in Ethiopia are Muslim and the Agew in Ethiopia and the Acholi in Uganda are Christian), most ethnicities, large and small, are multireligious. For example, the Oromo in Ethiopia, the Yoruba in Nigeria and the Sena in Mozambique are split between Christianity and Islam, whereas many groups in West Africa are split between Christianity and Traditional religions. There are dozens of ethnicities split between Traditional religions, Islam and Christianity, like the Gurma, the Basari and the Goulmancema in West Africa. Supplementary Figure D3 reports the religious IM gaps leveraging both cross-ethnicity and within-ethnicity variation (across individuals for whom IPUMS records ethnicity). (See also Supplementary Information, section D2.2).

## Explaining religious educational IM gaps

**Methodology.** To arrive at the Christian–Muslim and the Christian–Animist gaps in upward IM and downward IM in Fig. 2, we estimate the following regression with OLS:

$$\text{IM}_{ibchdt}^{\text{rel}} = \alpha_{cb} + \gamma_m \text{Muslim} + \gamma_a \text{Animist} + \delta_h \mathbf{H}_h' + \theta_h \mathbf{I}_h' + \psi \mathbf{E}_{e,r,b-1}' + \phi_{d,u/r} + \epsilon_{ibchdt}.$$

The dependent variable denotes upward or downward IM for child $i$ of religious affiliation rel, born in decade $b$, in household $h$, residing in district $d$ in country $c$, recorded in census $t$. 'Muslim' is an indicator for adherents to Islam; 'Animist' identifies children of Traditionalists. $\mathbf{H}_h'$ is a vector of household features, including size, composition and family organization. $\mathbf{I}_h'$ reflects occupation and industry indicators for the older generation in the household. $\mathbf{E}_{e,r,b-1}'$ denotes the share of the older generation with completed primary education in the district for each religious group. $\phi_{d,u/r}$ is a vector of district-specific constants interacted with an urban indicator. Parameters $\gamma_m$ and $\gamma_a$ reflect the educational gap of Muslims and Traditionalists vis-a-vis Christians, the omitted category.

## Childhood regional exposure effects

**Methodology.** To isolate regional childhood exposure effects from spatial sorting, we use an approach[37] that exploits differences in the timing of children's moves across districts with different levels of upward IM, adjusting it to derive religion-specific exposure[38]. The regional childhood exposure effects, reported in Fig. 3, are estimated from the following OLS specification:

$$\text{IM\_up}_{ihbmcod}^{\text{rel}} = \alpha_{ob} + \alpha_m + \sum_{m=1}^{18} \beta_m^{\text{rel}} \times I(m_i = m) \times \Delta_{odb}^{\text{nm}}$$
$$+ \sum_{b=b_0}^{B} k_b \times I(b_i = b) \times \Delta_{odb}^{\text{nm}} + \epsilon_{ihbmcod}$$

The specification relates primary education completion for child $i$, from household $h$, of birth cohort $b$, whose parents have not completed primary school, who moved from birthplace district $o$ in country $c$ to destination district $d$ at age $m$ in the same country, to differences in upward IM between origin and destination, among non-movers of the same cohort $b$ ($\Delta_{odb} = \widehat{\text{IM}}\_\text{up}_{bd}^{\text{nm}} - \widehat{\text{IM}}\_\text{up}_b^{\text{nm}}$). $\widehat{\text{IM}}\_\text{up}_b^{\text{nm}}$ summarizes the economic, social and institutional environment which shapes educational mobility in a district. We construct an overall measure of origin–destination differences in upward IM ($\Delta_{odb}^{\text{all}}$) and a religion-specific one ($\Delta_{odb}^{\text{rel}}$). Origin-region × birth-decade fixed effects, $\alpha_{ob}$, account for unobserved factors of the child's cohort and birthplace. The specification also includes interactions of destination–origin cohort IM differences with cohort effects to account for potential differential measurement error across cohorts (this has no effect). The parameters of interest, $\beta_m^{\text{rel}}$, capture how children's attainment varies with the age of their move to districts with higher or lower upward IM, conditional on age-of-move constants, $\alpha_m$, which absorb disruption and other age-specific features affecting education. If regions matter for mobility, the earlier the move, the greater the impact. As we include origin cohort specific constants, we leverage variation among children born in the same district and decade, moving to regions with different educational mobility. Differences in the age-of-move slopes, $\gamma_m = \beta_m^{\text{rel}} - \beta_{m+1}^{\text{rel}}$, reflect the impact of an extra year in the high-mobility district, regional childhood exposure effects.

**Sample.** For the implementation of the movers' design in the section on Sorting and childhood regional exposure (Fig. 3) that teases apart childhood exposure regional effects from spatial sorting, we need data not only on the district of birth and residence but also on the length of stay in the current location. IPUMS provides such information for 13 countries (Supplementary Table B1). The 13 countries come from all African regions. (1) Southern Africa: we have data from three (out of five) countries in the southern part of the continent, South Africa, Zambia and Malawi, missing Mozambique and Botswana. (2) Western Africa: the childhood exposure effects are estimated using data from six (out of ten) West African countries, Benin, Ghana, Guinea, Togo, Mali and Cameroon. (3) East Africa: we have all countries from Eastern Africa (Uganda, Rwanda and Ethiopia) except Mauritius. (4) North Africa: the mover's design includes Egypt, the only North African country. The 13-country sample includes French and British colonies and covers relatively poor (with meagre education) nations and more advanced economies.

## Regional correlates of interreligious educational mobility differences

**Religious differences in residence.** We explore differences in residence attributes among adherents of the three main religions, running country–birth cohort fixed-effects regressions associating geographic/location, at-independence development and historical features to indicator variables for Muslims and Traditionalists, respectively, with Christians serving as the omitted category. The specification reads:

$$Y_{i,c,b,r} = \alpha_{c,b} + \psi_1 \text{Muslim}_{i,c,b,r} + \psi_2 \text{Traditional}_{i,c,b,r} + \zeta_{i,r,c,b}.$$

Extended Data Figure 3 reports population-weighted least-squares estimates that reflect average differences in the respective outcomes, $Y_{i,c,b,r}$, between Christians and Muslims and Christian and Traditionalists partialling out country cohort constants, $\alpha_{c,b}$. Our exploration relates to research on the spread of Christianity and Islam in Africa. The following patterns emerge when we compare residence attributes between Muslims and Christians: (1) Muslims reside in less developed and densely populated regions, are more reliant on agriculture and are somewhat less urbanized; (2) Muslims reside in regions away from the capitals and the coastline, in line with works showing that missionaries mostly settled along the coast; and (3) Muslims reside in districts further away from colonial roads and railroads and far from Protestant and Catholic missions. Turning to Animists, the tabulations show the following: (1) similar to Muslims, Africans adhering to Traditional religions reside in less densely, more rural, agriculture-oriented regions; (2) adherents of Traditional religions reside in districts even further away from the capitals than Muslims; (3) Animists are more likely to settle in malaria-prevalent districts; and (4) Traditionalists are found in districts far from the colonial infrastructure.

**Specification of regional correlates of religious IM gap.** The specification on the correlates of regional differences in upward IM between Christians and Muslims and Christians and Traditionalists reads:

$$\widehat{\text{IM}}_{r,c,b}^{\text{C}} - \widehat{\text{IM}}_{r,c,b}^{\text{M,T}} = \gamma_c + \phi_1 D_{r,c} + \phi_2 G_{r,c} + \phi_3 H_{r,c} + \phi_4 R_{r,c} + \lambda E_{r,c}^{\text{old}} + \epsilon_{r,c,b}.$$

The dependent variable is the difference in upward IM between Christians and Muslims and between Christians and Traditionalists born in decade (birth cohort) $b$, in region $r$, in country $c$. (Hats denote regional averages across birth cohort regions). The explanatory variables are regional proxies of early (at independence) development ($D$), geography-location ($G$), historical aspects ($H$) and religious composition ($R$), which we include one by one in the empirical model as our objective is to characterize regional religious IM differences (rather than identify causal effects). As the specifications include country constants ($\gamma_c$), the coefficients capture the within-country correlation.

**Specification of regional correlates of IM, by religion.** We also estimate the regional specifications separately for Muslims, Christians and adherents of Traditional religions and compared the coefficient estimates of the location, early development and historical and compositional statistics. Extended Data Figure 4 reports the regional correlates of mobility separately for adherents of each religious group. Extended Data Figure 5 reports otherwise similar specifications, also controlling for the share of the older generation with completed primary education in the district that is the strongest correlate of educational mobility.

## Reporting summary

Further information on research design is available in the Nature Portfolio Reporting Summary linked to this article.

## Data availability

All newly built statistics on faith-specific absolute upward and downward educational IM by country, region and sex are based on census data compiled, processed and harmonized by IPUMS. IPUMS International microdata are publicly available free of charge. To access them, the prospective user may submit an electronic authorization form providing name, electronic address and institutional affiliation here: https://international.ipums.org/international/. Because our analyses are based on secondary, de-identified, publicly available data, we do not have an IRB waiver.

## Code availability

The code used to construct and analyse the data was written in R v.4.2.2 and Python v.3.11. The replication code and data files are available on https://github.com/imreligionafrica/imreligionafrica

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

**Acknowledgements** The paper is dedicated to A.A., who passed away while we were revising the manuscript. We will never forget his boundless curiosity, drive and charm. We terribly miss his wit, smile and can-do attitude. We thank M. Chatzigakis for his superb assistance with the data and codes and R. Chetty, E. Glaeser, N. Hendren and M. Poschke for their suggestions and comments. We have received helpful feedback and comments from seminar participants at the RIDGE Conference, the NBER Summer Institute, the AEA Conference (2021), the CEPR Conference on The Economics of Religion, King's College, Harvard, Zurich, the New Economic School, Trinity College Dublin, the World Bank, Hebrew University and Pantheon-Sorbonne. We also thank R. Jedwab, A. Storeygard, J. Cage, V. Rueda and N. Nunn for sharing their data. E.P. acknowledges support from the European Research Council (ERC consolidator grant ORDINARY) and the Wheeler Institute for Business and Development at the London Business School. All errors and omissions are our responsibility.

**Author contributions** A.A., S.H., S.M. and E.P. were the joint principal investigators for this project. A.A. designed the study, supervised the analysis and wrote the first draft of the paper. S.H. designed the study, supervised all parts of the analysis, analysed the data and wrote and edited the paper. S.M. designed the study, supervised all parts of the analysis, analysed the data and wrote and edited the paper. E.P. designed the study, supervised all parts of the analysis, analysed the data, wrote and edited the paper. The study has no local authors as the analysis is based on secondary data, harmonized and made publicly available by IPUMS International.

**Competing interests** The authors declare no competing interests.

**Additional information**
**Correspondence and requests for materials** should be addressed to Stelios Michalopoulos or Elias Papaioannou.

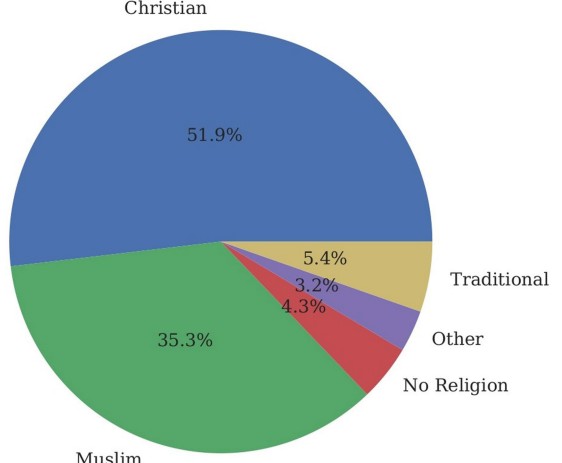

**(a)** Unweighted, All Individuals

Christian

51.9%

5.4%

3.2%

4.3%

Traditional

Other

35.3%

No Religion

Muslim

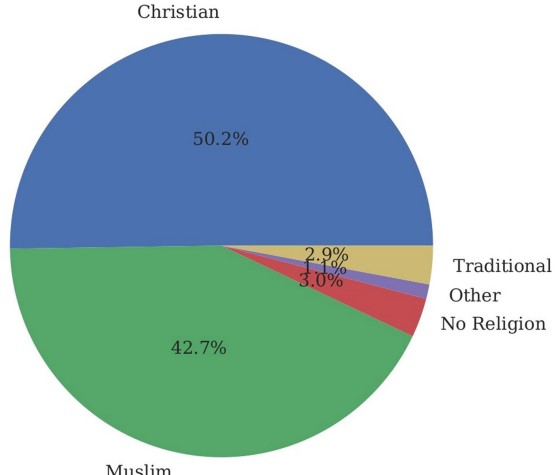

**(b)** 1980-population weighted, all individuals

Christian

50.2%

2.9%
1.1%
3.0%

Traditional

Other

42.7%

No Religion

Muslim

**Extended Data Fig. 1 | Major Religions. 21 African Countries.** The figure plots the population share of the main religions in our sample of 21 African countries. Panel (a) reports the shares across 82,037,564 individuals of all ages. Panel (b) reports weighted shares using the countries' populations in 1980. The Christian share combines various denominations, like Orthodox, Catholic, and Protestant, available in some censuses. Likewise, the Muslim share combines various branches, like Sunni and Shia. Traditional also combines various indigenous religions, like Vodun, Animist, and Traditional religions. Supplementary Fig. A2 gives country-specific details.

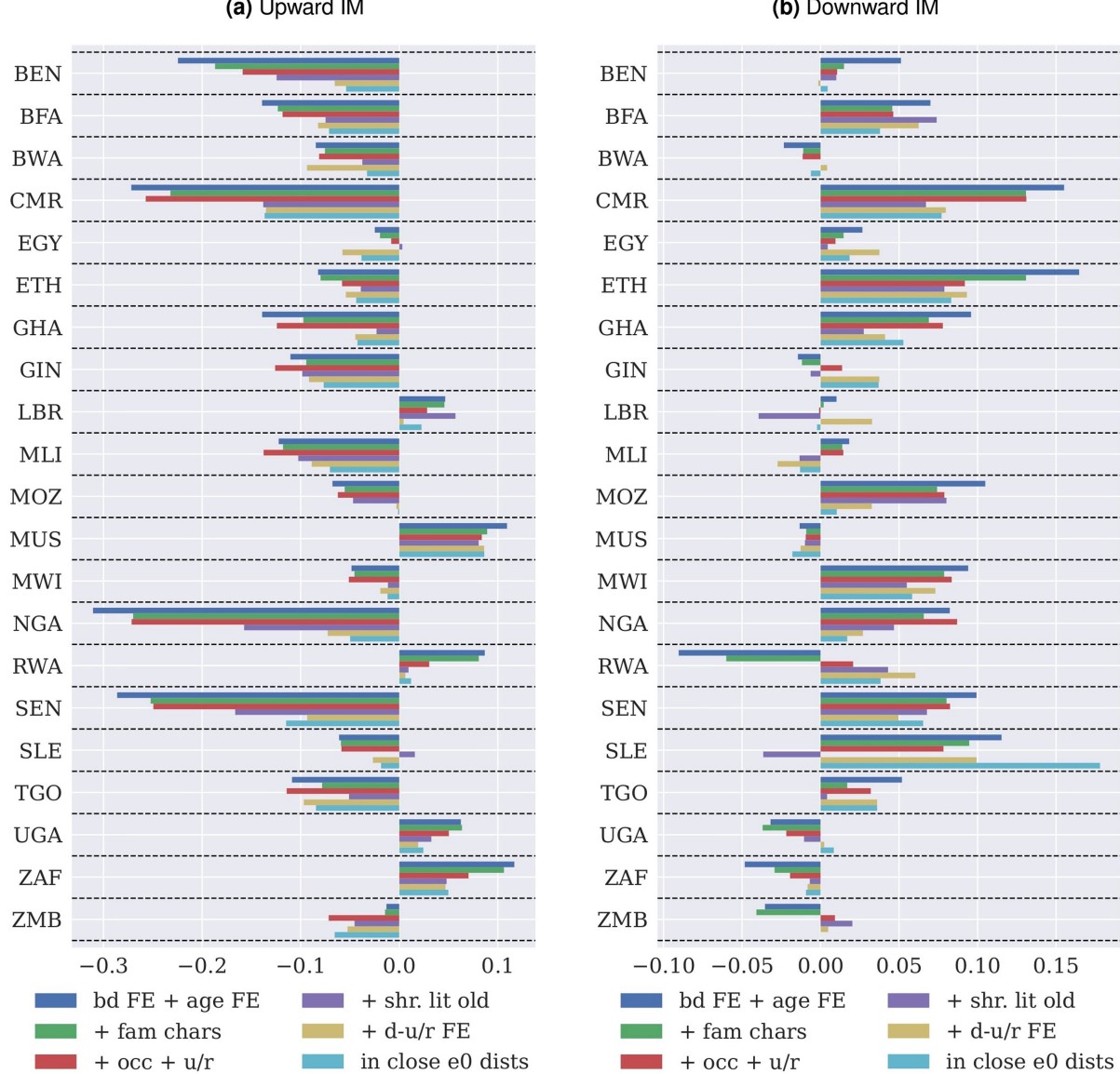

**(a)** Upward IM

**(b)** Downward IM

**Extended Data Fig. 2 | Christian–Muslim IM Gap Drivers, by Country.** The figure portrays how the Christian–Muslim gap in educational upward IM (panel (a)) and downward IM (panel (b)) changes as we add controls for the household structure (model (2), in green), parental occupational specialization, the industry of employment, and rural–urban residence (model (3), in red), the share completed primary education of the older generation of the same religious group in the district (model (4), in purple), and district x rural/urban fixed effects (model (5), in dark yellow) for each country. The last permutation (model (6), in light blue) restricts estimation in half of each country's districts, where differences in completed primary education of the older generation between Christians and Muslims are the smallest. The bars on the top (model (1), in dark blue) reflect the baseline inter-religious differences in IM, conditioning on birth cohort fixed effects and age constants. The figure gives linear probability model (OLS) estimates.

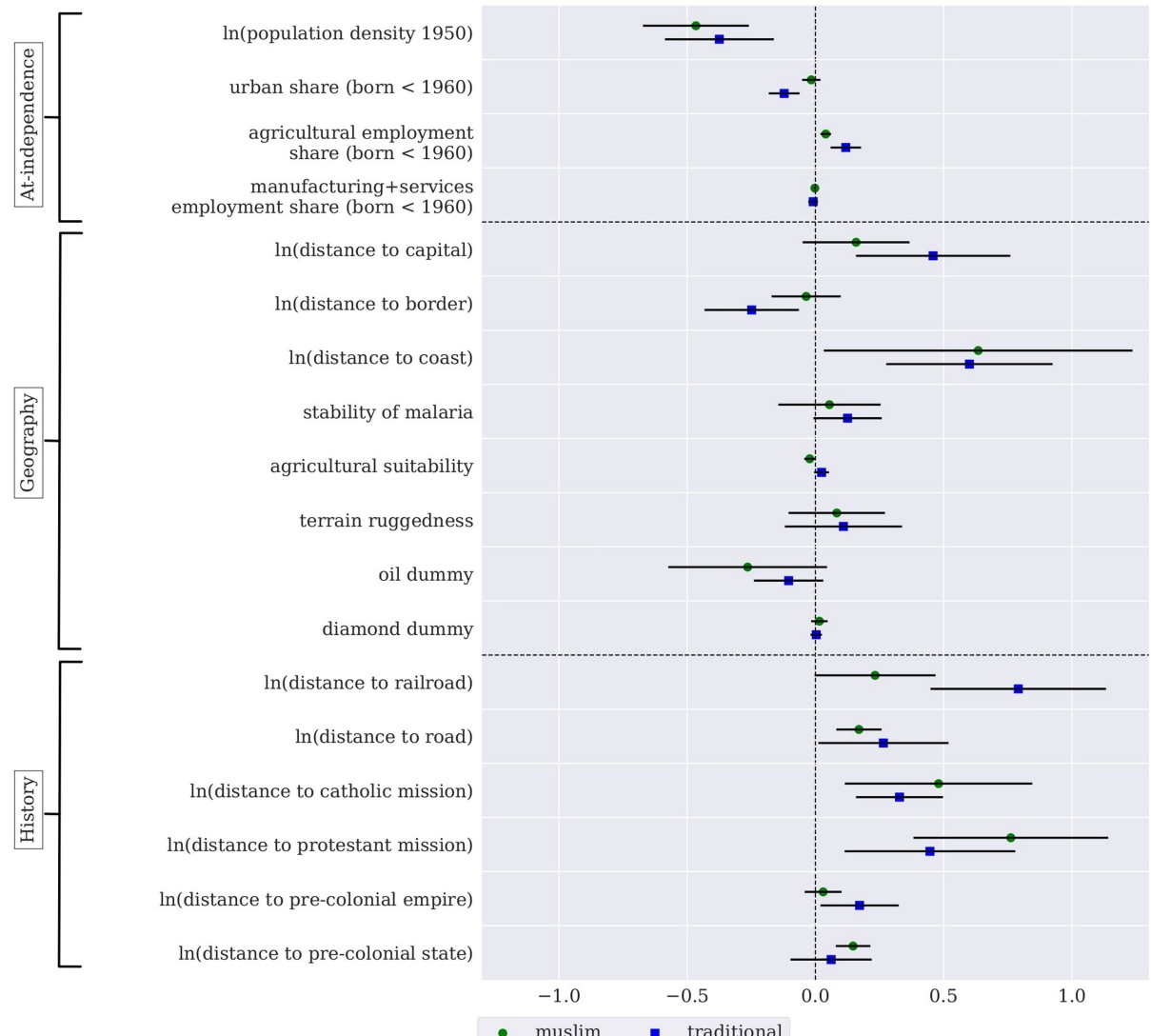

**Extended Data Fig. 3 | District Correlates of Residence by Religious Affiliation.** The figure plots OLS regression coefficients associating the variable listed on the left of the graph to indicator variables for Muslims and Traditionalists, with Christians serving as the omitted category, conditioning on country–birth cohort fixed effects. The estimates have, therefore, a within-country–birth-decade test of means interpretation. There are three categories of independent variables. (i) Regional proxies of development before independence. (ii) Regional geographic and location features. (iii) Historical variables of colonial investments and precolonial statehood. Supplementary Section F gives variable definitions and sources.

Two-standard-error bands based on heteroskedasticity adjusted clustered at the country level are also reported. The point estimates (green and blue dots in the figure) were obtained by running separate regressions of the district-level Christian–Muslim and Christian - Traditional IM gap respectively on each district-level variable (indicated on the vertical axis of the figure). The IM gap is defined as the average IM of Christians minus the average IM of Muslims or Traditionalists in the district. Before running each regression, we standardize the dependent and independent variable by subtracting its sample mean and dividing by its sample standard deviation.

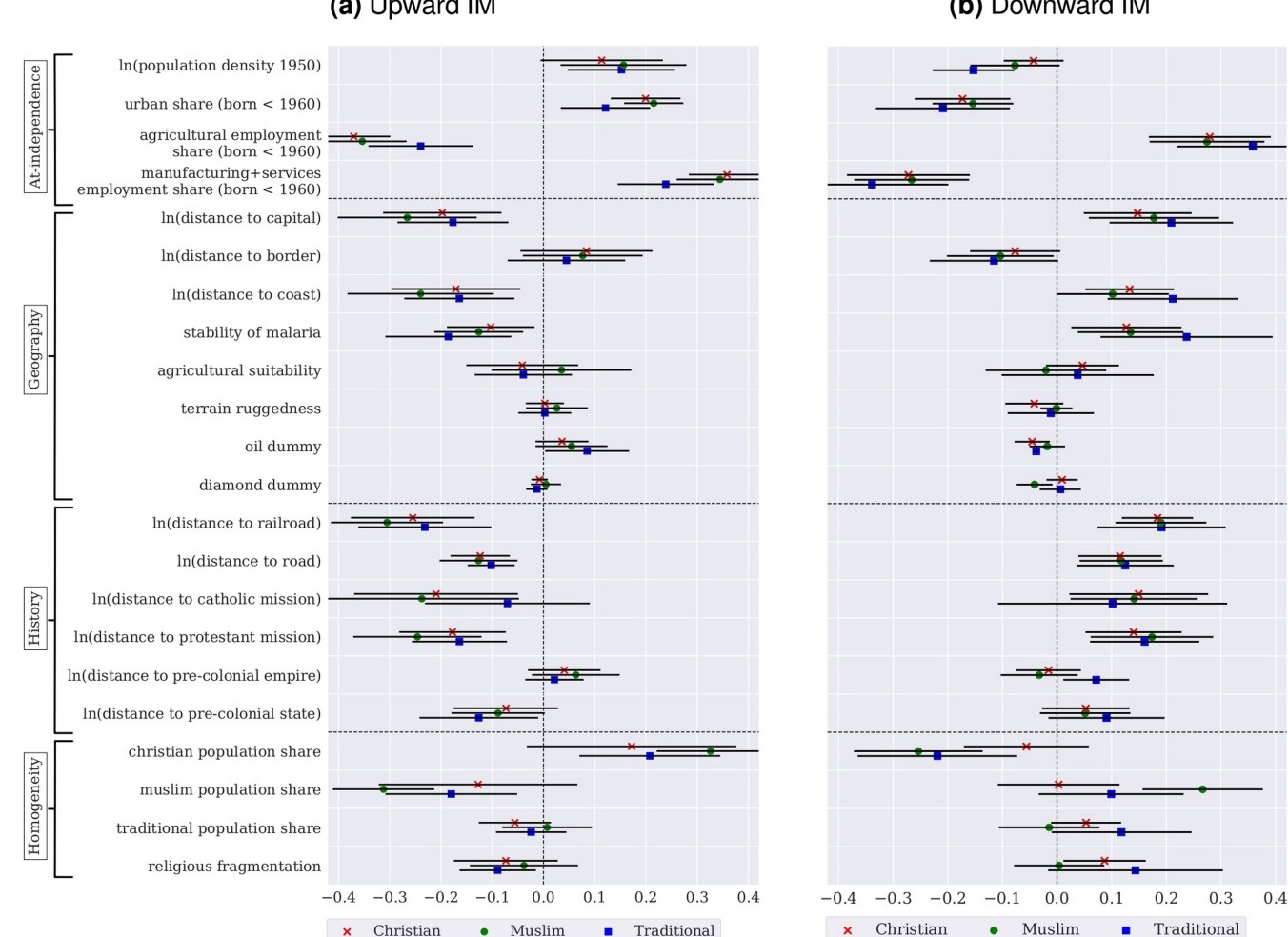

**(a)** Upward IM

**(b)** Downward IM

**Extended Data Fig. 4 | Regional IM Correlates by Religion, Country Fixed Effects.** The figure plots correlations (standardized "beta" coefficients) between intergenerational mobility (IM) and various regional characteristics for Christians (red star), Muslims (green rhombus), and Traditionalists (blue square). Panel (a) examines upward IM that reflects the likelihood that young individuals, aged 14–18, residing in households where the older generation has not completed primary schooling, will complete primary education. Panel (b) examines downward IM that reflects the likelihood that young individuals, aged 14–18, residing in households where the older generation has completed primary schooling will fail to do so. There are four categories of IM correlates. (i) Proxies of development before independence. (ii) Location and geographic features. (iii) Historical variables, including colonial-era investments and precolonial statehood. (iv) Homogeneity, captured by the shares of each of the three religious groups. Supplementary Section F gives variable definitions and sources. All specifications include country fixed effects (constants not reported). Standard errors are clustered at the country level. The point estimates (green and blue dots in the figure) were obtained by running separate regressions of the district-level Christian–Muslim and Christian-Traditional IM gap respectively on each district-level variable (indicated on the vertical axis of the figure). The IM gap is defined as the average IM of Christians minus the average IM of Muslims or Traditionalists in the district. Before running each regression, we standardize the dependent and independent variable by subtracting its sample mean and dividing by its sample standard deviation.

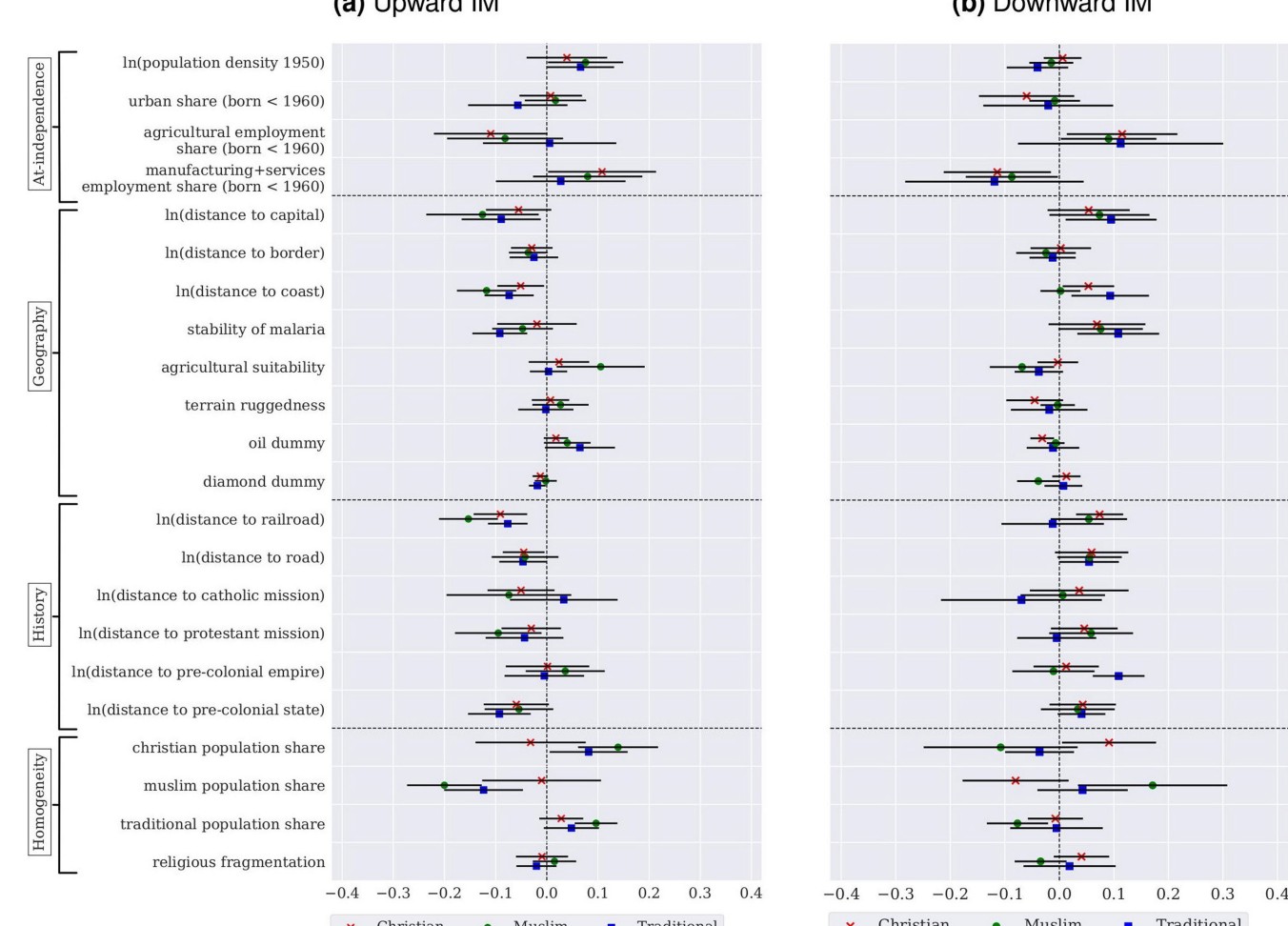

**(a) Upward IM**

**(b) Downward IM**

× Christian  ● Muslim  ■ Traditional

**Extended Data Fig. 5 | Regional IM Correlates, Country FE, Cond. on Own Religion Old's Completed Primary.** The figure plots correlations (standardized "beta" coefficients) between intergenerational mobility (IM) and various regional characteristics for Christians (red star), Muslims (green rhombus), and Traditionalists (blue square), conditioning on their own religious group older generation's completed primary education in the district. Panel (a) examines upward IM that reflects the likelihood that young individuals, aged 14–18, residing in households where the older generation has not completed primary schooling, will complete primary education. Panel (b) examines downward IM that reflects the likelihood that young individuals, aged 14–18, residing in households where the older generation has completed primary schooling will fail to do so. There are four categories of IM correlates. (i) Proxies of development before independence. (ii) Location and geographic

features. (iii) Historical variables, including colonial-era investments and precolonial statehood. (iv) Homogeneity, captured by the shares of each of the three religious groups. Appendix Section F gives variable definitions and sources. All specifications include country fixed effects (constants not reported). Standard errors are clustered at the country level. The point estimates (green and blue dots in the figure) were obtained by running separate regressions of the district-level Christian–Muslim and Christian–Traditionalist IM gap, respectively, on each district-level variable (indicated on the vertical axis of the figure). The IM gap is defined as the average IM of Christians minus the average IM of Muslims or Traditionalists in the district. Before running each regression, we standardize the dependent and independent variable by subtracting its sample mean and dividing by its sample standard deviation.

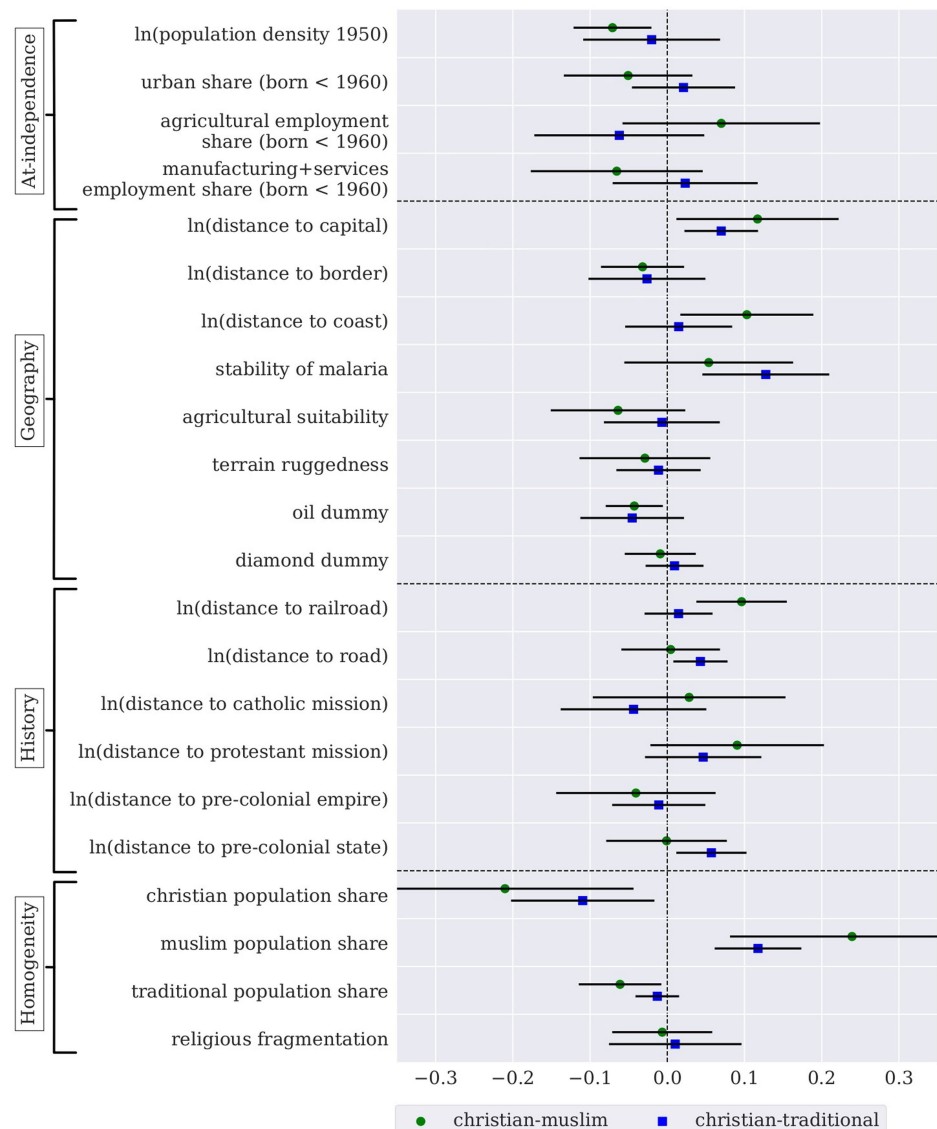

**Extended Data Fig. 6 | District Correlates of Christian–Muslim and Christian–Traditionalist IM Gaps, Conditional on Older Generation's Completed Primary.** The figure plots correlations (standardized "beta" coefficients) between upward IM gaps of Christians and Muslims (green circle) and Christians and Traditionalists (blue square), averaged across individuals in a district, and various regional characteristics. (i) Proxies of development before independence. (ii) Location and geographic features. (iii) Historical variables of colonial investments and precolonial statehood. (iv) Homogeneity, captured by the shares of each of the three religious groups. Supplementary Section F gives variable definitions and sources. All specifications include country fixed effects (constants not reported). Two-standard-error bands based on heteroskedasticity adjusted clustered at the country level are also reported. The point estimates (green and blue dots in the figure) were obtained by running separate regressions of the district-level Christian–Muslim and Christian–Traditionalist IM gap respectively on each district-level variable (indicated on the vertical axis of the figure). The IM gap is defined as the average IM of Christians minus the average IM of Muslims or Traditionalists in the district. Before running each regression, we standardize the dependent and independent variable by subtracting its sample mean and dividing by its sample standard deviation.

**Extended Data Table 1 | Major religion shares by country**

| country | Christian | Muslim | Traditional | Other | No Religion |
|---|---|---|---|---|---|
| Benin | 0.442 | 0.254 | 0.220 | 0.022 | 0.062 |
| Burkina Faso | 0.216 | 0.587 | 0.188 | 0.004 | 0.005 |
| Botswana | 0.760 | 0.006 | 0.049 | 0.008 | 0.177 |
| Cameroon | 0.692 | 0.209 | 0.056 | 0.010 | 0.033 |
| Egypt | 0.056 | 0.944 | 0.000 | 0.000 | 0.000 |
| Ethiopia | 0.640 | 0.311 | 0.039 | 0.010 | 0.000 |
| Ghana | 0.701 | 0.169 | 0.066 | 0.008 | 0.056 |
| Guinea | 0.062 | 0.880 | 0.023 | 0.002 | 0.033 |
| Liberia | 0.858 | 0.121 | 0.006 | 0.002 | 0.014 |
| Mali | 0.024 | 0.951 | 0.020 | 0.000 | 0.004 |
| Mozambique | 0.564 | 0.180 | 0.067 | 0.000 | 0.189 |
| Mauritius | 0.324 | 0.168 | 0.000 | 0.503 | 0.005 |
| Malawi | 0.814 | 0.129 | 0.024 | 0.000 | 0.033 |
| Nigeria | 0.525 | 0.466 | 0.009 | 0.000 | 0.000 |
| Rwanda | 0.932 | 0.018 | 0.003 | 0.006 | 0.041 |
| Senegal | 0.041 | 0.956 | 0.000 | 0.004 | 0.000 |
| Sierra Leone | 0.211 | 0.767 | 0.001 | 0.009 | 0.013 |
| Togo | 0.479 | 0.157 | 0.290 | 0.009 | 0.065 |
| Uganda | 0.852 | 0.124 | 0.005 | 0.016 | 0.004 |
| South Africa | 0.781 | 0.014 | 0.017 | 0.058 | 0.131 |
| Zambia | 0.917 | 0.005 | 0.043 | 0.001 | 0.034 |

The table reports the share of Christians, Muslims, Animists (Traditionalists), alongside the two residual religion categories, Other and No Religion across all 21 sample countries. The statistics are calculated using all censuses. Appendix Section B.3 gives details on the aggregation and the sample.

**Extended Data Table 2 | Summary Statistics. District-level Christian–Muslim IM Gaps**

### Panel A: Upward Intergenerational Mobility (IM)

| country | (1) $N_{districts}$ | (2) median | (3) min | (4) max | (5) mean | (6) std | (7) share($IM_c^{up} > IM_m^{up}$) |
|---|---|---|---|---|---|---|---|
| Cameroon | 165 | 0.184 | -0.415 | 0.924 | 0.181 | 0.255 | 0.764 |
| Senegal | 26 | 0.173 | -0.145 | 0.511 | 0.188 | 0.162 | 0.923 |
| Burkina Faso | 46 | 0.125 | -0.032 | 0.454 | 0.14 | 0.091 | 0.935 |
| Togo | 37 | 0.09 | -0.318 | 0.471 | 0.087 | 0.142 | 0.757 |
| Botswana | 8 | 0.085 | -0.399 | 0.864 | 0.238 | 0.54 | 0.5 |
| Ghana | 110 | 0.083 | -0.228 | 0.428 | 0.069 | 0.122 | 0.736 |
| Guinea | 26 | 0.079 | -0.253 | 0.836 | 0.156 | 0.274 | 0.808 |
| Nigeria | 21 | 0.075 | -0.546 | 0.538 | 0.074 | 0.278 | 0.619 |
| Benin | 75 | 0.067 | -0.653 | 0.935 | 0.072 | 0.216 | 0.72 |
| Zambia | 27 | 0.061 | -0.623 | 0.622 | 0.064 | 0.338 | 0.556 |
| Sierra Leone | 97 | 0.058 | -0.2 | 0.909 | 0.122 | 0.197 | 0.794 |
| Mali | 164 | 0.055 | -0.409 | 0.961 | 0.12 | 0.307 | 0.555 |
| Malawi | 146 | 0.055 | -0.711 | 0.319 | 0.026 | 0.142 | 0.685 |
| Egypt | 222 | 0.053 | -0.775 | 0.409 | 0.053 | 0.145 | 0.766 |
| Ethiopia | 84 | 0.032 | -0.399 | 0.681 | 0.062 | 0.209 | 0.619 |
| Liberia | 39 | 0.007 | -0.469 | 0.348 | 0.009 | 0.162 | 0.564 |
| Mozambique | 203 | 0.0 | -0.961 | 1.0 | -0.036 | 0.344 | 0.483 |
| Uganda | 131 | -0.051 | -0.656 | 0.5 | -0.079 | 0.173 | 0.313 |
| Rwanda | 29 | -0.062 | -0.336 | 0.226 | -0.046 | 0.135 | 0.345 |
| Mauritius | 37 | -0.083 | -0.3 | 0.061 | -0.085 | 0.083 | 0.027 |
| South Africa | 80 | -0.143 | -0.514 | 0.832 | -0.022 | 0.34 | 0.338 |

### Panel B: Downward Intergenerational Mobility (IM)

| country | (1) $N_{districts}$ | (2) median | (3) min | (4) max | (5) mean | (6) std | (7) share($IM_c^{down} < IM_m^{down}$) |
|---|---|---|---|---|---|---|---|
| Malawi | 95 | -0.182 | -0.643 | 0.765 | -0.141 | 0.323 | 0.705 |
| Sierra Leone | 88 | -0.173 | -0.8 | 0.69 | -0.2 | 0.283 | 0.773 |
| Ethiopia | 63 | -0.165 | -0.833 | 0.526 | -0.16 | 0.255 | 0.73 |
| Senegal | 25 | -0.13 | -0.592 | 0.389 | -0.128 | 0.211 | 0.8 |
| Ghana | 110 | -0.102 | -0.859 | 0.237 | -0.1 | 0.163 | 0.755 |
| Burkina Faso | 43 | -0.095 | -0.6 | 0.286 | -0.121 | 0.188 | 0.837 |
| Mali | 72 | -0.092 | -1.0 | 1.0 | -0.061 | 0.435 | 0.597 |
| Liberia | 30 | -0.081 | -0.48 | 0.67 | -0.088 | 0.254 | 0.633 |
| Cameroon | 152 | -0.052 | -0.981 | 0.75 | -0.14 | 0.294 | 0.625 |
| Togo | 32 | -0.052 | -0.386 | 0.333 | -0.024 | 0.159 | 0.594 |
| Rwanda | 24 | -0.031 | -0.413 | 0.664 | -0.011 | 0.248 | 0.583 |
| Nigeria | 23 | -0.028 | -0.444 | 0.759 | -0.015 | 0.218 | 0.565 |
| Egypt | 222 | -0.016 | -0.164 | 0.445 | -0.009 | 0.067 | 0.667 |
| Mozambique | 67 | -0.005 | -1.0 | 1.0 | -0.158 | 0.442 | 0.507 |
| Guinea | 30 | -0.003 | -0.541 | 0.675 | 0.019 | 0.304 | 0.5 |
| Uganda | 124 | -0.001 | -0.481 | 0.489 | 0.014 | 0.169 | 0.5 |
| Benin | 58 | 0.008 | -0.9 | 0.454 | -0.027 | 0.269 | 0.483 |
| Mauritius | 37 | 0.01 | -0.077 | 0.298 | 0.021 | 0.06 | 0.216 |
| Botswana | 9 | 0.064 | -0.284 | 0.134 | 0.044 | 0.127 | 0.111 |
| South Africa | 99 | 0.086 | -0.932 | 0.228 | 0.056 | 0.15 | 0.162 |
| Zambia | 25 | 0.103 | -0.751 | 0.498 | 0.001 | 0.368 | 0.4 |

The table reports Christian–Muslim differences (gaps) in intergenerational mobility (IM) for the 1980s cohort (the cohort with the broadest coverage) for individuals aged 14–18 cohabitating with older generation relatives by country. Panel A reports estimates for upward IM and panel B for downward IM. Because of differences in the timing of censuses, reported in IPUMS, the statistics for Liberia, Mali, Nigeria, and Togo correspond to the 1990s cohort. Column (1) gives the number of districts with information for both Christians and Muslims, required to calculate the regional IM statistics. Columns (2) - (6) report summary statistics (median, min, max, average, and standard deviation) for the Christian–Muslim IM gap across districts in the country. Column (7) gives the share of districts for which Christians have higher upward mobility than Muslims (Panel A) or lower downward mobility than Muslims (Panel B). Figure 1, panel (a), portrays the spatial distribution of differences in upward IM between Christians and Muslims across regions.

# Reporting Summary

## Statistics

For all statistical analyses, confirm that the following items are present in the figure legend, table legend, main text, or Methods section.

| n/a | Confirmed | |
|---|---|---|
| ☐ | ☒ | The exact sample size (*n*) for each experimental group/condition, given as a discrete number and unit of measurement |
| ☒ | ☐ | A statement on whether measurements were taken from distinct samples or whether the same sample was measured repeatedly |
| ☐ | ☒ | The statistical test(s) used AND whether they are one- or two-sided<br>*Only common tests should be described solely by name; describe more complex techniques in the Methods section.* |
| ☐ | ☒ | A description of all covariates tested |
| ☐ | ☒ | A description of any assumptions or corrections, such as tests of normality and adjustment for multiple comparisons |
| ☐ | ☒ | A full description of the statistical parameters including central tendency (e.g. means) or other basic estimates (e.g. regression coefficient) AND variation (e.g. standard deviation) or associated estimates of uncertainty (e.g. confidence intervals) |
| ☐ | ☒ | For null hypothesis testing, the test statistic (e.g. *F*, *t*, *r*) with confidence intervals, effect sizes, degrees of freedom and *P* value noted<br>*Give P values as exact values whenever suitable.* |
| ☒ | ☐ | For Bayesian analysis, information on the choice of priors and Markov chain Monte Carlo settings |
| ☒ | ☐ | For hierarchical and complex designs, identification of the appropriate level for tests and full reporting of outcomes |
| ☒ | ☐ | Estimates of effect sizes (e.g. Cohen's *d*, Pearson's *r*), indicating how they were calculated |

*Our web collection on statistics for biologists contains articles on many of the points above.*

## Software and code

Policy information about availability of computer code

| Data collection | We use individual-level data from IPUMS International, which collects, cleans, and harmonizes Census data from various African countries. Researchers were not involved in data collection. |
|---|---|
| Data analysis | The code used to construct and analyze the data was written in R v.4.2.2 and Python v.3.11. |

For manuscripts utilizing custom algorithms or software that are central to the research but not yet described in published literature, software must be made available to editors and reviewers. We strongly encourage code deposition in a community repository (e.g. GitHub). See the Nature Portfolio guidelines for submitting code & software for further information.

## Data

Policy information about availability of data

All manuscripts must include a data availability statement. This statement should provide the following information, where applicable:

- Accession codes, unique identifiers, or web links for publicly available datasets
- A description of any restrictions on data availability
- For clinical datasets or third party data, please ensure that the statement adheres to our policy

The replication code and data files are available on https://github.com/imreligionafrica/imreligionafrica The individual level data from IPUMS International are available here: https://international.ipums.org/international/. here

## Human research participants

Policy information about studies involving human research participants and Sex and Gender in Research.

| | |
|---|---|
| Reporting on sex and gender | *The main analysis does not distinguish by sex. In the Supplementary Information, we report some results separately for girls and boys. The underlying information on sex comes from IPUMS International, which in turn collects and harmonizes Census data collected by national statistical agencies. Primary data collection, recruitment, and consent was done by the national statistical agencies conducting the Censuses.* |
| Population characteristics | *Young individuals, aged 14-18 years and 14-25 years, who cohabitate with at least one older generation member in the household, typically a biological parent.* |
| Recruitment | *The paper conducts secondary data analysis only. Primary data collection, recruitment, and consent was done by the national statistical agencies and this is described in depth at the respective technical reports, alongside the manuals in IPUMS International.* |
| Ethics Oversight | *Because our analyses are based on secondary, de-identified, publicly available data, we do not have an IRB waiver.* |

Note that full information on the approval of the study protocol must also be provided in the manuscript.

# Field-specific reporting

Please select the one below that is the best fit for your research. If you are not sure, read the appropriate sections before making your selection.

☐ Life sciences ☒ Behavioural & social sciences ☐ Ecological, evolutionary & environmental sciences

For a reference copy of the document with all sections, see nature.com/documents/nr-reporting-summary-flat.pdf

# Life sciences study design

All studies must disclose on these points even when the disclosure is negative.

Sample size

Data exclusions

Replication

Randomization

Blinding

# Behavioural & social sciences study design

All studies must disclose on these points even when the disclosure is negative.

| | |
|---|---|
| Study description | We compile measures of absolute  intergenerational mobility in educational attainment for Africans adhering to different religions across African countries and regions and analyze the features shaping differences across faith in educational mobility. |
| Research sample | Individual level data of intergenerational mobility in education based on matched  children-parents educational attainment data from 21 African countries, available via IPUMS International. Data is disaggregated by religion, country, regions, gender, rural-urban, and schooling level (completed primary or higher versus non-completed primary). For most countries, IPUMS gives a representative 10% sample. |
| Sampling strategy | We use all observations from IPUMS International that meet the sample criteria; namely, 14-18 year old and 14-25 year old individuals who cohabitate with at least one older generation member in the household with available information on education and religious affiliation. IPUMS International uses mainly 10% sample from 21 African countries. See Supplementary Information for details, please. |

| Data collection | Researchers were not involved in data collection. All data are retrieved from IPUMS - International, which in turn collects and harmonizes Census data and provides to the public representative, typically 10%, samples. To access the data, the prospective user may submit an electronic authorization form providing name, electronic address, and institutional affiliation here: https://international.ipums.org/international/. |
|---|---|
| Timing | We use all Censuses from African countries with information on religious affiliation since independence. Most Censuses were conducted in the 1990s and 2000s. The earliest Census is in 1970 and the latest in 2016. See please the Supplementary Information for details. |
| Data exclusions | We did not exclude any individuals who met the Research Sample criteria. |
| Non-participation | N/A |
| Randomization | The data is observational and there is no random variation. We discuss how our correlational results should be interpreted. |

# Ecological, evolutionary & environmental sciences study design

All studies must disclose on these points even when the disclosure is negative.

| Study description | |
|---|---|
| Research sample | |
| Sampling strategy | |
| Data collection | |
| Timing and spatial scale | |
| Data exclusions | |
| Reproducibility | |
| Randomization | |
| Blinding | |

Did the study involve field work?  ☐ Yes  ☑ No

## Field work, collection and transport

| Field conditions | |
|---|---|
| Location | |
| Access & import/export | |
| Disturbance | |

# Reporting for specific materials, systems and methods

We require information from authors about some types of materials, experimental systems and methods used in many studies. Here, indicate whether each material, system or method listed is relevant to your study. If you are not sure if a list item applies to your research, read the appropriate section before selecting a response.

## Materials & experimental systems

| n/a | Involved in the study |
|-----|------------------------|
| ☑ | ☐ Antibodies |
| ☑ | ☐ Eukaryotic cell lines |
| ☑ | ☐ Palaeontology and archaeology |
| ☑ | ☐ Animals and other organisms |
| ☑ | ☐ Clinical data |
| ☑ | ☐ Dual use research of concern |

## Methods

| n/a | Involved in the study |
|-----|------------------------|
| ☑ | ☐ ChIP-seq |
| ☑ | ☐ Flow cytometry |
| ☑ | ☐ MRI-based neuroimaging |

## Antibodies

Antibodies used

Validation

## Eukaryotic cell lines

Policy information about cell lines and Sex and Gender in Research

Cell line source(s)

Authentication

Mycoplasma contamination

Commonly misidentified lines
(See ICLAC register)

## Palaeontology and Archaeology

Specimen provenance

Specimen deposition

Dating methods

☐ Tick this box to confirm that the raw and calibrated dates are available in the paper or in Supplementary Information.

Ethics oversight

Note that full information on the approval of the study protocol must also be provided in the manuscript.

## Animals and other research organisms

Policy information about studies involving animals; ARRIVE guidelines recommended for reporting animal research, and Sex and Gender in Research

Laboratory animals       *For laboratory animals, report species, strain and age OR state that the study did not involve laboratory animals.*

