## [Peer Review File · Nature]

Manuscript Title: Religion and Educational Mobility in Africa

Reviewer Comments & Author Rebuttals

Reviewer Reports on the Initial Version:

Referees' comments:

Referee #1 (Remarks to the Author):

In this paper the author(s) document patterns in upward and downward educational intergenerational mobility across religious groups in Africa using census and survey data from 20 African countries. They find that a) there are large differences in IM across religious groups, with Christians typically having greater upward and lower downward mobility than other religious groups, of which Muslims are numerically the largest, and b) the Christian-Muslim gap is most strongly correlated with the regional share of the Muslim population and that household-level attributes and economic differences explain little of this gap. This paper presents an important set of descriptive information about patterns of formal education in a region with some of the lowest level of economic development and lowest levels of educational attainment globally. The data compilation and analysis is well-done. I do not have concerns with the construction of the main quantities of interest -- IM and gaps therein -- with one exception about the fact that religious affiliation is measured many years after the age of primary school for adult respondents, which I discuss further below.

My comments and suggestions primary regard the discussion of significance, clarity and context, and conclusions of the paper. I believe the paper would be improved by 1) owning up to the puzzle the authors present at the outset, 2) finding clearer ways to present some of the main descriptives, 3) organizing the prose in way that is not just a march through results, 4) being clearer about what is at stake and into what conversations the piece is fitting.

This paper has important patterns to document, and does not present an "answer" to why these patterns exist, rather ruling out a set of plausible explanations while raising more questions. I think this is fine, but the paper would be made stronger by owning this puzzle. The first paragraph of the conclusion ("We uncover substantial differences in educational mobility between Muslims, Christians, and Animists across Africa post-independence. Regional effects appear first order causally affecting upward mobility for children of all faiths. Religious segregation is the most important correlate of Christian-Muslim educational differences, as Muslim children underperform in regions with considerable Muslim communities, a patter that is not present for Christians and Animists.") is the clearest statement of the findings and would serve the authors better in the introduction, followed by something along the lines of "factors X, Y, and Z do not seem to explain these patterns, and while religious segregation is the strongest correlate we are not able to say definitively *why* this is the case."

Relatedly, from the two paragraphs in the Introduction and three paragraphs in the Literature section, the authors note many different reasons why we might care about these patterns/gaps, but they end up mentioning so many different reasons that a clear motivation and sense of the stakes becomes muddled. They mention, among others – fast growing religious populations in Africa, high rates of religiosity, differentiating from work on ethnicity in Africa, post-independence policy and economic performance, religion and economics, and intergenerational transmission. While it is true that all these are related and relevant, it becomes hard to follow why specifically the authors think this is a pattern worth understanding and what the stakes are. I think they would be better served by picking a particular framing and bringing up later, perhaps in the conclusion, the many other reasons why we should care. I would also rather the literature

discussion be written in such a way that it informs the hypotheses they test rather than simply a list of contributions. That said, the authors should make sure that they note the several other works that have already documented level differences in educational attainment across religious groups in Africa (they already do this in the current manuscript, but I would not want it to get lost in the process of reworking the literature discussion). This latter discussion probably belongs in the introduction. Their new contribution is specifically to look at IM rather than level differences, which adds a lot of value, as well as testing possible explanations across many cases.

Regarding the results, there is a lot going on in Table 1, which makes it hard/tedious to read. This is of course up to the authors, but I wonder if they would be better served by a) moving some of this to a figure instead of a table and/or b) splitting up into multiple figures/tables where only one kind of outcome is shown at a time (e.g. either IM, the gap, districts/%of districts with a gap but not all of these at once). For example, I can imagine one set of figures that is a dot plot showing the IM estimates by country. This could even be ordered in such a way that the countries with the Christian-Muslim gap is at the top and smallest (negative gap) at the bottom. I didn't find the number of districts/regions (cols 6 and 13) that helpful and in fact it raised questions for me about whether these were indeed comparable units if Egypt could have 187 and Nigeria 14 while the population of Nigeria is twice that of Egypt. Should we be expecting similar patterns at any regional level/region size? Why or why not? I would be inclined to leave out cols 6/7 and 13/14 but in any case it would be helpful to say more about what regions actually are and how we should think about them politically, economically, and socially, particularly when it comes to sorting.

At the same time, the IM measures mask information about what percentage of cohorts are completing primary school or not, and thus what percentage could in theory be exhibiting upward/downward mobility. This information, provided in A.1, provides helpful context (though would help to keep the y-axis range constant across the subgraphs in this set of figures) – Uganda for example has quite high primary completion rates among Muslims whereas Nigeria does not yet the upward IM for Muslims is quite similar. How much should we care about the gap versus the absolute rate either from a theoretical or policy perspective? How should we expect IM rates to relate to country or region-level rates of parental primary schooling? In other words, while the authors focus primarily on the gap, it seems important as well to understand why IM is varying so dramatically across contexts in the first place. This discussion is quite limited in the current draft, though some of it can be found in the Appendix. Take Uganda and Nigeria again, the upward IM for Christian Nigerians is much higher than for Christian Ugandans. Both have relatively high rates of primary school completion in, say the 1960 cohort, though it is substantially higher among Christian Nigerians. Is the IM gap in Nigeria but not in Uganda really indicative of something about the Muslim population or rather the Christian population, since the Muslim IM is similar in both?

Regarding the overall organization, I would personally prefer a description of the various “theses” or hypotheses at the front end of the paper, followed by tests of all of them, rather than each individually. It makes it harder to follow the results and keep track of the punchline the way it is currently organized.

Other points:

- There is almost no discussion in the main paper about how the religious populations/demographics came to be, despite the fact that only a quarter of the (sub-Saharan) African population was either Christian or Muslim at the turn of the twentieth century and conversion was still ongoing into the period the authors are studying. As they are likely aware, the highest rates of conversion among Christians happened between the 1950s and 1970s, and the percentage of Christians and Muslims continued to rise, relative to traditional religions, up to the last decade (see for example: <https://www.pewforum.org/2010/04/15/executive-summary-islam-and-christianity-in-sub-saharan-africa/>). The authors note in the conclusion that “understanding conversion dynamics is vital” but don't say anything more about it in the paper. At the very least, some discussion would be helpful regarding the extent to which religious affiliation reported at the time of a census is likely to reflect religious affiliation at the time of primary education (e.g. how

often it was that someone identified as traditional religion as a child and only later as Christian or Muslim – I realize this cannot be answered with census data but we could at least be given a sense of how common this was), how this might vary across and within countries, and what implications it might have for the results. I think some brief discussions of conversion patterns across space and time within Africa might be important contextual information to include.

- Panel b of Figure 3 is hard to read because the labels for panel a do not line up with those in panel b. Also not clear why these are a different set of correlates in the first place? I think the homogeneity correlates are missing in panel b but I don't know why.
- Typo in the last sentence of the first para of section 7 "patter"
- I found this sentence confusing: "Religiosity is high funneling the salience of religious identities."
- The authors mention religious strife on several occasions – is there any evidence that these gaps are indeed related to religious strife? Political strife? This gets back to the stakes question – should we primarily care about this as an economic concern or something else? And is it actually an economic concern (perhaps Muslims are doing just as well as Christians in terms of income/well-being even in the absence of formal education, at least in some places)? If so, at the individual or country level (or something else)?

Referee #2 (Remarks to the Author):

Nature-2021-14644: "Religion and Educational Mobility in Africa"

I reviewed this paper at a prior journal and remain just as enthusiastic about its important contribution to the literature.

Summary

This paper provides some of the first systematic evidence on religious differences in intergenerational mobility in educational attainment across the African continent. The empirical setup builds on the authors' prior, foundational work on IM in Africa, recently published in *Econometrica*. The key innovation in the present study lies in the deep focus on the religious dimension. The findings are striking and point to pervasive educational disadvantages for non-Christian, primarily Muslim but also Animist, populations.

Overall, this paper adds important new evidence on religious divisions in the developing world. While neither the methods nor data are novel, the scope of analysis and findings are compelling. The results also raise interesting possibilities for future work aimed at understanding the historical origins and mechanisms for persistence (or lack thereof) in the religious gaps in IM. As is, the paper raises more questions than it answers in terms of the history- and policy-based drivers of these gaps. This is fine and may even be ideal for a Nature piece of this kind.

Below are a few comments that should be straightforward to address and hopefully add value on top of an otherwise strong paper.

Comments

1. At a few points in the intro, the authors allude to interreligious educational gaps being uniquely or especially large in Africa. Is this actually the case? It would be helpful to provide some substance, even in terms of simple descriptives, to support this claim. Here, it would also be useful to compare more deeply to the findings in Asher, Novosad, and Rafkin (2020) on the Hindu-Muslim IM gaps in India (I see some reference to this later in the paper in a footnote).

2. The paper rightly emphasizes the limited work on religious as opposed to ethnic divisions in Africa. However, it would be helpful to clarify empirically just how distinct religious and ethnic identities are. In particular, one wonders how much religion overlaps versus cuts across ethnicity. This is important for background but also on substance inasmuch as it would be interesting to know whether interreligious gaps in education levels and IM vary depending on whether those religious divisions occur within or solely across ethnic groups. That is, are the religious gaps smaller when the different religious groups share the same ethnicity? There is some limited discussion of this in Section 4.5 with all details relegated to Appendix D. Some of that detail would be useful here especially as I see one selling point of the paper's contribution being its effort to bring religion to the political economy literature on Africa which has mostly focused on ethnicity.

3. On presentation: (i) The paper needs some copy-editing as there a number of distracting typos, grammatical errors and missing words. (ii) The paper references many substantial and interesting appendices. Going back and forth between all of these and the paper will be difficult for some readers.(iii) The paper refers to many differences across groups and such but often without reference to standard errors. If these differences are always significant at conventional levels, then say so upfront. If not, then caveat accordingly. (iv) Sec. 4 has many interesting findings, but they read a bit like a laundry list. Better consolidating these differ-ent findings into a coherent framework that is also connected to the place-based correlates in Sec. 6 would help guide readers. For example, one suspects that occupational differences are more important in areas with more diversified economic activities close to transport infras-structure. Moreover, how might these occupational differences vary with segregation? While beyond the scope of a short paper, it would help to highlight some potential connections where instructive.

Referee #3 (Remarks to the Author):

- Key results: Please summarise what you consider to be the outstanding features of the work. The work uses large, comprehensive data sets from Africa, and establishes a pattern that Moslems have lower intergenerational mobility (IM), defined as increases in schooling attainment, than Christians. The approach is more of a 'rich description' of the data, though some groups of estimates might have stronger internal validity, such as those on moving to higher mobility regions. The key takeaway is that the Christian-Moslem gap in IM seems to stem largely from the initial geographic dispersion of the population, coupled with the fact that Moslems are more likely to live together, rather than a purposive decision by Moslems not to pursue education.
- Validity: Does the manuscript have flaws which should prohibit its publication? If so, please provide details.
There are no major 'validity' flaws that prevent publication. Some of the sub-samples analyzed are not representative of the entire continent (e.g. the data on movers), so there are concerns about external validity. Nevertheless, I think these are minor relative to what data is actually available (in other words, I don't think better data is available that could have been used). The authors could perhaps be more explicit and transparent about these sub-samples, and the extent to which they are select.
- Originality and significance: If the conclusions are not original, please provide relevant references. On a more subjective note, do you feel that the results presented are of immediate interest to many people in your own discipline, and/or to people from several disciplines?
The paper is closely related to the team's paper published this year in *Econometrica*, which uses the same data and methods, and looks at overall IM in the region, rather than Christian-Moslem differences. The originality stems from the research question (religious differences) and the comprehensiveness of the data used for the analysis. Other studies have looked at these differences at the local or country level, but never at this level with so much data.
On the question of whether the results are of immediate interest-I think that religiosity as a factor in social and economic well-being is of broad interest to researchers in many different disciplines in the Humanities and Social Sciences. The contribution of this paper is that it looks at Christian-Moslem differences comprehensively across the continent, using data from 20 countries over multiple time periods.

The paper goes beyond simply establishing religious differences. A key finding is that when Moslems move to high mobility regions, their IM goes up, and that contextual factors influence Christians and Moslems equally. This is important, though it doesn't quite come out clearly in the current presentation. The gap in IM is due to initial contextual conditions, and the clustering of Moslems in these regions. The paper should really highlight these key findings, and expand on their implications.

- Data & methodology: Please comment on the validity of the approach, quality of the data and quality of presentation. Please note that we expect our reviewers to review all data, including any extended data and supplementary information. Is the reporting of data and methodology sufficiently detailed and transparent to enable reproducing the results?

The data and methodology is identical to that used in their *Econometrica* paper (2021). The quality of the methodology is good. As mentioned above, the authors could be more transparent about some of the sub-sample analysis. Most important in this regard is the sub-samples used for the regional migration results, which use data only from countries that report place of birth and current residency and allow researchers to infer migration. As the more interesting results are based on this analysis, we need a more transparent reporting of the data limitations here.

The Appendix is very detailed and provides the necessary information to assess the work and what has been done.

- Appropriate use of statistics and treatment of uncertainties: All error bars should be defined in the corresponding figure legends; please comment if that's not the case. Please include in your report a specific comment on the appropriateness of any statistical tests, and the accuracy of the description of any error bars and probability values.

Figures 3 and 4 in the text do not define the error bars (nor are they clearly defined in the appendix). The statistical tests are appropriate.

- Conclusions: Do you find that the conclusions and data interpretation are robust, valid and reliable?

There is no conclusion, but rather a Section 7 on Future Research, which I found somewhat out of the blue, as the suggestions don't follow naturally from the empirical results. First and foremost we need a Conclusion and/or Discussion Section. There are a lot of data and estimates, the paper is very dense, aside from the 'Preview of Results', the reader has a hard time clearly identifying the key takeaways.

The 'Future Research' section can be dropped, or integrated into the conclusion, and trimmed down. As I said, they come out of nowhere. The key results of the paper as I see them is that Christian-Moslem gaps in IM are widespread, that when they favor Moslems it is when the Moslems population is a very small share of the population, and that the gaps are correlated with initial contextual factors, including the share of Moslems in the region. When Moslems move to higher mobility regions their IM goes up. One obvious future research question that stems from this is whether social infrastructure has been purposefully limited in predominantly Moslem areas, whether there have been policies to limit the mobility of Moslems (for example in Malawi residents of South Asian descent, mostly Moslems, were not allowed in certain parts of the country), and examples of how to reduce the IM gap. These all flow tightly from the evidence produced in the paper, and help us understand why we see the results that we do in the paper. In fact, the last sentence of the Abstract is: "Our findings call for more research on the origins of religious segregation and the role of religion-specific, institutional, and social conventions on education and opportunity." This sentence raises topics that come out of the research, unlike Section 7 of the text.

- Suggested improvements: Please list additional experiments or data that could help strengthening the work in a revision.

See above, the main concern is the presentation, and a focus on the key results, so more on the writing and presentation than on further analyses.

- References: Does this manuscript reference previous literature appropriately? If not, what references should be included or excluded?

Yes

- Clarity and context: Is the abstract clear, accessible? Are abstract, introduction and conclusions

appropriate?

Yes

- Please indicate any particular part of the manuscript, data, or analyses that you feel is outside the scope of your expertise, or that you were unable to assess fully.

As I do not routinely read nature, I don't know the audience, and so can't say whether they will find these results of interest. Technically though, the paper is sound.

Sudhanshu Handa

Referee #4 (Remarks to the Author):

This paper makes two contributions. First, it uses harmonized census data from nineteen African countries to estimate the intergenerational (upward and downward) success in finishing primary school across religious denominations. The dataset that the authors constructed is a public good that will enable future researchers who study education and religion in Africa to make further inroads. Second, in order to account for the (already well-known but here well-quantified) Muslim disadvantage, the paper runs a series of correlations (e.g. percentage of population of each religion; within country regional migration) to estimate the associations between religion and educational mobility. Significant results (in line with Chetty's work in the US) show that migration from a low education region in a country to a higher education region at an early age (for both religions) predicts higher intergenerational success in finishing primary school.

Alas, the paper is not ready for publication in a top tier journal. The writing and the data remain "uncooked" (Lévi-Strauss). Some examples follow:

*The acronym "IM" is not spelled out until the notes to Figures 2 and 3 and in SI.

*The legend and notes for Figure 1 are incommensurate or confusing.

*There is a sense that the authors are fishing with arbitrary sub-group analyses that were never posted in a pre-analysis plan (e.g. "Household structure and size play a somewhat bigger role in narrowing the Christian-Muslim IM gap in West Africa; their influence is negligible in East and Southern Africa. Family arrangements play a somewhat more significant role in explaining religious IM differences for girls; for downward mobility the gap drops from 0.08 to 0.06").

*The submitted paper reflects first drafted-ness in all too many uncorrected sentences (e.g. "As the specification includes origin-cohort constants, variation comes from children born in the same place and decade who move to regions with different mobility.")

As for substance, in a figure exploring the finding about migration across regions – a nice correlation – the data show a significant relationship to moving closer to where there was a Protestant missionary station. The authors ignore this in the body of the paper, but it is certainly worth greater attention, and is in accord with the dissertation of Platas (and co-authors in a preprint), whose findings are mentioned but not tested. Laitin, in his book *Hegemony and Culture* (1986) also emphasizes Lord Lugard's restriction of missionary activity in the Hausa (predominantly Muslim) region. (Laitin's book is a partial exception to author claims that religion has been ignored in work on cleavages in Africa, but also helps explain why ethnicity trumps religion in scholarly attention to African cleavage structures).

Not exploring the Protestant missionary factor (as Nunn and elsewhere Woodberry analyze) is indeed the biggest gap in this paper's account of the differential IM across the religious divide. One can easily see that over time schools with missionary roots were more successful in sustaining educational opportunity to those in their catchment areas. This demands exploration more than one variable on a box plot (Figure 3).

The authors divide the population into three categories: Christian, Muslim, and Animist. As many Africanists have pointed out (e.g. J.D.Y. Peel's book *Aladura*) animist prayer churches are often

complementary religious identifications within the same person and not substitutes. It could well be the case that the more educated the young generation, the more likely they will report the "Christian" complement of their dual identification rather than the "Animist".

The admission by authors that "Our study begets more questions than it answers" sums up my evaluation.

Author Rebuttals to Initial Comments:

Religion and Educational Mobility in Africa. Nature-2021-14644. Reply Referee Report 1.

We are thankful for your valuable comments and constructive feedback. We believe that the revision tackles all comments raised in your report. Below we copy your comment/suggestion (in *italics*) and detail how we address it in the revision. We also indicate (with underlined fonts) the part of the paper, main body, methods, and Supplementary Information where we discuss your remark.

General Comment *My comments and suggestions primarily regard the discussion of significance, clarity and context, and conclusions of the paper. I believe the paper would be improved by 1) owning up to the puzzle the authors present at the outset, 2) finding clearer ways to present some of the main descriptives, and 3) organizing the prose in a way that is not just a march through results, 4) being clearer about what is at stake and into what conversations the piece is fitting.*

Reply. Thanks for your helpful suggestion. We have followed them closely in the revision.

Comments. Paper Writing and Positioning. Introduction and Conclusion.

1. **Comment [Owning the Puzzle].** *"This paper has important patterns to document and does not present an "answer" to why these patterns exist, rather ruling out a set of plausible explanations while raising more questions. I think this is fine, but the paper would be made stronger by owning this puzzle. The first paragraph of the conclusion ("We uncover substantial differences in educational mobility between Muslims, Christians, and Animists across Africa post-independence. Regional effects appear first order causally affecting upward mobility for children of all faiths. Religious segregation is the most important correlate of Christian-Muslim educational differences, as Muslim children underperform in regions with considerable Muslim communities, a pattern that is not present for Christians and Animists.") is the clearest statement of the findings and would serve the authors better in the Introduction, followed by something along the lines of "factors X, Y, and Z do not seem to explain these patterns, and while religious segregation is the strongest correlate we are not able to say definitively *why* this is the case."*

Reply. Thanks. Following your suggestion, we try to "own" the puzzle from the onset. We have added a concise discussion of the regularities and the puzzle in the abstract, Introduction (Section 1), and the Discussion Section. Please see the revised abstract, Introduction (pages 1-2), and Discussion (pages 14-15).

2. **Comment [Introduction and Conclusion].** *"Relatedly, from the two paragraphs in the Introduction and three paragraphs in the Literature section, the authors note many different reasons why we might care about these patterns/gaps, but they end up mentioning so many different reasons that a clear motivation and sense of the stakes becomes muddled. They mention, among others – fast-growing religious populations in Africa, high rates of religiosity, differentiating from work on ethnicity in Africa, post-independence policy and economic performance, religion and economics, and intergenerational transmission. While it is true that all these are related and relevant, it becomes hard to follow why specifically the authors think this is a pattern worth understanding and what the stakes are. I think they would be better served by picking a particular framing and bringing up later, perhaps in conclusion, the many other reasons why we should care. I would also rather the literature discussion be written in such a way that it informs the hypotheses they test rather than simply a list of contributions. That said, the authors should make sure that they note the several other works that have already documented level differences in educational attainment across religious groups in Africa (they already do this in the current manuscript, but I would not want it to get lost in the process of reworking the literature discussion). This latter discussion probably belongs in the Introduction. Their new contribution is specifically to look at IM rather than level differences, which adds a lot of value, as well as testing possible explanations across many cases."*

Reply: Thanks for this suggestion.

First, in the revised draft, we discuss the large number of Christian and Muslim communities in Africa, which are expected to grow in the decades to come.

Second, we have shortened the Related Literature part of the introduction. Per the Editorial suggestion, we have tried to have a better flow of the argument without digressions.

Third, we have moved some technical issues in the online Supplementary Information (Appendix).

Most importantly, following your suggestion, we prominently stress (in a few parts of the paper) that our focus is on educational mobility (IM). We emphasize three core results. First, religion-specific differences in the level of education are important drivers of inter-religious differences in upward and downward IM. Second, IM gaps remain significant even when we account for differences in educational levels across groups. Besides, interreligious differences in IM are non-negligible, even when we compare children from (equally) uneducated parents, residing in the same region, with similar educational levels to the old generation, having comparable household characteristics, and parental occupational features. Moreover, the newly crafted section on internal migration differences

between religious groups attempts to go a step further in resolving the puzzle. [Please see (new) Section 7.]

We hope that you find the revised draft easier to follow. In this regard, we have added a summary paragraph at the end of each Section to allow readers to grasp our results' logic better.

Comments on Results.

3. **Comment [Table 1].** *"Regarding the results, there is a lot going on in Table 1, which makes it hard/tedious to read. This is of course up to the authors, but I wonder if they would be better served by a) moving some of this to a figure instead of a table and/or b) splitting up into multiple figures/tables where only one kind of outcome is shown at a time (e.g. either IM, the gap, districts/%of districts with a gap but not all of these at once). For example, I can imagine one set of figures that is a dot plot showing the IM estimates by country. This could even be ordered in such a way that the countries with the Christian-Muslim gap is at the top and smallest (negative gap) at the bottom. I didn't find the number of districts/regions (cols 6 and 13) that helpful and in fact it raised questions for me about whether these were indeed comparable units if Egypt could have 187 and Nigeria 14 while the population of Nigeria is twice that of Egypt. Should we be expecting similar patterns at any regional level/region size? Why or why not? I would be inclined to leave out cols 6/7 and 13/14 but in any case it would be helpful to say more about what regions actually are and how we should think about them politically, economically, and socially, particularly when it comes to sorting."*

Reply. Thank you for your helpful remark. Indeed Table 1 of the initially submitted draft was dense and hard to follow. Apologies. We have dropped from Table 1 the columns with the regional information to make it more transparent and easier to follow. Please see (new) Table 1 (page 4). We also follow your suggestion of ordering countries by Christian-Muslim differences in upward IM. [We do so also in Online Supplementary Information Tables].

We have moved the district statistics to another Table. Please see Extended Data Table 2 for regional statistics per country. Moreover, we hope that the new Section on Internal Migration and the rewritten role of religious segregation gives a more concrete idea on the role of regions.

4. **Comment [IM Statistics].** *"At the same time, the IM measures mask information about what percentage of cohorts are completing primary school or not, and thus what percentage could in theory be exhibiting upward/downward mobility. This information, provided in A.1, provides helpful context (though would help to keep the y-axis range constant across the subgraphs in this set of figures) – Uganda for example has quite high primary completion rates among Muslims whereas*

Nigeria does not yet the upward IM for Muslims is quite similar. How much should we care about the gap versus the absolute rate either from a theoretical or policy perspective? How should we expect IM rates to relate to the country or region-level rates of parental primary schooling? In other words, while the authors focus primarily on the gap, it seems important as well to understand why IM is varying so dramatically across contexts in the first place. This discussion is quite limited in the current draft, though some of it can be found in the Appendix. Take Uganda and Nigeria again, the upward IM for Christian Nigerians is much higher than for Christian Ugandans. Both have relatively high rates of primary school completion in, say the 1960 cohort, though it is substantially higher among Christian Nigerians. Is the IM gap in Nigeria but not in Uganda really indicative of something about the Muslim population or rather the Christian population, since the Muslim IM is similar in both?"

Reply: Thanks for raising this issue that we did not discuss much in the initial submission.

First, we have added a brief discussion of educational differences across religious affiliations at independence in the very beginning. Please see the first paragraph of the Introduction (page 1).

Second, as we had to comply with the Editorial suggestion of keeping the paper short and your core suggestion to have a good flow of the argument, we include a more thorough discussion of initial (at independence) differences across religious affiliations in education in the Supplementary Information (SI). Please see SI Section A [pages 1-6]. Please also see SI Tables A.1-A.2 and SI Figures A.1 and A.2.

Third, in the *Methods* sub-section “*Educational Dynamics, Upward and Downward Intergenerational Mobility,*” we show how the newly-compiled upward and downward IM measures can help us quantify the steady state level of primary school completion across religious groups. Please see Methods (page 1).

Fourth, to explore how the level of education of the old generation of coreligionists in the district influences the estimated religious gaps in upward IM and downward IM we (i) control directly for the former and (ii) we zoom in on regions where the primary school attendance rates in the old generations of Muslims, Christians, and Animists are comparable. Please see Figure 2, rows 4 and 6, alongside the discussion in Section 4, Initial Literacy (page 7) and Regional Features (page 8).

5. Comment [Results Presentation, Drivers of IM]. *"Regarding the overall organization, I would personally prefer a description of the various "theses" or hypotheses at the front end of the paper, followed by tests of all of them, rather than each individually. It makes it harder to follow the results and keep track of the punchline the way it is currently organized."*

Reply. Thanks for your remark. We thought of restructuring the Section on the drivers of the inter-religious gaps in upward and downward IM (Section 4). Yet, we were not very satisfied, and thus we reverted to the original style. However, as we have trimmed Section 4 to comply with *Nature's* style and length guidelines, we believe the discussion is now more coherent.

Moreover, to address your suggestion to keep track of the punchline, we conclude the Section with a paragraph summarizing the significant results. Doing so allows the reader to follow the results. Of course, if you find the Section still hard to follow, we can rework it. Please see the reworked Section 4 and, particularly the Taking Stock concluding paragraph (page 8).

Other Comments

6. **Comment [Origins of Educational Differences].** *"There is almost no discussion in the main paper about how the religious populations/demographics came to be, despite the fact that only a quarter of the (sub-Saharan) African population was either Christian or Muslim at the turn of the twentieth century and conversion was still ongoing into the period the authors are studying. As they are likely aware, the highest rates of conversion among Christians happened between the 1950s and 1970s, and the percentage of Christians and Muslims continued to rise, relative to traditional religions, up to the last decade (see for example <https://www.pewforum.org/2010/04/15/executive-summary-islam-and-christianity-in-sub-saharan-africa/>). The authors note in the conclusion that "understanding conversion dynamics is vital" but don't say anything more about it in the paper. At the very least, some discussion would be helpful regarding the extent to which religious affiliation reported at the time of a census is likely to reflect religious affiliation at the time of primary education (e.g. how often it was that someone identified as traditional religion as a child and only later as Christian or Muslim – I realize this cannot be answered with census data but we could at least be given a sense of how common this was), how this might vary across and within countries, and what implications it might have for the results. I think some brief discussions of conversion patterns across space and time within Africa might be important contextual information to include."*

Reply. You are gently raising a critical point. We had abstained from discussing conversion dynamics because we thought this is such an essential element and addressing it satisfactorily would not be possible. Of course, you are correct that leaving such a discussion out was not appropriate. Therefore, in the revision, we thought long about how to incorporate the conversion discussion in a meaningful manner without derailing the flow.

First, we formally explore religious conversion using Census data. Looking at multi-generational families, we estimate the probability of a young individual reporting being Muslim/Christian/Animist conditional on his/her parent's creed. We find high inertia in the inter-generational transmission of Christianity and Islam. The intergenerational transmission coefficient exceeds 0.97 for Christians and

Muslims across all birth cohorts, irrespective of the child's education. Converting out of Christianity and Islam has been the exception in post-colonial Africa. The pattern is different among Animists. A sizable fraction of children born to Animist parents converts to Christianity –less to Islam. Besides, whether a child of Animist parents converts to Christianity depends on education. Roughly half of children with completed primary education born to Animist parents report they are Christian and the other half (46%) report an indigenous religion. The corresponding statistics for children without primary schooling are 25% and 70%, respectively. This 25-percentage points gap reflects how educational improvements of the Animist population in post-colonial Africa moved in tandem with its Christianization. This pattern may have implications for the religion-specific IM estimates since we categorize a household as Animist/Christian/Muslim if the child reports being Animist/Christian/Muslim, respectively. We replicated the benchmark analysis, reported in Figure 2, focusing on households where both the old and the young report the same religion. The estimates (not shown for brevity but reported below for your convenience) are similar, as the overall number of Animists is small. See the Figures below.

Thanks to your remark -and a similar comment from R4, we discuss these issues in the revision. Please see the Subsection “Inter-generational Transmission of Religious Affiliation” in the Methods Section (pages 1-2). We have included a detailed discussion in the online SI Section B.3 (pages 17-19) and the SI Figures B1, B2, and B3.

Second, we discuss the *Pew Research Center* report you recommended on religious conversion. Interestingly, our census-based estimates of minimal Christian-Muslim conversion and vice versa square well with the estimates in this report, documenting minimal conversion between Christians and Muslims. Please see our discussion in the Subsection “Inter-generational Transmission of Religious Affiliation” in the Methods (pages 1-2).

Figure 1: Current draft: full sample

Figure 2: Religion of kid = previous generation religion

7. Comment [Typos and Errors].

"Panel b of Figure 3 is hard to read because the labels for panel a do not line up with those in panel b. Also not clear why these are a different set of correlates in the first place? I think the homogeneity correlates are missing in panel b but I don't know why.

Typo in the last sentence of the first para of section 7 "patter"

I found this sentence confusing: "Religiosity is high funneling the salience of religious identities."

Reply. Thanks for spotting these typos. Apologies. We have corrected them. We have also proofread the paper repeatedly. Hopefully, there are no more errors and typos.

8. Comment [Link to Religious Strife]. "The authors mention religious strife on several occasions – is there any evidence that these gaps are indeed related to religious strife? Political strife? This gets back to the stakes question – should we primarily care about this as an economic concern or something else? And is it actually an economic concern (perhaps Muslims are doing just as well as Christians in terms of income/well-being even in the absence of formal education, at least in some places)? If so, at the individual or country level (or something else)?"

Reply. We have removed the phrases linking inter-religious education differences to religious tensions and civil conflict from the introduction and the main sections. We have kept a brief (half) sentence in the conclusion, as we believe the link is worth examining in follow-up work. Moreover, we mention in the main text that there appear to be no striking differences in terms of living conditions, occupational structure, and household income from the various surveys between Muslims and Christians. Seen in this light the considerable educational gaps both in levels and in IM between the two groups suggest that we need to revisit the differential economic role of education for each group. We mention this in conclusion. Please see the last section, Discussion (pages 14-15).

Thanks again for your helpful feedback and positive reaction to our work. We sincerely appreciate the time you put into our work and your constructive feedback.

We are thankful for your valuable comments and constructive feedback. We believe that the revision tackles all comments raised in your report. Below we copy your comment/suggestion (in *italics*) and detail how we address it in the revision. We also indicate (with underlined fonts) the part of the paper, main body, methods, and Supplementary Information where we discuss your remark. [For brevity, we omit the Summary of your Referee Report].

Muslim-Christian Educational Differences in Africa and Elsewhere. *"At a few points in the intro, the authors allude to interreligious educational gaps being uniquely or especially large in Africa. Is this actually the case? It would be helpful to provide some substance, even in terms of simple descriptives, to support this claim. Here, it would also be useful to compare more deeply to the findings in Asher, Novosad, and Rafkin (2020) on the Hindu-Muslim IM gaps in India (I see some reference to this later in the paper in a footnote)."*

Reply. Thanks for this remark that we address twofold.

First, we discuss the similarities of our findings with India, drawing on the parallel study of Asher, Novosad, and Rafkin (2022). Specifically, we write: *"In parallel work, Asher et al. (2020) explore regional, caste, and religious differences in IM across India; like us, they uncover low and declining educational mobility for the Muslim population, which contrasts to the rising mobility of low socio-economic status castes."* Please see our discussion in Section 1 (main paper, page 2) and at the end of Section 4 (Regional Features, page 8).

Second, in the revised draft of their paper, Asher *et al.* (2021), building on our work, also find a negative correlation between segregation and upward mobility. Therefore, we also comment on this similarity when we present our results. Please see the end of Section 6 (page 12).

Religious Affiliation vs Ethnicity. *"The paper rightly emphasizes the limited work on religious as opposed to ethnic divisions in Africa. However, it would be helpful to clarify empirically just how distinct religious and ethnic identities are. In particular, one wonders how much religion overlaps versus cuts across ethnicity. This is important for the background but also on substance inasmuch as it would be interesting to know whether interreligious gaps in education levels and IM vary depending on whether those religious divisions occur within or solely across ethnic groups. That is, are the religious gaps smaller when the different religious groups share the same ethnicity? There is some limited discussion of this in Section 4.5 with all details relegated to Appendix D. Some of that detail would be useful here especially as I see one selling point of the paper's contribution being its effort to bring religion to the political economy literature on Africa which has mostly focused on ethnicity."*

Reply. Thanks for stressing the link between ethnicity and religion, which we did not discuss much in the initial draft. We now discuss the relationship between ethnicity and religion in the main paper and comment on intra-ethnicity differences in religious affiliation and within-religion ethnic fragmentation. Please see the Methods Section [Ethnicity and Religion], page 2.

We have added descriptive statistics and a richer discussion with many examples [multi-religious ethnicities] in the online Supplementary Information (SI). Please see Supplementary Information (SI) Section B.4 (pages 19-20). We have added a graph with the HHI index of religious concentration by ethnicity (Figure B.5)

Moreover, in the Supplementary Information (SI Section D.2.2) we discuss how the benchmark patterns change when we compare individuals within the same ethnic group but of different religious affiliations. Please also see Figure D.3 (page 29) in the SI.

Comments on the Presentation.

Typos. *"The paper needs some copy-editing as there are number of distracting typos, grammatical errors, and missing words."*

Reply. Apologies for the typos and mistakes. We have proofread the paper, and hopefully, we have eliminated such errors.

Appendixes. *"The paper references many substantial and interesting appendices. Going back and forth between all of these and the paper will be difficult for some readers."*

Reply. We have now written the paper following *Nature's* style. As we had to keep the article short -and per the Editor's suggestion, somewhat shorten it- we moved auxiliary results to the Supplementary Information Section (Appendix). Besides, we have rewritten the paper to make it sharper, more concise, and with a better flow of its core ideas. We hope you find our argument's flow clearer and the writing better.

Standard Errors. *"The paper refers to many differences across groups and such but often without reference to standard errors. If these differences are always significant at conventional levels, then say so upfront. If not, then caveat accordingly."*

Reply. Thanks for flagging inference. We omit standard error bands to avoid cluttering the figures and allow the reader to visualize the patterns more easily.

We have amended the notes of Figures 2 and 3 to discuss statistical significance and inference. In Figure 2 notes, we write. "The figures omit standard-error bands to enable clear visualization of the

patterns. The estimates on the Muslim indicator are statistically significant across all twelve specifications, while the Animist coefficient is significant across all upward IM specifications.” In Figure 3 notes we state at the end: “The age-of-move estimates are statistically significant at standard confidence levels. The panels omit standard errors to avoid cluttering.”

Streamline Results Presentation. *"Section 4 has many interesting findings, but they read a bit like a laundry list. Better consolidating these different findings into a coherent framework that is also connected to the place-based correlates in Sec. 6 would help guide readers. For example, one suspects that occupational differences are more important in areas with more diversified economic activities close to transport infrastructure. Moreover, how might these occupational differences vary with segregation? While beyond the scope of a short paper, it would help to highlight some potential connections where instructive."*

Reply. We have tried to connect more tightly the various sections and have a better flow of the argument. In this regard, we have added summaries at the end of each section with the key takeaways. While doing so takes some space, it is quite helpful to the reader.

Thanks to your comment, we caveat that the multiple aspects we consider may interact, for example, segregation and occupational differences. Please see Section 4 (approach) after presenting the main variables (page 4). We also discuss the country-specific estimates on the Christian-Muslim upward and downward IM gap in the main paper, reported in Extend Data Fig. 2. Please see the end of page 4 and the discussion in Section 4.

Thanks again for your helpful feedback and positive reaction to our work. We sincerely appreciate the time you put into our work and your constructive feedback.

We are thankful for the valuable comments and constructive feedback. We believe that the revision tackles all comments raised in your report. Below we copy your comment/suggestion (in *italics*) and detail how we address it in the revision. We also indicate (with underlined fonts) the part of the paper, main body, methods, and Supplementary Information (SI) where we discuss your remark.

Presentation. *“The main concern is the presentation, and a focus on the key results, the writing and presentation than on further analyses.”*

Reply. Thanks for pushing us to rework the presentation of the empirical regularities. We have rewritten the paper, following the structure guidelines of *Nature*. Doing so allows us to stress the most novel patterns. We have also tried to enhance the flow of the argument across the paper sections, relegating non-essential material in the online Supplementary Information [Appendix].

Overview. *“There are no major ‘validity’ flaws that prevent publication. Some of the sub-samples analyzed are not representative of the entire continent (e.g., the data on movers), so there are concerns about external validity. Nevertheless, I think these are minor relative to what data is actually available (in other words, I don't think better data is available that could have been used). The authors could perhaps be more explicit and transparent about these sub-samples, and the extent to which they are select.” ... “As mentioned above, the authors could be more transparent about some of the sub-sample analysis. Most important in this regard is the sub-samples used for the regional migration results, which use data only from countries that report place of birth and current residency and allow researchers to infer migration. As the more interesting results are based on this analysis, we need a more transparent reporting of the data limitations here.”*

Reply. Thanks for your positive comment and valuable suggestion.

In the revised draft, we discuss the differences and similarities of the sample used in section that aims to isolate childhood regional exposure effects from spatial sorting with the larger sample used in the other sections. We have added a discussion on the representativeness of the 13 countries with the wider 21-country sample. We do so both when we discuss the mover's design approach that isolates regional childhood exposure effects and in the Methods part. Please see Section 5 (end of Empirical Design, page 9). Please also see the Methods (Sample in page 3), where we have added a detailed discussion.

Figures standard errors. *“Figures 3 and 4 in the text do not define the error bars (nor are they clearly defined in the appendix).”*

Reply. Thanks for flagging inference. We omit standard error bands to avoid cluttering the figures and allow the reader to visualize the patterns more easily. We have amended the notes of Figures 2 and 3 to discuss statistical significance and inference. In Figure 2 notes, we write. “The figures omit standard-error bands to enable clear visualization of the patterns. The estimates on the Muslim indicator are statistically significant across all twelve specifications, while the Animist coefficient is significant across all upward IM specifications.” In Figure 3 notes we state at the end: “The age-of-move estimates are statistically significant at standard confidence levels. The panels omit standard errors to avoid cluttering.”

Besides, we report parametric estimates (with standard errors) of childhood regional exposure effects alongside standard errors in the Supplementary Information [Section E]. The regression estimates, which exploit within-household across sibling variation, also reveal childhood regional exposure effects for Christians and Muslims. Please see SI Section E, Tables E.3-E.5 [pages 38-43].

Contribution. *"The paper is closely related to the team's paper published this year in *Econometrica*, which uses the same data and methods, and looks at overall IM in the region, rather than Christian-Moslem differences. The originality stems from the research question (religious differences) and the comprehensiveness of the data used for the analysis. Other studies have looked at these differences at the local or country level, but never at this level with so much data. On the question of whether the results are of immediate interest-I think that religiosity as a factor in social and economic well-being is of broad interest to researchers in many different disciplines in the Humanities and Social Sciences. The contribution of this paper is that it looks at Christian-Moslem differences comprehensively across the continent, using data from 20 countries over multiple time periods. The paper goes beyond simply establishing religious differences. A key finding is that when Moslems move to high mobility regions, their IM goes up, and that contextual factors influence Christians and Moslems equally. This is important, though it doesn't quite come out clearly in the current presentation. The gap in IM is due to initial contextual conditions, and the clustering of Moslems in these regions. The paper should really highlight these key findings, and expand on their implications."*

Reply. Thanks for your kind remark. In the revised draft, we stress the vast differences in educational mobility between Christians and Muslims, even when we look at individuals with similar observable features residing in the same region. Please see the reworked Abstract, Introduction (Section 1), and Discussion (Section 8), alongside the ending lines of Sections 4 and 5 (Taking Stock).

Conclusion. *"There is no conclusion, but rather a Section 7 on Future Research, which I found somewhat out of the blue, as the suggestions don't follow naturally from the empirical results. First and foremost, we need a Conclusion and/or Discussion Section. There are a lot of data and estimates, the paper is very dense, aside from the 'Preview of Results', the reader has a hard time clearly identifying the key takeaways. The 'Future Research' Section can be dropped, or integrated into the conclusion, and trimmed down. As I said, they come out of nowhere. The key results of the paper as I see them is that Christian-Moslem gaps in IM are widespread, that when they favor Moslems it is when the Moslems population is a very small share of the population, and that the gaps are correlated with initial contextual factors, including the share of Moslems in the region. When Moslems move to higher mobility regions their IM goes up. One obvious future research question that stems from this is whether social infrastructure has been purposefully limited in predominantly Moslem areas, whether there have been policies to limit the mobility of Moslems (for example in Malawi residents of South Asian descent, mostly Moslems, were not allowed in certain parts of the country), and examples of how to reduce the IM gap. These all flow tightly from the evidence produced in the paper, and help us understand why we see the results that we do in the paper. In fact, the last sentence of the Abstract is: "Our findings call for more research on the origins of religious segregation and the role of religion-specific, institutional, and social conventions on education and opportunity."*

Reply. Thanks for these helpful suggestions on the exposition of the many results.

First, we have added short summaries at the end of each of the main section to guide the reader and enhance the flow of our argument. Second, we have reworked the concluding Section, along your suggestions. We have integrated the two subsections (summary and future research). In the revised draft, the concluding Section summarizes the paper's findings, discusses implications, listing some inquiries that deserve further research. We hope that you find the reworked conclusion clearer and sharper. But if not, we can have another go. Please see Section 8 (Discussion), pages 14-15.

Thanks again for your helpful feedback and positive reaction to our work. We sincerely appreciate the time you put into our work and your constructive feedback.

We are thankful for your valuable comments and constructive feedback. We believe that the revision tackles all comments raised in your insightful report. Below we copy your comment/suggestion (in *italics*) and detail how we address it in the revision. We also indicate (with underlined fonts) the part of the paper, main body, methods, and Supplementary Information (SI) where we discuss your remark.

Writing and Data

Acronym. *“The acronym “IM” is not spelled out until the notes to Figures 2 and 3 and in SI.”*

Reply. Apologies for this omission. We now define IM, intergenerational mobility, in the abstract and the Introduction of the main paper.

Notes. Figure 1. *“The legend and notes for Figure 1 are incommensurate or confusing.”*

Reply. Please accept our apologies for the unclear notes in Table 1. This table was quite dense, an issue that other reviewers also raised. We have reworked the notes of all tables and figures. We have also streamlined tables and figures. As Table 1 was dense, we dropped several columns. And following another referee’s advice, we sorted countries according to the Christian-Muslim differences in upward intergenerational mobility (IM). Please see the revised Table 1 and its notes (page 4).

Conditioning Sets Comment. *“There is a sense that the authors are fishing with arbitrary sub-group analyses that were never posted in a pre-analysis plan (e.g., “Household structure and size play a somewhat bigger role in narrowing the Christian-Muslim IM gap in West Africa; their influence is negligible in East and Southern Africa. Family arrangements play a somewhat more significant role in explaining religious IM differences for girls; for downward mobility the gap drops from 0.08 to 0.06”).*

Reply. Thanks to your remark, we now clarify the objectives of the descriptive part of the paper and the variance decomposition, which, as you point out, was not based on a pre-analysis plan. Our first aim is to document the patterns in educational mobility of Christians, Muslims, and Africans adhering to Traditional Religions using the largest sample available across African countries, regions, and birth cohorts. Please note that we are using all censuses from IPUMS with information on religious affiliation. Besides having the largest possible sample, we want to minimize concerns about sample selection. As there was an update in IPUMS, we have expanded coverage, integrating Mauritius and adding recent censuses from the other countries.

Our second task is to quantify how much of the disparities in educational mobility between Christians, Muslims, and Animists reflect inter-religious differences in family arrangements, household size, and structure that differ across faiths. As our objective is not to study the role of a particular household

feature, like size or composition, there should not be concerns about cherry-picking the set of explanatory variables. We use household structure and size measures and parental occupation-profession available from censuses to explore how much they explain the considerable inter-religious differences in upward and downward educational intergenerational mobility in education.

Thanks to your remark, we now clarify these issues at the beginning of the relevant Section. Please see our discussion at the beginning of Section 4 [page 4] and our discussion of the results in Figure 2 [subsection Family and Household Features], top of page 7. There we write: “We include all variables concurrently, as our objective is not to identify the most relevant feature, but to examine how much the religion estimates move when we account for a saturated set of household traits.”

Writing. *“The submitted paper reflects first drafted-ness in all too many uncorrected sentences (e.g., “As the specification includes origin-cohort constants, the variation comes from children born in the same place and decade who move to regions with different mobility.”)”*

Reply. Apologies for unclear wording, typos, and grammatical mistakes. We have reworked our paper, spell-checked it, and proofread it. We hope you find the revised draft more fluid, without unclear sentences and typos.

Substantive Comments

Comment. Migration and Missions’ Role. *“As for substance, in a figure exploring the finding about migration across regions – a nice correlation – the data show a significant relationship to moving closer to where there was a Protestant missionary station. The authors ignore this in the body of the paper, but it is certainly worth greater attention, and is in accord with the dissertation of Platas (and co-authors in a preprint), whose findings are mentioned but not tested. Laitin, in his book *Hegemony and Culture* (1986) also emphasizes Lord Lugard’s restriction of missionary activity in the Hausa (predominantly Muslim) region. (Laitin’s book is a partial exception to author claims that religion has been ignored in work on cleavages in Africa, but also helps explain why ethnicity trumps religion in scholarly attention to African cleavage structures). Not exploring the Protestant missionary factor (as Nunn and elsewhere Woodberry analyze) is indeed the biggest gap in this paper’s account of the differential IM across the religious divide. One can easily see that over time schools with missionary roots were more successful in sustaining educational opportunity to those in their catchment areas. This demands exploration more than one variable on a box plot (Figure 3).”*

Reply. Thank you for pushing us to clarify the role of (Protestant) missionary activity and rethink the section on the regional correlates of the religious IM gap. Let us first describe what the key findings are.

First, we show that compared to Christians, Muslims reside in districts further away from colonial roads and railroads, and in regions far from Protestant and Catholic Missions; this pattern is in line with a voluminous literature on Africa's historical development; examples include Nunn and Woodberry. Please see the results in Extended Data Figure 3 and our discussion in the main paper (Section 6, page 10).

Second, in Extended Data Fig. 4 we show that individuals residing in districts closer to Protestant missions register higher IM rates. This result is also in line with in Alesina *et al.* (ECTA, 2021). This finding is consistent with your prior, i.e., “*over time schools with missionary roots were more successful in sustaining educational opportunity to those in their catchment areas*”. So, indeed the lower numbers of Muslims in regions with higher initial economic and educational conditions as captured by Protestant and Catholic missionary activity (but also in other factors including proximity to the coast, capital cities, and the colonial transportation network (roads and railroads)), is of first-order importance for explaining the Christian-Muslim IM differences across regions.

Third, in Section 6 when we compare interfaith IM gaps **within the same region** the missionary activity factor enters marginally insignificant (see Figure 4), suggesting that when one measures the Christian-Muslim gap in the same district it does not seem to vary systematically by the extent of historical missionary activity in the district. This pattern is consistent again with your prior that over time schools with missionary roots were more successful in sustaining educational opportunity to those in their catchment areas with the qualification that this pattern seems to apply to all in the catchment areas irrespective of their religious identity. Of course, this is a descriptive and a not a causal pattern. Figure 4 shows that segregation is the most important correlate of the within-region interreligious gap in IM.

Motivated by your comment, we highlight these patterns early, the role of segregation and internal migration. Specifically, in the second paragraph of the “Introduction” we write: “Given the explanatory power of religious segregation and regions’ causal effects on adherents of all creeds, we conclude with a primer on the internal migration patterns across faiths. Christians are considerably more likely to emigrate and exploit opportunities outside their birthplace. Muslims’ low propensity to move accentuates their initial educational disadvantage, as they reside in remote places, far from the capital and the coastline, with limited colonial missionary activity and transportation investments.”

Third, we have added a new set of results zooming into internal migration and differences across faith to advance on the mechanisms. Please see our discussion in your last comment below.

Comment. Dual religious affiliation. *“The authors divide the population into three categories: Christian, Muslim, and Animist. As many Africanists have pointed out (e.g. J.D.Y. Peel’s book Aladura) animist prayer churches are often complementary religious identifications within the same person and not substitutes. It could well be the case that the more educated the young generation, the more likely they will report the “Christian” complement of their dual identification rather than the “Animist”.*

Reply. Thank you for your spot-on remark. We address it in the revision as follows:

First, we make an explicit reference to your point that one occasionally observes "mixed" religious affiliation, citing John D.Y Peel's book. Please see the paragraph on the “Inter-generational Transmission of Religious Affiliation” in the Methods (pages 1-2).

Second, we have formally explored religious affiliation and conversion within households using Census data [this also follows a suggestion of R1]. We formally explore religious conversion using Census data from IPUMS. Looking at multi-generational families, we estimate the probability of a young individual reporting being Muslim/Christian/Animist conditional on his/her parent’s creed. We find high inertia in the inter-generational transmission of Christianity and Islam. The intergenerational transmission coefficient exceeds 0.97 for Christians and Muslims across all birth cohorts, irrespective of the child’s education. Converting out of Christianity and Islam has been the exception in post-colonial Africa. The pattern is different among Animists. A sizable fraction of children born to Animist parents converts to Christianity –less so to Islam. Besides, whether a child of Animist parents converts to Christianity depends on education. Roughly half of the children with completed primary education born to Animist parents report they are Christian and the other half (46%) reports an indigenous religion. The corresponding statistics for children without primary schooling are 25% and 70%, respectively. This 25-percentage-points gap reflects how educational improvements of the Animist population in post-colonial Africa moved in tandem with its Christianization. This pattern may have implications for the religion-specific IM estimates since we categorize a household as Animist/Christian/Muslim if the child reports being Animist/Christian/Muslim, respectively. We replicated the benchmark analysis, reported in Figure 2, focusing on households where both the old and the young report the same religion. The estimates (not shown for brevity but reported below for your convenience) are similar, as the number of Animists is very small. See the Figures below.

Thanks to your remark -and a similar comment from R1, we discuss these issues in the revision. Please see the Subsection “Inter-generational Transmission of Religious Affiliation” in the Methods Section (pages 1-2). We have included a detailed discussion in the online SI Section B.3 (pages 17-19) and the SI Figures B1, B2, and B3.

Figure 1: Current draft: full sample

FIGURE NOTES

Figure 2: Religion of kid = previous generation religion

FIGURE NOTES

Conclusion and Migration, Comment. “*The admission by the authors that "Our study begets more questions than it answers" sums up my evaluation.*”

Reply. We thought it was helpful to pinpoint avenues for future research, as there is relatively little economics research on religion in Africa; this is especially the case when compared to the voluminous research on the political economy of ethnicity in Africa. As some reviewers found this conclusion problematic, we have reworked the last Section. In the revised draft's concluding Section, we summarize the main new results, discussing policy implications alongside some limited open questions that deserve -in our view- research exploration. Please see Section 8 [Discussion], page 14-15.

More importantly, the addition of a whole new Section on Internal Migration was prompted by the observation that indeed “our study begets more questions than it answers” and your remark to advance on the mechanisms and the suggestion of other referees to explore avenues that may help rationalize the new puzzle. We think that these new results complement the key findings on the significant differences in educational mobility between Christians and Muslims.

Please note that we are keen on exploring more mechanisms and channels. Still, given the space restriction and the paper's narrative on the patterns, the puzzle, and its origins, we must leave many intriguing questions for future work.

Thanks again for your helpful feedback and comments. Thinking about your constructive remarks helped us reposition our paper, add new results, and streamline our message. We sincerely appreciate the time you put into our work and your constructive feedback.

Reviewer Reports on the First Revision:

Referees' comments:

Referee #1 (Remarks to the Author):

I thank the authors for their thoughtful responses to the various queries raised. I think the flow has improved. The piece remains very dense and with so many separate findings that it requires a lot of work on the part of the reader. I don't see a way around this, only to say that the authors might have been better served if they had more space to elaborate rather than sending the reader to the SI many times. Obviously, I see the appeal of this outlet however, and the ability to get the core findings out to a broad audience. I think the article will make important contributions.

With that in mind I have a few follow-up comments with respect to take aways, terminology, sample, and references. I believe these are minor and straightforward to address.

Take aways:

The authors have made clearer what the punchlines are, but a) they might consider front loading the punchline of each section rather than stating it at the end, and b) state more clearly upfront what I see as both an important and fascinating finding, which is that despite a large C-M IM gap, this gap does not seem to impact economic well-being. In other words, I am still not quite satisfied with the discussion of the stakes – we care about the C-M IM gap not primarily because it suggests Muslims are economically worse off than Christians as a result of educational disadvantage. In fact the authors state more clearly in the response letter than they do in the paper, "Seen in this light the considerable educational gaps both in levels and in IM between the two groups suggest that we need to revisit the differential economic role of education for each group." This is huge. SSA has some of the lowest levels of education and income per capita, and a large literature suggests that human capital (measured by years of education, etc) matters for economic growth, and that there are substantial economic returns to schooling. Large amounts of money, as a percent of government spending, are going toward education in these countries. If it is somehow the case that education of this type does not impact income equally across groups that is worth demonstrating and exploring further in future work. It is of course up to the authors how to frame their contribution but this strikes me as a central implication of the work that is currently buried.

Terminology and other small writing notes:

- The authors seem to use traditional religion, indigenous religious, and Animism interchangeably (also, "Religions" need not be capitalized when following any of these terms, I believe). I could be mistaken but I don't think all African religions are Animist. Moreover, I would prefer that one term be selected and used throughout. Traditional seems to be more commonly used in the text than indigenous so perhaps the authors want to stick with that. There may be a case to be made for simply using the term "African religions," but maybe that is wading into debates where the authors don't care to venture.
- One of the other reviewers requested that IM be defined earlier on, and I agree, but I found it odd that where it emerges in the abstract it follows "educational mobility measures (IM)". The authors sometimes use the term "educational mobility" and sometimes "intergenerational mobility in education." This is nitpicky but IM seems like an odd abbreviation if the term being used throughout the text is educational mobility. If the correct term is "intergenerational mobility in

education” then perhaps use that term in the abstract when it is being introduced.

- On page 1 the authors write “Muslim children underperform, particularly in regions where they are numerous, a pattern that is not present for Christians and Animists.” Is this about the difference among Muslims in areas where they are and aren’t a majority? It’s not clear who they underperform relative to. If it is a comparison among Muslims, I would just say “Muslim children underperform in regions where they are numerous compared to where they are a minority, a pattern....”
- I’m not sure what is meant by “Muslim girls are more likely to be spouses of the head...” (7). Does this mean that there are more Muslim females under 18 who are spouses of household heads than Christian females under 18 in that position? Or Muslim females under 18 are more likely to be spouses than Muslim females over 18?

Sample and Data presentation

- The authors might say something about the countries omitted from the study (by necessity) and what implications this may have for our generalizing from their results to the rest of the region. I suspect something about state capacity and conflict may feature in census data availability (e.g. Somalia, Niger, Chad, Sudan), but also North Africa is wholly absent excepting Egypt, and if I am not mistaken some countries do not release data on religious affiliation (e.g. Tanzania in most government collected data sources, Nigeria does not ask this either on its census, an issue helpfully skirted here since IPUMS uses HH surveys for that country).
- Perhaps this is mentioned somewhere and I missed it, but what is the cutoff for the population/sample size of the minority groups in subnational analyses? I.e. how small of a sample is too small among the minority religious group to calculate a meaningful C-M difference at the subnational level?
- The authors might consider either a greyscale format or colors that are more friendly to those who are color-blind and/or black and white printing. If one prints in black and white the color references in the notes are not helpful.

Citations/References:

- While it is true there has been relatively little work on religion and education in Africa in recent years, I think there is more out there than the authors suggest (p1). On Nigeria alone, two notable books that come to mind are David Abernethy’s (1969) *The Political Dilemma of Popular Education: An African Case* and Babs Fafunwa’s (1970) *A History of Education in Nigeria*, both of which discuss the role of religion in explaining patterns of formal education in that case. This has likely been a much more frequently discussed topic in Nigeria than elsewhere given the colonial history and magnitude of the gap, the latter of which the authors also note on numerous occasions, but I share these citations to say that I am not sure justice has been done to those who have examined this subject to date.
- I think it may be worth noting that Platas in both her dissertation (also mentioned by another reviewer) and in the 2018 paper cited also found, using IPUMS data across numerous African countries, that the level difference in education between Christians and Muslims increased with the size of the district-level population (also within Malawi, where she also shows, as the authors do here, that income and HH factors did not seem to explain this gap). This was a core finding in both and accords with one of the key take aways presented in the paper.
- Relatedly, the citation to Platas (2018) on page 2 highlights the role of religious leaders, but the argument of the paper focuses more differences in social norms related to schooling across religious groups. Religious leaders are discussed in the origins of these norms but the argument is not centrally about them.
- Finally, as its title suggests, the argument in Bauer et al (2022) is as much about colonial responses to Muslim populations as vice versa. This might be reflected briefly in the author’s description of the work.

Referee #2 (Remarks to the Author):

Thanks to the authors for putting together a compelling revision. I am happy with all of the changes pursuant to my comments and am now confident that the paper will make a large contribution to the literature.

Editorial Note:

Reviewer 3 was unable to re-review the paper. However, based on the overlap between the concerns raised by Reviewer 3 and the other referees in the first round of review, we were able to determine that R3's concerns had been addressed.

Referee #4 (Remarks to the Author):

I have three reactions to this re-submitted manuscript.

First, as I believe all reviewers agreed, once the Tables and Figures were more carefully presented, the data on religious differences in Africa on intergenerational shifts in completing primary school is deeply impressive and will be a gold-mine for future researchers especially in following up the unanswered questions raised by the data that the authors highlight.

Second, the authors were assiduous in addressing all the queries and suggestions from the reviewers, and their answers were cogent. I have nothing more to say about the fulfillment of reviewer queries.

Third, perhaps because of the clarity that reviewers called for, the authors' explanation for the outcomes (inter alia, the significantly greater intergenerational upward mobility and (less so) smaller rates of downward mobility in education for African Christians) remains obscure. They write about "regions' causal effects" but I find it hard to picture how a region can have causal effect (after all, it is not a treatment). I think there is a simple story here, one looking at investment decisions by African families, that gets hidden from the readers' view. The explanatory variable (to take the major outcome of interest) is the expected returns to migration to a higher quality district. The differential returns to migrating lead to the higher likelihood of Christians with illiterate parents (i.e. no completion of primary school) more so than Muslims, to migrate to regions that have higher rates of educational achievement. And in this richer environment for education, their children are more likely to complete primary school. As for measuring cost, the data reveal that the geographic distance to a higher quality district is lower for Christians, lowering the cost of moving to a high quality district. Other costs may be relevant: we don't know about reduction of school fees and other inducements offered at the local level, again enhancing the returns to migration. Also, no data are provided on the size of districts, but larger-sized districts in Muslim areas would also raise the costs of migrating to another district.

But there is a problem with this theory of optimum investment in migration. We're told (in the balance tests) that "there are no major differences between Christians and Muslims in living conditions and access to basic necessities in Afrobarometer Surveys and differences in income appear negligible in the Pew Research Center (2016b) surveys." This is basically telling us (I think) that the returns to educational investment at the primary level for Africans are negligible. If this is so, there is a problem in motivating the paper. Why should we care about differential IM if there is no connection to social welfare?

In sum, I can recommend publication for the public good provided by the data, the neat specification of the dependent variable (IM), the descriptive information embedded in many of the tables and figures, and the serious fulfillment of all issues raised by four reviewers. However, the authors have not convinced me why the differential values on IM are important, and if they are, what explains the individual decisions by non-literate Christian parents to migrate and to fork out school fees to get their kids through primary school while this is much less the case for Muslim and Traditional Religion parents (who are more likely than Christians to bring up children with less education than theirs). If the whole story is "cost", the authors should tell us that.

Author Rebuttals to First Revision:

Religion and Educational Mobility in Africa. *Nature*-2021-14644.

Reply Referee 1 (second round of comments).

Thank you for your constructive feedback. We appreciate the time you spent reviewing the revised draft and the “*follow-up comments with respect to takeaways, terminology, sample, and references.*”

In the revision, we have taken on board your suggestions. Below we copy your comments (in *italics*) and point where each is addressed. We also indicate (with underlined fonts) the part of the paper, main body, methods, and Supplementary Information where we discuss your remark.

Introductory Comment. *I thank the authors for their thoughtful responses to the various queries raised. I think the flow has improved. The piece remains very dense with so many separate findings that it requires a lot of work on the reader's part. I don't see a way around this, only to say that the authors might have been better served if they had more space to elaborate rather than sending the reader to the SI many times. Obviously, I see the appeal of this outlet, however, and the ability to get the core findings out to a broad audience. I think the article will make important contributions.*

Reply. We appreciate your generous assessment, constructive guidance, and overall positive reaction to our work. We have further edited the paper, making it easier (hopefully) for the reader to navigate the various results. In particular, we have streamlined the introduction, redrafted the conclusion (Discussion), and, per *Nature* style, added a quite detailed introductory summary.

Comment 1. Take Aways

“The authors have made clearer what the punchlines are, but a) they might consider front-loading the punchline of each section rather than stating it at the end, and b) state more clearly upfront what I see as both an important and fascinating finding, which is that despite a large C-M IM gap, this gap does not seem to impact economic well-being. In other words, I am still not quite satisfied with the discussion of the stakes – we care about the C-M IM gap, not primarily because it suggests Muslims are economically worse off than Christians as a result of educational disadvantage. In fact, the authors state more clearly in the response letter than they do in the paper, “Seen in this light, the considerable educational gaps both in levels and in IM between the two groups suggest that we need to revisit the differential economic role of education for each group.” This is huge. SSA has some of the lowest levels of education and income per capita, and a large literature suggests that human capital (measured by years of education, etc.) matters for economic growth and that there are substantial economic returns to schooling. Large amounts of money, as a percent of government spending, are going toward education in these countries. If it is somehow the case that education of this type does not impact income equally across groups, that is worth demonstrating and exploring further in future work. It is, of course, up to the authors how to frame their contribution, but this strikes me as a central implication of the work that is currently buried.

Reply. Following your suggestion, we have rewritten parts of the paper to better frame our contribution. Regarding your initial remark (part (a)) to stress clearer the results, we have done the following.

- We have redrafted the Summary Paragraph (per *Nature* style), where we highlight the main takeaways. Specifically, in the concluding sentences of the summary, we follow your suggestion and stress the following takeaways: (i) The policy implications of our results, related to the vast educational investments by African governments and international policy institutions. (ii) Given the uncovered evidence, we need to rethink the private and social returns to schooling across religious affiliations, a critical omission of the large literature on Mincerian returns to education. Please see the new introductory summary (abstract), especially the concluding sentences.
- Besides, we comment on this issue also in the last paragraph of the conclusion. Please see the reworked Discussion.
- We have reworked the introduction to stress the main results more. So now the reader gets a summary of the takeaways up front. Please see Introduction, second paragraph. While there is some overlap with the new introductory summary, having this results overview paragraph allows us to stress further the main takeaways.
- We have kept the summary paragraphs at the end of each (sub)section. We added those in response to your earlier remark to summarize the many results as we discussed them. This has been a helpful suggestion, as our paper presents many regularities. We considered moving these summary paragraphs at the beginning of each (sub)section, but to us, it looked kind of non-standard to summarize the results before presenting them. As we now summarize them in

the reworked summary/abstract and the revised introduction, we feel that it is helpful for the reader to maintain the brief takeaways of each section in the end.

Your second remark (part (b)) relates to the importance of schooling for growth (at the individual and social level), and our result that Christian-Muslim differences in proxies of household well-being (rural-urban status, profession, industry of employment) are minor. You urge us to bring this result earlier and stress the need for “*exploring further in future work.*” The revision implements both suggestions.

- First, we refer to works showing that individual (Mincerian) returns to education -and primary schooling in particular- are considerable overall in low-income countries and Africa. We do so in the very beginning. Please see the new introductory summary (very beginning). We have also added a discussion on Mincerian returns in Africa (and other low-income countries), in the Appendix. Please see SI A.1 “Returns to Schooling in Africa.”
- Second, as works on returns to schooling do not distinguish by religion, following your suggestion (at the end of your comment), we stress the need for follow-up research. As referee 4 had a similar remark, we discuss this issue in two parts of the revised manuscript. (i) At the end of the new summary. Please see the new summary/abstract (ending). (ii) In the concluding section (Discussion). Please see the reworked concluding paragraph (Discussion).
- Third, following your suggestion, we stress that given considerable government investments, often funded by aid agencies and international organizations, on education, we need direct evidence on the impact of educational interventions (e.g., school construction programs, abolition of fees) on inequality across

religious lines. As this is a great idea, we discuss it at the end of the new introductory Summary and the Discussion (Conclusion).

Comment 2. Terminology and other small writing notes

Traditional Religions. *The authors seem to use traditional religion, indigenous religion, and Animism interchangeably (also, “Religions” need not be capitalized when following any of these terms, I believe). I could be mistaken, but I don’t think all African religions are Animist. Moreover, I would prefer that one term be selected and used throughout. Traditional seems to be more commonly used in the text than indigenous, so perhaps the authors want to stick with that. There may be a case to be made for simply using the term “African religions,” but maybe that is wading into debates where the authors don’t care to venture.*

Reply Thanks. Following your suggestion, we first state explicitly that we use the names Traditional African Religions and Animists interchangeably, noticing that sometimes local religions are coined as folk. [The terminology Animists is widely accepted to characterize Africans adhering to indigenous religions. It is also helpful to use *Animists*, one word, compared to Africans following/adhering to traditional religions.] Please see Introduction, beginning of the second paragraph on page 1 (Preview). Second, in the revised draft, we mainly refer to Africans (people) adhering to Traditional (African) religions. Third, we do not capitalize religions, using the term Traditional religions.

Define Intergenerational Mobility. *One of the other reviewers requested that IM be defined earlier on, and I agree, but I found it odd that where it emerges in the abstract, it*

follows “educational mobility measures (IM).” The authors sometimes use the term “educational mobility” and sometimes “intergenerational mobility in education.” This is nitpicky but IM seems like an odd abbreviation if the term being used throughout the text is educational mobility. If the correct term is “intergenerational mobility in education,” then perhaps use that term in the abstract when it is being introduced.

Reply. Thanks. Following your suggestion, we define intergenerational mobility in education in the New Summary Paragraph and again in the Introduction of the paper (page 1, paragraph 2). Besides, we define it again in Methods and Supplementary Information (SI). And in the definitions, we now write Absolute IM in Education.

Clarification 1. *On page 1, the authors write, “Muslim children underperform, particularly in regions where they are numerous, a pattern that is not present for Christians and Animists.” Is this about the difference among Muslims in areas where they are and aren’t a majority? It’s not clear to whom they underperform. If it is a comparison among Muslims, I would just say, “Muslim children underperform in regions where they are numerous compared to where they are a minority, a pattern....”*

Reply. Thank you for spotting this unclear sentence. We have rewritten it following your exact suggestion. Please see the second paragraph of the Introduction, page 1.

Clarification 2. *I’m not sure what is meant by “Muslim girls are more likely to be spouses of the head...” (7). Does this mean that there are more Muslim females under 18 who are spouses of household heads than Christian females under 18 in that position? Or Muslim females under 18 are more likely to be spouses than Muslim females over 18?*

Reply. Thank you for spotting this unclear sentence. We have rewritten it. It now reads: “14-18 years old Muslim girls are more likely to be spouses of the household head than Christian girls of the same age.” Please see Section 4, paragraph Household Features.

Comment 3. Sample and Data presentation

Sample. *The authors might say something about the countries omitted from the study (by necessity) and what implications this may have for our generalizing from their results to the rest of the region. I suspect something about state capacity and conflict may feature in census data availability (e.g., Somalia, Niger, Chad, Sudan), but also North Africa is wholly absent except Egypt. If I am not mistaken, some countries do not release data on religious affiliation (e.g., Tanzania in most government-collected data sources, Nigeria does not ask this either on its census, an issue helpfully skirted here since IPUMS uses HH surveys for that country).*

Reply. We are using all African countries with information on religion in the international vintage of IPUMS. Our analysis is based on Census information (and, in Nigeria, Household Surveys) spanning 21 African countries. However, as you point out, one may wonder how representative the country sample is. We thus added some discussion in the main paper, alongside a more extensive discussion in the Supplementary Information (SI). Specifically:

1. In the paper, we have added the following: “*While North Africa is under-represented, the sample includes both relatively richer (e.g., South Africa and Botswana with GDP p.c. of about 4,000-4,500 USD in 1995), and quite poor African countries (e.g., Ethiopia, Malawi, and Mozambique with GDP p.c. of*

about 250 USD in 1995). We have info on former British (Nigeria, Sierra Leone, Malawi), French (Burkina Faso, Senegal, Guinea), German-Belgian (Rwanda), and Portuguese (e.g., Mozambique) colonies and protectorates, besides Liberia and Ethiopia. The sample also includes countries with lasting civil wars (Sierra Leone, Mozambique, Rwanda, Liberia, and Ethiopia), using Censuses conducted after the end of hostilities.” Please see Data Paragraph in Section 2.

2. We have added a new Sub-section in the new Supplementary Information (Section B.5). We compare the two groups of countries, in-sample (in-IPUMS) and not in the sample (not in IPUMS), across income (GDP p.c.), years of schooling (using the Barro-Lee (2020) statistics), institutional quality measures (using data from the World Bank’s Governance Indicators), and proxies of state fragility and civil war intensity. There are no systematic differences between the two groups. This is because our sample includes both relatively rich, with sound institutions, governance, and state capacity countries, like South Africa, Botswana, and Egypt, but also poor, fragile, and conflict-prone countries, like Ethiopia, Mozambique, and Liberia. Please see the new results in Supplementary Information (SI), Section B.5, SI Tables B.6, B.7, B.8, and B.9.

Cutoffs for Religious IM. *Perhaps this is mentioned somewhere, and I missed it, but what is the cutoff for the population/sample size of the minority groups in subnational analyses? i.e., how small of a sample is too small among the minority religious group to calculate a meaningful C-M difference at the subnational level?*

Reply. Thanks for flagging this issue. We now discuss this in the Sample paragraph in Section 2. We follow the underlying census data for the country-level statistics as

closely as possible. For the regional statistics, we do not restrict the number of observations for estimating the C-M and the C-T gap in IM. However, in the Supplementary Information, we now report statistics requiring at least 10 observations (14-18-year-old individuals) for each religious group (Muslim, Traditional African religion, and Christian) to calculate the intergenerational mobility in education statistics -and the corresponding differences/gaps. Please see SI Table C.4, which gives summary statistics without any restriction (Panel A) and the corresponding statistics requiring at least ten individual observations from each religious group for calculating regional statistics of educational IM (Panel B).

Colors (Black and White). *“The authors might consider either a greyscale format or colors that are more friendly to those who are color-blind and/or black-and-white printing. If one prints in black and white the color references in the notes are not helpful.”*

Reply. Thanks for spotting this issue. Upon acceptance of the manuscript, we will liaise with the production team to address this issue following their guidelines.

Citations/References

Literature on Religion and Education in Africa. *While it is true there has been relatively little work on religion and education in Africa in recent years; I think there is more out there than the authors suggest (p1). On Nigeria alone, two notable books that come to mind are David Abernethy’s (1969) *The Political Dilemma of Popular Education: An African Case* and Babs Fafunwa’s (1970) *A History of Education in Nigeria*, both of discussing the role of religion in explaining patterns of formal education*

in that case. This has likely been a much more frequently discussed topic in Nigeria than elsewhere, given the colonial history and magnitude of the gap, the latter of which the authors also note on numerous occasions, but I share these citations to say that I am not sure justice has been done to those who have examined this subject to date.

Reply. Thank you for bringing to our attention the works of these scholars. First, to avoid broader statements we have rephrased that *research in economics* on the political economy of African development primarily focuses on ethnicity rather than religion. [We think this is a fair assessment of the economics literature.] Second, we state that there have been country-specific works on religion and education in Nigeria and some other countries in political science, citing the two books you recommended, alongside the work of David Laitin and John David Yeadon Peel on the Yoruba. Please see our revised introduction (pages 1-2).

Platas' works. *I think it may be worth noting that Platas, in both her dissertation (also mentioned by another reviewer) and in the 2018 paper cited, also found, using IPUMS data across numerous African countries, that the level difference in education between Christians and Muslims increased with the size of the district-level population (also within Malawi, where she also shows, as the authors do here, that income and HH factors did not seem to explain this gap). This was a core finding in both and accords with one of the key takeaways presented in the paper. Relatedly, the citation to Platas (2018) on page 2 highlights the role of religious leaders, but the argument of the paper focuses more on differences in social norms related to schooling across religious groups. Religious leaders are discussed in the origins of these norms, but the argument is not centrally about them.*

Reply. First, following your suggestion, we now cite Platas' dissertation (The Religious Roots of Inequality in Africa, Stanford University, Department of Political Science, May 2016). Second, we elaborate on Platas' working paper (Culture and the Persistence of Educational Inequality: Lessons from the Muslim-Christian Education Gap in Africa, 2018), discussing both her descriptive results on school attendance of children aged 6-17 and religion across 11 African countries in 2017 (Figures 1-2) and her micro-evidence (based on self-conducted interviews and qualitative data) linking Christian-Muslim educational differences to social norms and religious leaders' role in Malawi (Figures 3-8 and Table 1). We also comment briefly on her work in Uganda, reported in her Stanford University Ph.D. thesis. Please see Introduction (extended paragraph 3). Moreover, in the edited conclusion, we refer to Platas (2018), alongside Bauer et al. (2022). Please see the concluding paragraph of the Discussion.

Bauer et al. (2022). *"Finally, as its title suggests, the argument in Bauer et al. (2022) is as much about colonial responses to Muslim populations as vice versa. This might be reflected briefly in the author's description of the work."*

Reply. Apologies. We have now clarified the contribution of Bauer, Platas, and Weinstein (*World Development*, 2022). We write: *"Bauer et al. (2022) show that Muslims under-perform (in education and health) in areas with strong pre-colonial Islamic states (e.g., in Northern Nigeria and Cameroon, Senegal) due to weak penetration of the colonial state and limited school and good public investments by Christian missionaries"*. Please see paragraph Segregation in Section 6. Moreover, we refer to Bauer et al. (2022), alongside Platas (2018) in the edited conclusion, when discussing

the need to dig into the mechanisms, teasing apart social norms, religious leaders' roles, and history. Please see the concluding paragraph of the Discussion.

Thanks again for your second round of constructive and detailed feedback on the revised draft of our work. We sincerely appreciate the time you put into our paper and your constructive comments. We hope that you deem the revisions satisfactory, although we would be happy to have another go if you have additional remarks. Thanks again.

Religion and Educational Mobility in Africa. *Nature*-2021-14644.

Reply Referee Report 4

We deeply appreciate the time you spent going after the revised draft. We are thankful for your feedback and your positive attitude toward our work.

Below we copy your three comments/reactions (in *italics*) and detail our response.

Results. *First, as I believe all reviewers agreed, once the Tables and Figures were more carefully presented, the data on religious differences in Africa on intergenerational shifts in completing primary school is deeply impressive and will be a gold mine for future researchers especially in following up the unanswered questions raised by the data that the authors highlight.*

Reply. Thanks for your nice remarks. As you rightly point out, the patterns we establish with the newly-compiled data on religious intergenerational mobility (IM) in education across African countries and regions raise many questions that will hopefully motivate subsequent research. To facilitate further explorations, we have done the following.

- First, not only will we release the data on education, upward and downward IM, and the demographics across all African countries and regions to the research community, but we are also developing an interactive website portal where scholars and the public can access the data, make useful illustrations, and infographics.
- Second, we have reworked the conclusion discussing the possible avenues for future research and have taken on board the useful suggestions you and R1 have given. Specifically, we discuss the need to better understand the private

and social returns to education by religious affiliation. Please see the revised Discussion. We also mention this core issue in the concluding sentence of the Extended Summary.

- Third, thanks to your earlier suggestion (in the initial round), we have explored one mechanism underlying the considerable gaps in religious IM across regions, internal migration. Despite the constraints of *Nature's* style and size, the revised draft includes a Section documenting: (i) considerable differences in internal migration across religious lines across almost all African countries in our sample; (ii) mirroring the patterns of IM in education, the significant inter-religious gaps in internal migration go hand in hand with religious segregation. Please see Section 7 (pages 13-15) and the results in Figures 6-7.

Revisions. *Second, the authors were assiduous in addressing all the queries and suggestions from the reviewers, and their answers were cogent. I have nothing more to say about the fulfillment of reviewer queries.*

Reply. Thanks for your nice comments on the revisions. We worked hard, also using the updated data from IPUMS, to address the many spot-on comments raised by the four referees. Besides addressing all comments and adding a new section on internal migration, we have managed to keep the paper's size compatible with *Nature's* style, a non-trivial task, as you and the other three reviewers had important comments and helpful remarks.

We are splitting your third remark below to answer it more succinctly.

Interpretation of Findings 1. Regions Causal Effect. *Third, perhaps because of the clarity that reviewers called for, the authors' explanation for the outcomes (inter alia, the significantly greater intergenerational upward mobility and (less so) smaller rates of downward mobility in education for African Christians) remains obscure. They write about "regions' causal effects" but I find it hard to picture how a region can have a causal effect (after all, it is not a treatment).*

Reply. Thanks for flagging this issue. We understand that our discussion may have been unclear on what we mean by "*regional childhood exposure causal effects*". Hence, we tried to streamline the text and clarify this issue in the revised draft. The considerable differences in upward and downward intergenerational mobility (IM) in education may originate from two (broad) distinct forces. First, families valuing education most or facing lower costs (as your comment nicely suggests) sort into districts/areas/neighborhoods offering more/better opportunities for completing primary education (*spatial sorting/selection*). Second, some regions/districts/neighborhoods offer better educational opportunities (perhaps by providing more and better schools, with lower fees and better infrastructure) and thus affect education in a causal sense ("*regional childhood exposure effects*"). Following the intuitive and much-cited work of Chetty *et al.* (QJE 2018a, QJE 2018b) and a considerable body of follow-up work (e.g., Chetty et al. (QJE 2020), we refer to the two mechanisms as "*spatial sorting/selection*" and "*childhood regional causal effects*." [Others call them "place effects". See, please Finkelstein, Gentzkow, and Williams (*American Economic Review*, 2021), who develop

a similar mover's design to approximate the effect of locations on elderly mortality across US regions.]

As your statement rightly implies, the Chetty and Hendren (*QJE* 2018a) methodology, which relies on children whose families move from low (high) to high (low) educational IM districts below the age of primary school completion (12-14 years of age), does not identify which regional features are responsible for shaping educational attainment. If you wish, exposure to a given region reflects the bundled treatment from the many different regional attributes. As you correctly anticipate, these could reflect lower school fees, better teachers, higher quality schooling infrastructure, lower pupil-teacher ratios, and local labor markets with higher returns to schooling, among many others. We also refer to the working paper of Torsten Figueiredo Walter (2022), who documents wide spatial differences in educational quality across low-income countries (as this evidence is in line, we think, with your intuition).

Thanks to your remark, we elaborate on these issues in Section 5; in particular, please see the Empirical Design. To avoid any confusion (and prevent extrapolating economists jargon in a general-science outlet), we have also changed the Section's title from "The Causal Effects of Regions" to "Spatial Sorting and Childhood Regional Exposure Effects".

Moreover, we refer to this issue at the beginning of Section 6 (Religious Educational IM Gaps across African Regions), stating that *"Besides, the correlational analysis complements the results in Figure 6, as the movers' design that distinguishes spatial sorting from childhood exposure effects does not pin down which regional features correlate with religious IM."* Please see the revised Section 6.

We hope these revisions clarify what one can learn from the movers' design (and what one cannot).

Interpretation of Findings 2. Regions Causal Effect. *"I think there is a simple story here, one looking at investment decisions by African families, that gets hidden from the readers' view. The explanatory variable (to take the major outcome of interest) is the expected returns to migration to a higher-quality district. The differential returns to migrating lead to the higher likelihood of Christians with illiterate parents (i.e., no completion of primary school), more so than Muslims, to migrate to regions that have higher rates of educational achievement. And in this richer environment for education, their children are more likely to complete primary school. As for measuring cost, the data reveal that the geographic distance to a higher-quality district is lower for Christians, lowering the cost of moving to a high-quality district. Other costs may be relevant: we don't know about the reduction of school fees and other inducements offered at the local level, again enhancing the returns to migration. Also, no data are provided on the size of districts, but larger-sized districts in Muslim areas would also raise the costs of migrating to another district."*

Reply. Thanks for your spot-on remark. We have done the following to address it.

- First, we state in Section 7: "Migration decisions reflect the associated costs and benefits of doing so that may differ across religious lines. Muslims most likely face higher migration costs as they reside in relatively remote regions with limited investments. Thus, in Figure 6 - panel (b), we report internal migration shares, netting out the mean at the individual's birthplace (weighted by the region's

population in the country) to account for inter-religious differences in residence (Ext. Data 3). Christian-Muslim and Christian-Animist differences in internal migration are evident, even when we compare individuals born in the same district. Only in Uganda, South Africa, and Rwanda, where Muslims are (small) minorities, do Muslims move from the same birth region at a higher rate than Christians. Which other factors shape the uncovered differences, economic, cultural, or institutional (interacted with religion), remains an open issue.” Please see Section 7 (Differential Internal Migration), pages 13-14.

- Second, we stress in the Summary and Discussion the need to estimate individual (Mincerian) returns to education by religious affiliation, a critical omission of the current research. Please see Discussion (Section 8) and the last sentence of the summary.
- Third, in the reworked conclusion (Discussion), we also mention the importance of better understanding migration decisions. Specifically, we write: *“Third, as millions are moving to Africa’s new megacities, research should explore, and policymakers should rethink potential heterogeneity along religious lines of the economic return to migration, linking it with migration costs as well as labor markets both at the origin and the destination.”* Please see Discussion (Section 8)
- Finally, regarding your inquiry/note on the size of districts. We now provide summary statistics on each country's district size (land area, population). Moreover, motivated by your suggestion, we performed a test of means comparison of Muslim versus Christian-majority/plurality districts in terms of area.

Within countries, the two groups have no significant differences in the district area. Please see SI Table C.3 and the discussion in the main paper in the “Regional Patterns” Paragraph in Section 3.

Interpretation of Findings 3. Why do we care? *But there is a problem with this theory of optimum investment in migration. We’re told (in the balance tests) that “there are no major differences between Christians and Muslims in living conditions and access to basic necessities in Afrobarometer Surveys, and differences in income appear negligible in the Pew Research Center (2016b) surveys.” This is basically telling us (I think) that the returns to educational investment at the primary level for Africans are negligible. If this is so, there is a problem in motivating the paper. Why should we care about differential IM if there is no connection to social welfare?*

Reply. Thank you for pushing us on this issue. Your question, “*Why should we care about differential IM if there is no connection to social welfare?*” is spot on.

Referee 1 was also intrigued by the observation that the documented educational IM gaps do not seem to map to substantial well-being differences between Muslims and Christians. He/she thought that this actually is one of the central implications of our (admittedly too many) findings. Convinced also by your reaction, we have done the following to highlight but also caveat this implication.

- First, we stress the need better to understand this somewhat paradoxical finding from the onset. As Nature’s guidelines for the Extended Summary urges authors to add in the conclusion implication for policy and future research, we now discuss this issue in the Summary’s concluding phrase as follows: “*As African*

governments and international organizations invest heavily in educational programs, our findings highlight the need to understand better the private and social returns to schooling across faiths in religiously segregated communities and carefully think about religious inequalities in the take-up of educational policies.”

- Second, we discuss this issue in the Conclusion. Specifically, in the reworked Discussion, we write, “*despite the first-order differences in education both in levels and in IM between African Christians and Muslims, the (admittedly less precise) available evidence does not suggest an equally stark interfaith gap in well-being and occupational specialization. This pattern suggests that we need to revisit the differential economic role of education for each denomination. The large body of research on returns to schooling does not study the religious dimension, which documents higher returns to primary education in low-income settings and Africa in particular (Psacharopoulos and Patrinos, 2018), (Montenegro and Patrinos, 2014). See the related discussion in SI Section A.1. Estimating faith-specific private and social returns to schooling, both actual and perceived, in the context of religiously segregated local labor markets and religion-specific risk-sharing institutions appears crucial.*- Third, we now refer to works on returns to schooling, and primary education, both across the world and Africa, in the Summary and the Discussion Sections. We have also added a discussion on Mincerian returns in Africa in the Appendix. Please see SI A.1 “Returns to Schooling in Africa.” The literature documents sizable private economic and non-pecuniary returns to (primary) schooling

across Africa. Critically, this voluminous literature has not considered whether these returns vary along religious lines.

- Fourth, we caveat that the finding that household economic features do not explain much of the religious differences in intergenerational mobility in education does not necessarily imply that individual returns to schooling are nil or differ for Africans of a different faith. In particular, following your remark, we mention that the sectoral and occupational categorization of the household head in IPUMS is coarse and may mask differences within broad sectors. For example, Muslims may work in subsistence agriculture, while Christians may work in larger farms with better infrastructure. Christians may work in skill-intensive services, while Muslims in labor-intensive ones. Thanks to your remark, we caution the relevant part of the paper. Please see Section (page 8), paragraph Household and Parental Occupational Differences, where we also clarify the coarseness of the underlying data, which corresponds to the household head, not the 14–18-year-old individual.

Thanks again for your helpful feedback and comments. Thinking about your constructive remarks helped us reposition our paper, add new results, and streamline our message. We sincerely appreciate the time you put into our work. And we also appreciate your positive reaction to our work.

Reviewer Reports on the Second Revision:

Referees' comments:

Referee #1 (Remarks to the Author):

I appreciate the time and thought that has gone into the multiple rounds of revisions and I am satisfied with the responses, additions, and edits the authors have made. I look forward to seeing the article in print.

Referee #4 (Remarks to the Author):

Alesina et al "Religion and Educational Mobility in Africa"
Re-re-review of R4 for Nature

The submission provides an extraordinary public good (cross-sectional data on intergenerational educational mobility (IM) in Africa) with a large number of descriptive statistical findings on the mechanisms that drive religious variance in IM that are provocative and worthy of future testing. For Nature, the headline would be something like "Christians maintain significant educational advantages in Africa". Exploiting significant variation on school attainment, the authors provide interesting clues as to causes and mechanisms.

But clues are not slam-dunk findings. Mainly repeating myself from earlier reviews, I give two examples where the reader is left somewhat bewildered about their results and claims.

On the effects of living in a particular region, they write, "We construct novel religion-specific educational mobility measures since independence across African countries and regions and explore their origins. Three regularities emerge. First, there are significant differences in IM between Christians and Muslims, even comparing Africans living in the same District...Second...Muslims' comparatively low internal mobility accentuates their educational IM deficit, as they...reside in less urbanized, far from the capital and the coastline areas with limited infrastructure." It is hard for this reader to reconcile the IM differences even when the religious groups living in the same region and then using region to explain why productive migration is lower for Muslims.

On the economic returns to IM, they write "in line with these patterns, differences between Christians and Muslims in living conditions and access to necessities in Afrobarometer Surveys and household income in the surveys¹¹ are also small...broad economic features play no role in the Christian-Muslim upward gap." How to account for this and still motivate the paper by expected returns from educational investment. In their response to R1, they underline the issue by acknowledging that "Christian-Muslim differences in proxies of household well-being (rural-urban status, profession, industry of employment) are minor." They attribute this in part to the coarseness of their data (which is a bad move, as it lends suspicion on their good data) but then point to a literature in the SI showing significant economic gains to education in Africa. This literature review is not fully convincing. For one, in several of these studies, the results are upwardly biased as those with full primary education are positively selected (from wealthier parents or households). Second, most (what are referred to as Mincerian) estimates of the returns to education only include wage workers, which poses a particular problem in Sub-Saharan Africa, where fewer than 24 percent of workers receive wages or salaries. They do not report studies with more ambiguous results. For example, one of the papers they cite to make their case is Montenegro and Patrinos (2014) who report significant gains to education in Ethiopia. However, see A. Singh (2019, DOI: 10.1093/jeea/jvz033) where we observe minimal cognitive returns to primary education in their African example (Ethiopia), far less than countries (India, Peru, and Vietnam) from other regions. Similarly, examination of international test scores (Sandefur 2018,

(<https://www.sciencedirect.com/science/article/abs/pii/S0272775717300055>) also reveal very low returns to primary education in Eastern and Southern Africa, lower than predicted by GDP/cap. See also Laitin and Ramachandran (APSR 2016; JDE 2022) who provide evidence that the cognitive returns to African primary education are disappointing given the resources brought to bear. Indeed DHS surveys of African primary students reveal only paltry percentages who can read a full sentence in their language of choice. On balance, see the SI in Evans and Acosta (2020, <https://academic.oup.com/jae/article/30/1/13/5999001>) that provides a comprehensive literature review, with some (but not consistent) support for high returns to additional years of primary education in Africa.

Combining these two issues, I wonder if there is another way to explain low Muslim migration to high IM regions if the concomitant access to education seems to show (in their data) low rewards, viz. that Muslims are rational consumers of public goods take-up and compared to opportunity costs, aren't convinced of the primary school ROI for their children. The authors do not consider this possibility. At best, they should be more modest in their assumption that Muslims are suffering substantial economic losses due to lower IM.

I have learned a great deal from this article. It pays homage to Alberto Alesina, one of the most dynamic and creative scholars of economic development of the past generation, and it would be a fitting testimony to his great influence to have his final paper on your pages.

David Laitin